# Metropolis Sampling for Constrained Diffusion Models

**Nic Fishman**
University of Oxford

**Leo Klarner**
University of Oxford

**Emile Mathieu**
University of Cambridge

**Michael Hutchinson**
University of Oxford

**Valentin De Bortoli**
ENS Ulm

`{njwfish,leojklarner}@gmail.com`

## Abstract

Denoising diffusion models have recently emerged as the predominant paradigm for generative modelling on image domains. In addition, their extension to Riemannian manifolds has facilitated a range of applications across the natural sciences. While many of these problems stand to benefit from the ability to specify arbitrary, domain-informed constraints, this setting is not covered by the existing (Riemannian) diffusion model methodology. Recent work has attempted to address this issue by constructing novel noising processes based on the reflected Brownian motion and logarithmic barrier methods. However, the associated samplers are either computationally burdensome or only apply to convex subsets of Euclidean space. In this paper, we introduce an alternative, simple noising scheme based on Metropolis sampling that affords substantial gains in computational efficiency and empirical performance compared to the earlier samplers. Of independent interest, we prove that this new process corresponds to a valid discretisation of the reflected Brownian motion. We demonstrate the scalability and flexibility of our approach on a range of problem settings with convex and non-convex constraints, including applications from geospatial modelling, robotics and protein design.

## 1 Introduction

In recent years, denoising diffusion models (Sohl-Dickstein et al., 2015; Song et al., 2019; Song et al., 2021; Ho et al., 2020) have emerged as a powerful paradigm for generative modelling, achieving state-of-the-art performance across a range of domains. They work by progressively adding noise to data following a Stochastic Differential Equation (SDE)—the forward *noising* process—until it is close to the invariant distribution of the SDE. The generative model is then given by an approximation of the associated time-reversed *denoising* process, which is also an SDE whose drift depends on the gradient of the logarithmic densities of the forward process, referred to as the *Stein score*. Building on the success of diffusion models for image generation tasks, De Bortoli et al. (2022) and Huang et al. (2022) have recently extended this framework to a wide range of Riemannian manifolds, enabling the specification of inherent structural properties of the modelled domain. This has broadened the applicability of diffusion models to problems in the natural and engineering sciences, including the conformational modelling of small molecules (Jing et al., 2022; Corso et al., 2022), proteins (Trippe et al., 2022; Watson et al., 2022; Yim et al., 2023) and robotic platforms (Urain et al., 2022).

However, in many data-scarce or safety-critical settings, researchers may want to restrict the modelled domain even further by specifying problem-informed constraints to make maximal use of limited experimental data or prevent unwanted behaviour (Morris, 2002; Han et al., 2006; Thiele et al., 2013; Lukens et al., 2020). As illustrated in Figure 1, such domain-informed constraints can be naturally represented as a *Riemannian manifold with boundary*. Training diffusion models on such constrained manifolds is thus an important problem that requires principled noising processes—and corresponding discretisations—that stay within the constrained set.

37th Conference on Neural Information Processing Systems (NeurIPS 2023).

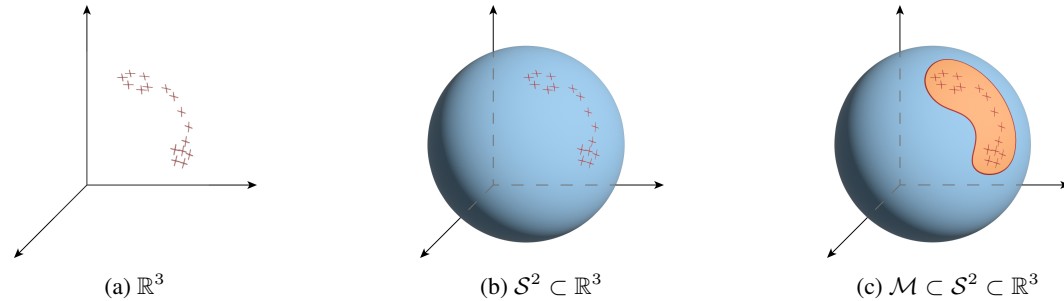

|  (a) $\mathbb{R}^3$  |  (b) $\mathcal{S}^2 \subset \mathbb{R}^3$  |  (c) $\mathcal{M} \subset \mathcal{S}^2 \subset \mathbb{R}^3$  |

Figure 1: When operating in data-scarce settings, it may often be beneficial to specify as much prior knowledge of the modelled domain as possible. Consider a distribution over a subset $\mathcal{M}$ of the unit sphere $\mathcal{S}^2 \subset \mathbb{R}^3$. While a Euclidean diffusion model can approximate the distribution in $\mathbb{R}^3$ (a), directly modelling it on $\mathcal{S}^2$ can make learning significantly easier (b). Restricting the problem space even further by only constructing a diffusion model on $\mathcal{M}$ can lead to even better performance (c).

Recent work by Fishman et al. (2023) has attempted to derive such processes and extend the applicability of diffusion models to inequality-constrained manifolds by investigating the generative modelling applications of classic sampling schemes based on log-barrier methods (Kannan et al., 2009; Lee et al., 2017; Noble et al., 2022; Kook et al., 2022; Gatmiry et al., 2022; Lee et al., 2018) and the reflected Brownian motion (Williams, 1987; Petit, 1997; Shkolnikov et al., 2013). While empirically promising, the proposed algorithms can be computationally and numerically burdensome, and require bespoke implementations for different manifolds and constraints. Concurrently, Lou et al. (2023) have investigated the use of reflected diffusion models for image applications. They focus on the high-dimensional hypercube, as this setting admits a theoretically grounded treatment of the *static thresholding* method which is widely used in image models such as Saharia et al. (2022). More recently, Liu et al. (2023) have investigated the use of log-barrier-based mirror maps to transform a constrained domain into an unconstrained dual space for applications in image watermarking. While both methods exhibit robust scaling properties and impressive results, they only consider convex subsets of Euclidean space and do not extend to more general manifolds.

Here, we propose a new method for generative modelling on constrained manifolds based on a Metropolis-based discretisation of the reflected Brownian motion. The Metropolised process' chief advantage is that it is lightweight: the only additional requirement over those outlined in De Bortoli et al. (2022) that is needed to implement a constrained diffusion model is an efficient binary function that indicates whether any given point is within the constrained set. This Metropolised approximation of the reflected Brownian motion is substantially easier to implement, faster to compute and more numerically stable than the previously considered sampling schemes. Our core theoretical contribution is to show that this new discretisation converges to the reflected SDE by using the invariance principle for SDEs with boundary (Stroock et al., 1971). To the best of our knowledge, this is the first time that such a process has been investigated. We demonstrate that our method attains improved empirical results on diverse manifolds with convex and non-convex constraints by applying it to a range of problems from geospatial modelling, robotics and protein design.

## 2 Background

**Riemannian manifolds.** A Riemannian manifold is defined as a tuple $(\mathcal{M}, \mathfrak{g})$ with $\mathcal{M}$ a smooth manifold and $\mathfrak{g}$ a metric defining an inner product on tangent spaces. In this work, we will use the exponential map $\exp_x : \mathrm{T}_x\mathcal{M} \to \mathcal{M}$, as well as the extension of the gradient $\nabla$, divergence $\mathrm{div}$ and Laplace $\Delta$ operators to $\mathcal{M}$. All of these quantities can be defined in local coordinates in terms of the metric. The extension of the Laplace operator to $\mathcal{M}$ is called the Laplace-Beltrami operator, also denoted $\Delta$ when there is no ambiguity. Using $\Delta$, we can define a Brownian motion on $\mathcal{M}$, denoted $(\mathbf{B}_t)_{t \geq 0}$ and with density w.r.t. the volume form of $\mathcal{M}$ denoted $p_t$ for any $t > 0$. We refer to Appendix B for a more detailed exposition, to Lee (2013) for a thorough treatment of Riemannian manifolds and to Hsu (2002) for details on stochastic analysis on manifolds. In the following, we consider a constrained manifold $\mathcal{M}$ defined by

$$\mathcal{M} = \{x \in \mathcal{N} \ : \ f_i(x) < 0, \ i \in \mathcal{I}\}, \tag{1}$$

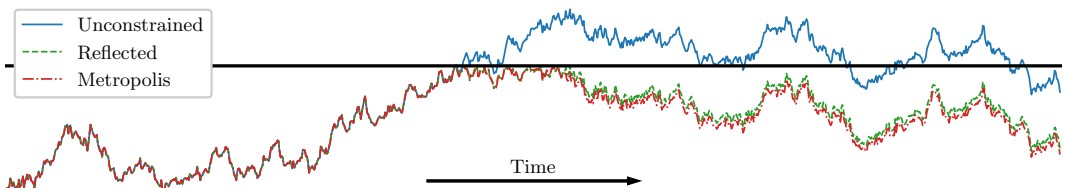

Figure 2: Visual comparison of a discretisation of the unconstrained Brownian motion (blue) and two discretisations of the reflected Brownian motion: one based on a reflection scheme (green) and the other based on our Metropolis sampler (red). The Metropolised trajectory is very close to that of the reflected one while being significantly easier to implement and cheaper to compute.

where $(\mathcal{N}, \mathfrak{g})$ is a Riemannian manifold, $\mathcal{I}$ is an arbitrary finite indexing family and for any $i \in \mathcal{I}$, $f_i \in \mathrm{C}(\mathcal{N}, \mathbb{R})$. Since $\mathcal{I}$ is finite and $f_i$ continuous for any $i \in \mathcal{I}$, $\mathcal{M}$ is an open set of $\mathcal{N}$ and inherits its metric $\mathfrak{g}$. This captures simple Euclidean polytopes and complex constrained spaces like Figure 1.

**Denoising diffusion models.** Denoising diffusion models (Song et al., 2019; Ho et al., 2020; Song et al., 2021) work as follows: let $(\mathbf{X}_t)_{t \in [0,T]}$ be a *noising process* that corrupts the original data distribution $p_0$. We assume that $(\mathbf{X}_t)_{t \geq 0}$ converges to $\mathrm{N}(0, \sigma^2 \, \mathrm{Id})$, with $\sigma > 0$. Several such processes exist, but in practice we consider the Ornstein-Uhlenbeck (OU) process, also referred to as VP-SDE, which is defined by the following Stochastic Differential Equation (SDE)

$$\mathrm{d}\mathbf{X}_t = -\tfrac{1}{2}\mathbf{X}_t \mathrm{d}t + \sigma \mathrm{d}\mathbf{B}_t, \qquad \mathbf{X}_0 \sim p_0. \tag{2}$$

Under conditions on $p_0$, for any $T > 0$, $(\mathbf{Y}_t)_{t \in [0,T]} = (\mathbf{X}_{T-t})_{t \in [0,T]}$ is also the (weak) solution to a SDE (Anderson, 1982; Haussmann et al., 1986; Cattiaux et al., 2021)

$$\mathrm{d}\mathbf{Y}_t = \{\tfrac{1}{2}\mathbf{Y}_t + \sigma^2 \nabla \log p_{T-t}(\mathbf{Y}_t)\}\mathrm{d}t + \sigma \mathrm{d}\mathbf{B}_t, \ \mathbf{Y}_0 \sim p_T, \tag{3}$$

where $p_t$ denotes the density of $\mathbf{X}_t$. In practice, $\nabla \log p_t$ is approximated with a score network $(t, x) \mapsto s_\theta(t, x)$ trained by minimising either a denoising score matching (dsm) loss or an implicit score matching (ism) loss (Vincent, 2011)

$$\ell(\theta) = \mathbb{E}_{t \sim \mathcal{U}([0,T]), (\mathbf{X}_0, \mathbf{X}_t)}[\lambda_t(\tfrac{1}{2}\|s_\theta(t, \mathbf{X}_t)\|^2 + \mathrm{div}(s_\theta)(t, \mathbf{X}_t))], \tag{4}$$

where $\lambda_t > 0$. For a flexible score network, the global minimiser $\theta^\star = \arg\min_\theta \mathcal{L}(\theta)$ satisfies $s_{\theta^\star}(t, \cdot) = \nabla \log p_t$. De Bortoli et al. (2022) and Huang et al. (2022) have extended denoising diffusion models to the Riemannian setting. The time-reversal formula (3) remains the same, replacing the Euclidean gradient with its Riemannian equivalent. The ism loss can still be computed in that setting. However, the samplers used in the Riemannian setting differ from the classical Euler-Maruyama discretisation used in the Euclidean framework. De Bortoli et al. (2022) use Geodesic Random Walks (Jørgensen, 1975), which ensure that the samples remain on the manifold at every step. In this paper, we propose a sampler with similar properties in the case of *constrained* manifolds.

**Reflected SDE.** We conclude this section by recalling the framework for studying reflected SDEs, which is introduced via the notion of the *Skorokhod problem*. For simplicity, we focus on Euclidean space $\mathbb{R}^d$ here, but note that reflected processes can be defined on arbitrary smooth manifolds $\mathcal{N}$. In the case of the Brownian motion, a solution to the Skorokhod problem is a process of the form $(\bar{\mathbf{B}}_t, \mathbf{k}_t)_{t \geq 0}$. Locally, $(\bar{\mathbf{B}}_t)_{t \geq 0}$ can be seen as a regular Brownian motion $(\mathbf{B}_t)_{t \geq 0}$ while $(\mathbf{k}_t)_{t \geq 0}$ forces $(\bar{\mathbf{B}}_t)_{t \geq 0}$ to remain in $\mathcal{M}$. Under mild additional regularity conditions on $\mathcal{M}$ and $(\bar{\mathbf{B}}_t, \mathbf{k}_t)_{t \geq 0}$, see Skorokhod (1961), $(\bar{\mathbf{B}}_t, \mathbf{k}_t)_{t \geq 0}$ is a solution to the *Skorokhod problem* if for any $t \geq 0$

$$\bar{\mathbf{B}}_t = \bar{\mathbf{B}}_0 + \mathbf{B}_t - \mathbf{k}_t \in \mathcal{M}, \tag{5}$$

$|\mathbf{k}|_t = \int_0^t \mathbf{1}_{\bar{\mathbf{B}}_s \in \partial\mathcal{M}}\mathrm{d}|\mathbf{k}|_s$ and $\mathbf{k}_t = \int_0^t \mathbf{n}(\bar{\mathbf{B}}_s)\mathrm{d}|\mathbf{k}|_s$, where $(|\mathbf{k}|_t)_{t \geq 0}$ is the total variation of $(\mathbf{k}_t)_{t \geq 0}$[1]. Let us provide some intuition on this definition. When $(\bar{\mathbf{B}}_t)_{t \geq 0}$ hits the boundary $\partial\mathcal{M}$, $-\mathbf{k}_t$ pushes the process back into $\mathcal{M}$ along the inward normal $-\mathbf{n}(\bar{\mathbf{B}}_t)$, according to $\mathbf{k}_t = \int_0^t \mathbf{n}(\bar{\mathbf{B}}_s)\mathrm{d}|\mathbf{k}|_s$.

---

[1] In this case $(\mathbf{k}_t)_{t \geq 0}$ is not regular enough, but if it were in the class $\mathrm{C}^1$, its total variation would be given by $\int_0^t |\partial_t \mathbf{k}_t|\mathrm{d}s$ in the one-dimensional case.

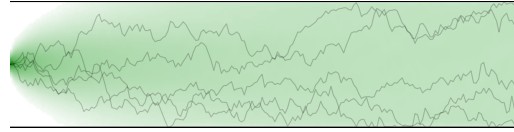
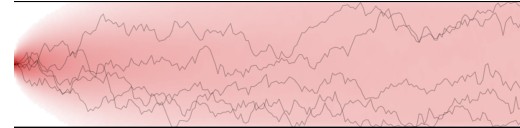

(a) Reflected Brownian motion          (b) Metropolised approximation of Brownian motion

Figure 3: Evolution of the density of the reflected Brownian motion and its Metropolis sampling-based approximation on the unit interval starting from a delta mass.

The condition $|\mathbf{k}|_t = \int_0^t \mathbf{1}_{\bar{\mathbf{B}}_s \in \partial \mathcal{M}} \mathrm{d}|\mathbf{k}|_s$ is more technical and can be seen as imposing that $\mathbf{k}_t$ remains constant so long as $(\bar{\mathbf{B}}_t)_{t \geq 0}$ does not hit $\partial \mathcal{M}$. We refer to Fishman et al. (2023) and Lou et al. (2023) for a more thorough introduction of these concepts in the context of diffusion models.

# 3 Diffusion models for constrained manifolds via Metropolis sampling

In Section 3.1, we highlight the practical limitations of existing constrained diffusion models and propose an alternative Metropolis sampling-based approach. In Section 3.2, we outline our proof that this process corresponds to a valid discretisation of the reflected Brownian motion, justifying its use in diffusion models. An overview of the samplers we cover in this section is presented in Table 1.

## 3.1 Practical limitations of existing constrained diffusion models

**Barrier methods.** In the barrier approach, a constrained manifold is transformed into an unconstrained space via a barrier metric. This metric is defined by $\nabla^2 \phi(x)$ with $\phi(x) = \sum_{i \in \mathcal{I}} \phi_i(d(x, f_i))$ where $d(x, f_i)$ is the minimum distance from the point $x$ to the set defined by $f_i(x) = 0$ and $\phi_i$ is a monotonically decreasing function such that $\lim_{z \to 0} \phi_i(z) = \infty$ . Under additional regularity assumptions, $\phi$ is called a *barrier function* (see Nesterov et al. (1994)). This definition ensures that the barrier function induces a well-defined exponential map on the manifold, making the Riemannian diffusion model frameworks of De Bortoli et al. (2022) and Huang et al. (2022) applicable. In the log-barrier method of Fishman et al. (2023), evaluating $\phi$ requires computing $d(x, \partial \mathcal{M})$ (and its derivatives), which can be prohibitively expensive. Furthermore, since the exponential map under the induced manifold is difficult to compute, it is approximated by projecting the exponential map on the original manifold back onto the constraint set, incurring an additional bias. Liu et al. (2023) propose a more tractable method by constructing a mirror map that transforms a constrained domain into an unconstrained dual space, in which one can train a standard Euclidean diffusion model. However, this approach is only applicable to convex subsets of $\mathbb{R}^d$ and does not extend to arbitrary Riemannian manifolds. More generally, warping the geometry of the modelled domain can adversely impact the interpolative performance of log-barrier-based diffusion models, as the space between data points expands rapidly when approaching the boundary.

**Reflected stochastic processes.** Fishman et al. (2023) and Lou et al. (2023) introduce diffusion models based on the *reflected Brownian motion* (RBM). In Fishman et al. (2023), the reflected SDE is discretised by (i) considering a classical step of the Euler-Maruyama discretization (or the Geodesic Random Walk in the Riemannian setting) and (ii) reflecting this step according to the boundary defined by $\partial \mathcal{M}$. To compute the reflection, one must check whether the step crosses the boundary. If it does, the point of intersection needs to be calculated in order to reflect the ray and continue the step in the reflected direction. This can require an arbitrarily large number of reflections depending on the step size, the geodesic on the manifold, and the geometry of the bounded region within the manifold. We refer to Appendix C for the pseudocode of the reflection step and additional comments. An alternative approach to discretising a reflected SDE is to replace the reflection with a projection (Słomiński, 1994). However, the projection requires the most expensive part of the reflection algorithm: computing the intersection of the geodesic with the boundary. Lou et al. (2023) propose a more tractable approach that exploits the product structure of the unit hypercube to afford simulation-free sampling but does not extend to arbitrary Riemannian manifolds. Additionally, specifying convex constraints in their framework requires a bijection onto the hypercube, distorting the modelled geometry and incurring the same issues as outlined above.

Table 1: Comparison of the advantages and disadvantages of the different constrained (Riemannian) diffusion models covered in Section 3.1.

| | | Both required for fast DSM loss | | |
| | | | | |
| **DIFFUSION MODEL** | TRACTABLE CONDITIONAL SCORE | SIMULATION-FREE FORWARD SAMPLING | MODELLING DOMAIN | PRESERVES METRIC OF $\mathcal{M}$ |
|---|---|---|---|---|
| Reflected Diffusions | | | | |
| LOU ET AL. (2023) | ✓ | ✓ | convex $\subset \mathbb{R}^d$ | ✗ |
| FISHMAN ET AL. (2023) | ✗ | $\mathcal{O}(d^2)$ | Any $\mathcal{M}$ | ✓ |
| METROPOLIS (OURS) | ✗ | $\mathcal{O}(d)$ | Any $\mathcal{M}$ | ✓ |
| Barrier Diffusions | | | | |
| FISHMAN ET AL. (2023) | ✗ | ✗ | convex $\subset$ any $\mathcal{M}$ | ✗ |
| LIU ET AL. (2023) | ✓ | ✓ | convex $\subset \mathbb{R}^d$ | ✗ |

**Metropolis approximation.** Existing approaches to constrained (Riemannian) diffusion models either only apply to convex subsets of $\mathbb{R}^d$ or require manifold- and constraint-specific implementations that become computationally intractable as the complexity of the modelled geometry increases. This limits their practicality even for relatively simple manifolds with well-defined exponential maps and linear inequality constraints such as for example polytopes.

In the following, we introduce a method that aims to solve both of these problems. The sampler we propose is similar to a classical Euler-Maruyama discretisation of the Brownian motion, except that, whenever a step would carry the Brownian motion out of the constrained region, we reject it (see Algorithm 1). This is a *Metropolised* version of the usual discretisation and is trivial to implement compared to the existing barrier, reflection and projection methods. Hence, this method enables the principled extension of diffusion models to arbitrarily constrained manifolds at virtually *no added implementational complexity or computational expense*.

---

**Algorithm 1** *Metropolis approx. of RBM*

**Require:** $p \in \mathcal{M}$, $\{f_i\}_{i \in \mathcal{I}}$
  Sample $\boldsymbol{v} \sim \mathrm{N}(0, \mathrm{Id}) \in \mathrm{T}_p\mathcal{M}$
  $p' \leftarrow \exp_p(\boldsymbol{v})$
  **if** $f_i(p') < 0 \; \forall \, i$ **then**
    $p \leftarrow p'$
  **end if**
  **return** $p$

---

### 3.2 Relating the Metropolis sampler to the reflected Brownian motion

In this section, we prove that the proposed Metropolis sampling-based process (Algorithm 1) corresponds to a valid discretisation of the reflected process, justifying its use in diffusion models. Here we focus on a concise presentation of the core concepts and the main results. A full proof can be found in Appendix D. For simplicity, we restrict ourselves to the Euclidean setting. All of our results require particular assumptions on $\mathcal{M}$, which we discuss at the end of this section. We begin with a definition of the Metropolis approximation of RBM.

**Definition 1.** *For any $\gamma > 0$ and $k \in \mathbb{N}$, let $X_0^\gamma \in \mathcal{M}$ and $X_{k+1}^\gamma = X_k^\gamma + \sqrt{\gamma}Z_k^\gamma$ if $X_k^\gamma + \sqrt{\gamma}Z_k^\gamma \in \mathcal{M}$ and $X_k^\gamma$ otherwise. The sequence $(X_k^\gamma)_{k \in \mathbb{N}}$ is called the Metropolis approximation of RBM.*

For any $\gamma > 0$, we consider $(\mathbf{X}_t^\gamma)_{t \geq 0}$, the linear interpolation of $(X_k^\gamma)_{k \in \mathbb{N}}$ such that for any $k \in \mathbb{N}$, $\mathbf{X}_{k\gamma}^\gamma = X_k^\gamma$. The following result is the main theoretical contribution of our paper.

**Theorem 2.** *Under assumptions on $\mathcal{M}$, for any $T \geq 0$, $(\mathbf{X}_t^\gamma)_{t \in [0,T]}$ weakly converges to the RBM $(\bar{\mathbf{B}}_t)_{t \in [0,T]}$ as $\gamma \to 0$.*

The rest of the section is devoted to a high level presentation of the proof of Theorem 2. It is theoretically impractical to work directly with the Metropolis approximation of RBM. Instead, we introduce an auxiliary process, show this converges to the RBM, and finally prove that the convergence of the auxiliary process implies the convergence of our Metropolis discretisation.

**Definition 3.** *For any $\gamma > 0$ and $k \in \mathbb{N}$, let $\hat{X}_0^\gamma = x \in \mathcal{M}$ and $\hat{X}_{k+1}^\gamma = \hat{X}_k^\gamma + \sqrt{\gamma}Z_k^\gamma$ with $Z_k^\gamma$ a Gaussian random variable conditioned on $\hat{X}_k^\gamma + \sqrt{\gamma}Z_k^\gamma \in \mathcal{M}$. The sequence $(\hat{X}_k^\gamma)_{k \in \mathbb{N}}$ is called the Rejection approximation of RBM.*

We call this process *Rejection approximation of RBM* since in practice, $Z_k^\gamma$ is sampled using rejection sampling, see Algorithm 2. For any $\gamma > 0$, we also consider $(\hat{\mathbf{X}}_t^\gamma)_{t\geq 0}$, the linear interpolation of $(\hat{X}_k^\gamma)_{k\in\mathbb{N}}$ such that for any $k \in \mathbb{N}$, $\hat{\mathbf{X}}_{k\gamma}^\gamma = \hat{X}_k^\gamma$. In Appendix D, we prove the following result.

**Theorem 4.** *Under assumptions on $\mathcal{M}$, for any $T \geq 0$, $(\hat{\mathbf{X}}_t^\gamma)_{t\in[0,T]}$ weakly converges to the Reflected Brownian Motion $(\bar{\mathbf{B}}_t)_{t\in[0,T]}$ as $\gamma \to 0$.*

---

**Algorithm 2** *Rejection approx. of RBM*

---

**Require:** $p \in \mathcal{M}$, $\{f_i\}_{i\in\mathcal{I}}$
    Sample $\boldsymbol{v} \sim \mathrm{N}(0, \mathrm{Id}) \in \mathrm{T}_p\mathcal{M}$
    $p' \leftarrow \exp_p(\boldsymbol{v})$
    **while** $f_i(p') \geq 0$ for any $i$ **do**
        Sample $\boldsymbol{v} \sim \mathrm{Id}(0, 1) \in \mathrm{T}_p\mathcal{M}$
        $p' \leftarrow \exp_p(\boldsymbol{v})$
    **end while**
    **return** $p'$

---

*Proof.* Here we give some elements of the proof. Details and full derivations are postponed to Appendix D. Our approach is based on the invariance principle of Stroock et al. (1971). More precisely, we show that we can compute an equivalent 'drift' and 'diffusion matrix' for the discretised process and that, as $\gamma \to 0$, the drift converges to zero and the diffusion matrix converges to $\mathrm{Id}$. In the Euclidean setting, this result, accompanied by mild regularity and growth assumptions, ensures that the discretization weakly converges to the original SDE. However, the case with boundary is much more complicated, primarily because the approximate drift might explode near the boundary, thus we need to verify exactly how the drift behaves as $\gamma \to 0$ and as the process approaches the boundary. We show that the *normalised* drift converges to the inward normal near the boundary. This ensures that (a) in the interior of $\mathcal{M}$ the drift converges to zero, i.e. locally in the interior of $\mathcal{M}$ the Brownian motion and the Reflected Brownian Motion coincide, (b) on the boundary, the drift pushes the samples inside the manifold according to the inward normal, mimicking $(\mathbf{k}_t)_{t\geq 0}$ in (5). Finally, with results from Stroock et al. (1971) and Kang et al. (2017), we show the convergence to the RBM. $\qquad\square$

Our next step is to show that the approximate drift and diffusion matrix of the Metropolised process are upper and lower bounded by their counterparts in the rejection process. While the upper-bound is easy to derive, the lower-bound requires the following result.

**Proposition 5.** *Under assumptions on $\mathcal{M}$, $\forall\, \varepsilon > 0$, $\exists\, \bar{\gamma} > 0$ such that for any $\gamma \in (0, \bar{\gamma})$ and for any $x \in \mathcal{M}$, $\gamma \in (0, \bar{\gamma})$ and $Z \sim \mathrm{N}(0, \mathrm{Id})$ we have $\mathbb{P}(x + \sqrt{\gamma}Z \in \mathcal{M}) \geq 1/2 - \varepsilon$, with $Z \sim \mathrm{N}(0, \mathrm{Id})$.*

Proposition 5 tells us that *locally* the boundary looks like a half-space when integrating w.r.t. a Gaussian measure. A corollary is that, for $\gamma > 0$ small enough and for any $k \in \mathbb{N}$, the probability that $X_{k+1}^\gamma = X_k^\gamma$ is upper bounded *uniformly* w.r.t. $X_k^\gamma \in \mathcal{M}$. The proof of Proposition 5 uses Theorem 7 in Appendix D, whose proof relies on the concept of *tubular neighborhoods* (Lee et al., 2012).

Having established the lower and upper bound, we can conclude the proof by noting that the approximate drift and the diffusion matrix in the rejection and Metropolis case coincide as $\gamma \to 0$. This is enough to apply the same results as before, giving the desired convergence.

**Assumptions on $\mathcal{M}$.** Before concluding this section, we detail the assumptions we make on $\mathcal{M}$. For Theorem 2 to hold, we assume that $\mathcal{M} = \{x \in \mathbb{R}^d : \Phi(x) > 0\}$ is bounded, with $\Phi \in \mathrm{C}^2(\mathbb{R}^d, \mathbb{R})$ concave. We have that $\partial\mathcal{M} = \{x \in \mathbb{R}^d : \Phi(x) = 0\}$. In addition, we assume that for any $x \in \partial\mathcal{M}$, $\|\nabla\Phi(x)\| = 1$. These assumptions match those Stroock et al. (1971) use for their study of the existence of solutions to the RBM. While it seems possible to relax the *global* existence of $\Phi$ to a *local* one, the regularity assumption of the domain is key. This regularity is essential to establish Proposition 5 and the associated geometrical result on tubular neighbourhoods. We also emphasize that the smoothness of the domain is central in the results of Kang et al. (2017) on the equivalence of two definitions of RBMs which we rely on.

## 4 Related work on approximations of reflected SDEs

Several schemes have been introduced to approximately sample from reflected Stochastic Differential Equations. They can be interpreted as modifications of classical Euler-Maruyama schemes used to discretise SDEs without boundary. One of the most common approaches is to use the Euler-Maruyama discretisation and project the solution onto the boundary if it escapes from the domain $\mathcal{M}$.

Table 2: Log-likelihood (↑) of a held-out test set from a synthetic bimodal distribution over convex subsets of $\mathbb{R}^d$ bounded by the hypercube $[-1, 1]^d$ and unit simplex $\Delta^d$. Means and standard deviations are computed over 3 different runs. Average training time is provided in hours.

| MANIFOLD | DIMENSION | REFLECTED | | METROPOLIS | |
| --- | --- | --- | --- | --- | --- |
| | | log-likelihood | runtime | log-likelihood | runtime |
| | 2 | $2.25{\scriptstyle\pm.01}$ | 8.95 | $\mathbf{2.32}{\scriptstyle\pm.05}$ | $\mathbf{0.72}$ |
| $[-1, 1]^d$ | 3 | $3.77{\scriptstyle\pm.13}$ | 8.97 | $\mathbf{4.15}{\scriptstyle\pm.15}$ | $\mathbf{0.71}$ |
| | 10 | $7.42{\scriptstyle\pm.77}$ | 10.1 | $\mathbf{10.80}{\scriptstyle\pm.34}$ | $\mathbf{0.90}$ |
| | 2 | $1.01{\scriptstyle\pm.01}$ | 9.17 | $\mathbf{1.06}{\scriptstyle\pm.02}$ | $\mathbf{0.82}$ |
| $\Delta^d$ | 3 | $2.64{\scriptstyle\pm.01}$ | 9.43 | $\mathbf{3.23}{\scriptstyle\pm.17}$ | $\mathbf{0.78}$ |
| | 10 | $7.00{\scriptstyle\pm.13}$ | 10.5 | $\mathbf{7.81}{\scriptstyle\pm.20}$ | $\mathbf{0.97}$ |

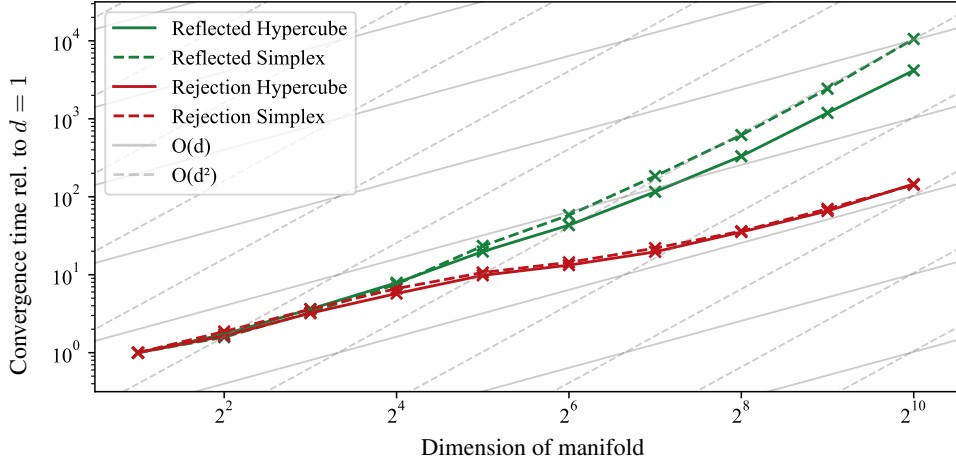

Figure 4: Convergence time of the Reflected (green) and Metropolis (red) forward noising processes to the uniform distribution on the hypercube $[-1, 1]^d$ and unit simplex $\Delta^d$. The lines indicate functions fit with the PYSR symbolic regression package (Cranmer, 2023) and correspond to empirical runtime complexities of $\mathcal{O}(d^2)$ and $\mathcal{O}(d)$, respectively, matching the superimposed scaling law isocontours.

In this case, mean-square error rates of order *almost* $1/2$ have been proven under various conditions (Liu, 1995; Chitashvili et al., 1981; Pettersson, 1995; Słomiński, 1994). Concretely this means that $\mathbb{E}[\|\bar{\mathbf{B}}_t - X_n^{t/n}\|^2] = O(n^{-1+\varepsilon})$ with $\varepsilon > 0$ arbitrary small and where $(X_k^\gamma)_{k\in\mathbb{N}}$ is the projection scheme. The rate $1/2$ is tight (Pacchiarotti et al., 1998). It is possible to use the Euler-Peano method to get slight improvements for a mean-square error rate of order of $1/2$, but this is impractical as it assumes that one can solve a (simplified) Skorokhod problem, which is usually intractable. Liu (1993) introduced a *penalised* method which pushes the solution away from the boundary and shows a mean-square error of order $1/4$, see also Pettersson (1997). Weak errors of order 1 have been obtained in Bossy et al. (2004) and Gobet (2001) by introducing a reflection component in the discretisation or using some local approximation of the domain to a half-space. We refer to Pilipenko (2014) for an introduction to the discretisation of reflected SDEs. Closer to our work, Burdzy et al. (2008) consider three different methods to approximate reflected Brownian motions on general domains (two based on discrete methods and one based on killed diffusions). Only qualitative results are provided. To the best of our knowledge, no previous work in the probability literature has investigated the *Metropolised* scheme we propose. Our Metropolis scheme is also related to the ball walk (Applegate et al., 1991), which replaces the Gaussian random variable with a uniform over the ball (or the Dikin ellipsoid). Applegate et al. (1991) and Lovász et al. (2007) have studied the asymptotic convergence rate of the ball walk, but, to the best of our knowledge, its limiting behaviour when the step size goes to zero has not been investigated.

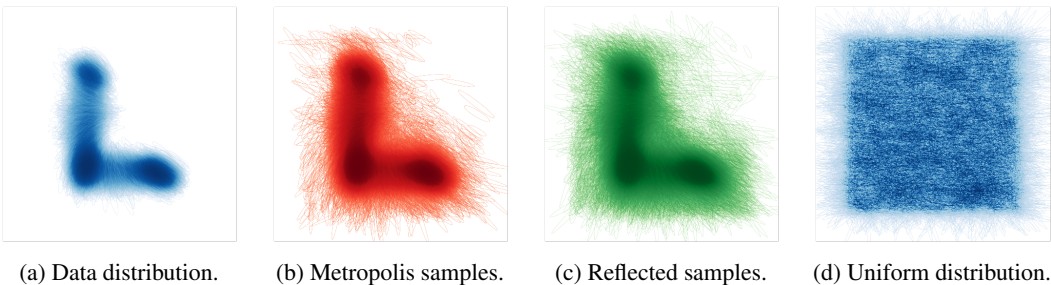

| (a) Data distribution. | (b) Metropolis samples. | (c) Reflected samples. | (d) Uniform distribution. |

Figure 5: A qualitative visual comparison of $10^6$ samples from the data distribution, our Metropolis diffusion model, a reflected diffusion model and the uniform distribution for the constrained configurational modelling of robotic arms on $\mathcal{S}_{++}^2 \times \mathbb{R}^2$.

# 5 Experimental results

To demonstrate the practical utility and empirical performance of the proposed Metropolis diffusion models, we conduct a comprehensive evaluation on a range of synthetic and real-world tasks. In Section 5.1, we assess the scalability of our method by applying it to synthetic distributions on hypercubes and simplices of increasing dimensionality. In Section 5.2, we extend the evaluation to real-world tasks on manifolds with convex constraints by applying our method to the robotics and protein design datasets presented in Fishman et al. (2023). In Section 5.3, we additionally demonstrate that our method extends to constrained manifolds with highly *non-convex* boundaries—a setting that is intractable with existing approaches.

As we found—in line with Fishman et al. (2023)—that log-barrier diffusion models perform strictly worse than reflected approaches across all experimental settings, we focus on a more detailed comparison with the latter here and postpone additional empirical results to Appendix F.1. These include additional performance metrics and a comparison to an unconstrained Euclidean diffusion model on the synthetic datasets presented in Section 5.1.

For all experiments, we use a simple 6-layer MLP with sine activations and a score rescaling function to ensure that the score reaches zero at the boundary, scaling linearly into the interior of the constrained set as in Liu et al. (2022) and Fishman et al. (2023). We set $T = 1$, $\beta_0 = 1 \times 10^{-3}$ and tune $\beta_1$ to ensure that the forward process reaches the invariant distribution with a linear $\beta$-schedule. We use a learning rate of $2 \times 10^{-4}$ with a cosine learning rate schedule and an ism loss with a modified loss weighting function of $(1 + t)$, a batch size of 1024 and 8 repeats per batch. All models were trained on a single NVIDIA GeForce GTX 1080 GPU. Additional details are provided in Appendix F.2.

All source code that is needed to reproduce the results presented below is made available under https://github.com/oxcsml/score-sde/tree/metropolis, which requires a supporting package to handle the different geometries that is available under https://github.com/oxcsml/geomstats/tree/polytope.

## 5.1 Synthetic distributions on simple polytopes

In this section, we investigate the scalability of the proposed Metropolis diffusion models by applying them to synthetic bimodal distributions over the $d$-dimensional hypercube $[-1, 1]^d$ and unit simplex $\Delta^d$. A quantitative comparison of the log-likelihood of a held-out test set is presented in Table 2, while a visual comparison is postponed to Appendix F.3. We find that our Metropolis models outperform reflected approaches across all dimensions and constraint geometries by a substantial and statistically significant margin while training in one tenth of the time. The degree of improvement seems to scale with the dimensionality of the problem: the larger the dimension of the experiment, the larger the gain in performance from using our proposed Metropolis scheme.

We observe a similar difference in the scaling properties of reflected and Metropolis models when measuring the convergence times of the respective forward noising processes to the uniform distribution on hypercubes $[-1, 1]^d$ and simplices $\Delta^d$ of increasing dimensionality. The results are presented in Section 4 and show that the convergence time of the Metropolis process scales linearly in the dimension, while the reflected process scales quadratically.

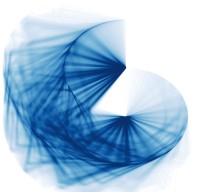 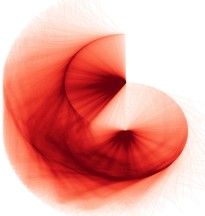 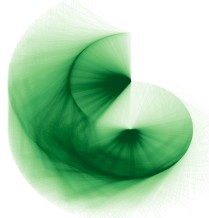 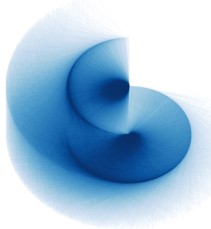

(a) Data distribution. (b) Metropolis samples. (c) Reflected samples. (d) Uniform distribution.

Figure 6: A qualitative comparison of $10^5$ samples from the data distribution, our Metropolis diffusion model, a reflected diffusion model and the uniform distribution for the constrained conformational modelling of cyclic peptide backbones. For visual clarity, the figures only show the constrained planar projections encoded by $\mathbb{P} \subset \mathbb{R}^3$.

## 5.2 Modelling proteins and robotic arms under convex constraints

In addition to illustrating our method's scalability on high-dimensional synthetic tasks, we follow the experimental setup from Fishman et al. (2023) to additionally demonstrate its practical utility and favourable empirical performance on two real-world problems from robotics and protein design.

**Constrained configurational modelling of robotic arms.** The problem of modelling the configurations and trajectories of a robotic arm can be formulated as learning a distribution over the locations and manipulability ellipsoids of its joints, parameterised on $\mathbb{R}^d \times \mathcal{S}_{++}^d$, where $\mathcal{S}_{++}^d$ is the manifold of symmetric positive-definite (SPD) $d \times d$ matrices (Yoshikawa, 1985; Jaquier et al., 2021). For practical robotics applications, it may be desirable to restrict the maximal velocity with which a robotic arm can move or the maximum force it can exert. This manifests in a trace constraint $C > 0$ on $\mathcal{S}_{++}^d$, resulting in a constrained manifold $\{M \in \mathcal{S}_{++}^d : \sum_{i=1}^d M_{ii} < C\}$. Following Fishman et al. (2023), we parametrise this constraint via the Cholesky decomposition (Lin, 2019) and use the resulting setup to model the dataset presented in Jaquier et al. (2021).

**Conformational modelling of protein backbones.** Modelling the conformational ensembles of proteins is a data-scarce problem with a range of important applications in biotechnology and drug discovery (Lane, 2023). In many practical settings, it may often be unnecessary to model the structural ensembles of an entire protein, as researchers are primarily interested in specific functional sites that are embedded in a structurally conserved scaffold (Huang et al., 2016). Modelling the conformational ensembles of such substructural elements requires positional constraints on their endpoints to ensure that they can be accommodated by the remaining protein. Using the parametrisation and dataset presented in Fishman et al. (2023), we formulate the problem of modelling the backbone conformations of a cyclic peptide of length $N = 6$ as learning a distribution over the product of a polytope $\mathbb{P} \subset \mathbb{R}^3$ and the hypertorus $\mathbb{T}^4$.

Table 3: Log-likelihood ($\uparrow$) of a held-out test set for the robotics and protein applications. Means and standard deviations are computed over 3 different runs. Average training time is provided in hours.

| DATASET | DOMAIN | REFLECTED | | METROPOLIS | |
| --- | --- | --- | --- | --- | --- |
| | | log-likelihood | runtime | log-likelihood | runtime |
| Robotics | $\mathcal{S}_{++}^2 \times \mathbb{R}^2$ | $8.39_{\pm.06}$ | 9.52 | $\mathbf{9.13_{\pm.03}}$ | $\mathbf{1.36}$ |
| Proteins | $\mathbb{P} \subset \mathbb{R}^3 \times \mathbb{T}^4$ | $15.20_{\pm.06}$ | 24.80 | $\mathbf{15.33_{\pm.02}}$ | $\mathbf{3.12}$ |

We quantify the empirical performance of different methods by evaluating the log-likelihood of a held-out test set and present the resulting performance metrics in Table 3. Again, we find that our Metropolis model outperforms the reflected approach by a statistically significant margin while training 7-8 times as fast. Qualitative visual comparisons of samples from the true distribution, the trained diffusion models and the uniform distribution are presented in Figures 5 and 6, with full univariate marginal and pairwise bivariate correlation plots postponed to Appendices F.4 and F.5.

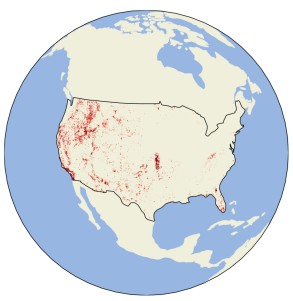 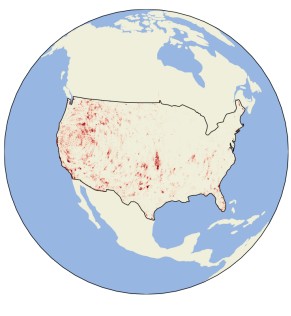 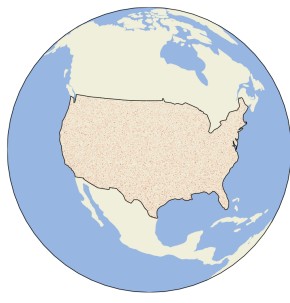

(a) Data distribution.         (b) Metropolis samples.         (c) Uniform distribution.

Figure 7: Orthographic projection of $10^5$ samples from (a) the data distribution, (b) our Metropolis diffusion model, and (c) the uniform distribution, for geospatial data (wildfire incidence rates) within a non-convex boundary (the continental United States). The projections are aligned with the geometric centre of the boundary and zoomed in ten-fold for visual clarity.

## 5.3   Modelling geospatial data within non-convex country borders

Motivated by the strong empirical performance of our approach on tasks with challenging convex constraints, we investigated its ability to model distributions whose support is restricted to manifolds with highly non-convex boundaries—a setting that is intractable with existing approaches. To this end, we derived a geospatial dataset based on wildfire incidence rates within the continental United States (see Appendix E for full details) and trained a Metropolis diffusion model constrained by the corresponding country borders on the sphere $\mathcal{S}^2$. A qualitative visual comparison of samples from the true distribution, our model, and the uniform distribution is presented in Figures 7a to 7c and a quantitative comparison to a Riemannian diffusion model on $\mathcal{S}^2$ (De Bortoli et al., 2022) is given in Table 4. Both demonstrate that our approach is able to successfully capture challenging multimodal and sparse distributions on constrained manifolds with highly non-convex boundaries.

Table 4: MMD ($\downarrow$) of a held-out test set for the geospatial modelling dataset. Means and standard deviations are computed over 3 different runs. Average training time is provided in hours.

| MODEL | DOMAIN | MMD | RUNTIME | % IN BOUNDARY |
|---|---|---|---|---|
| Unconstrained | $\mathcal{S}^2$ | $0.1567_{\pm 0.013}$ | **0.81** | 63.3 |
| Metropolis | $\mathbb{P} \subset \mathcal{S}^2$ | $\mathbf{0.1388}_{\pm 0.015}$ | 3.86 | **100.0** |

## 6   Conclusion

Accurately modelling distributions on constrained Riemannian manifolds is a challenging problem with a range of impactful practical applications. In this work, we have proposed a mathematically principled and computationally tractable extension of existing diffusion model methodology to this setting. Based on a *Metropolisation* of random walks in Euclidean spaces and on Riemannian manifolds, we have shown that our approach corresponds to a valid discretisation of the reflected Brownian motion, justifying its use in diffusion models. To demonstrate the practical utility of our method, we have performed an extensive empirical evaluation, showing that it outperforms existing constrained diffusion models on a range of synthetic and real-world tasks defined on manifolds with convex boundaries, including applications from robotics and protein design. Leveraging the flexibility and simplicity of our method, we have also demonstrated that it extends beyond convex constraints and is able to successfully model distributions on manifolds with highly non-convex boundaries. While we found our method to perform well across the synthetic and real-world applications we considered, we expect it to perform poorly on certain constraint geometries. For instance, the current implementation relies on an isotropic noise distribution which could impede its performance on exceedingly narrow constraint geometries, even with correspondingly small step sizes. In this context, an important direction of future research would be to investigate whether we can instead sample from more suitable distributions, e.g. a Dikin ellipsoid, while maintaining the simplicity and efficiency of the Metropolis approach.

## Acknowledgements

NF thanks the Rhodes Trust for supporting their studies at Oxford and this work. LK acknowledges support from the University of Oxford's Clarendon Fund.

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

# Supplementary to:

# Metropolis Sampling for Constrained Diffusion Models

## A  Overview

In Appendix B, we recall some basic concepts of Riemannian geometry which are key to defining discretisations of the reflected Brownian motion. In Appendix C, we give some details on the reflection step in reflected discretizations. In Appendix D, we prove the convergence of the rejection and Metropolis discretizations to the true reflected Brownian Motion. The geospatial dataset with non-convex constraints based on wildfire incidence rates in the continental United States is presented Appendix E. All supplementary experimental details and empirical results are gathered in Appendix F.

## B  Manifold concepts

In the following, we aim to introduce key concepts that underpin diffusion models on Riemannian manifolds, with a particular focus on notions relevant to the reflected Brownian motion that we build on in Appendix C. For a more thorough treatment with reference to reflected diffusion models, we refer to (Fishman et al., 2023). For a detailed presentation of smooth manifolds, see Lee (2013).

A Riemannian manifold is a tuple $(\mathcal{M}, \mathfrak{g})$ with $\mathcal{M}$ a smooth manifold and $\mathfrak{g}$ a metric that imbues the manifold with a notion of distance and curvature and is defined as a smooth positive-definite inner product on each of the tangent spaces of the manifold:

$$\mathfrak{g}(p) : \mathrm{T}_p\mathcal{M} \times \mathrm{T}_p\mathcal{M} \to \mathbb{R}.$$

The tangent space $\mathrm{T}_p$ of a point $p$ on a manifold is an extension of the notion of tangent planes and can be thought of as the space of derivatives of scalar functions on the manifold at that point.

To establish how different tangent spaces relate to one another, we need to additionally introduce the concept of a *connection*. This is a map that takes two vector fields and produces a derivative of the first with respect to the second, typically written as $\nabla(X, Y) = \nabla_X Y$. While there are infinitely many connections on any given manifold, the *Levi-Cevita* emerges as a natural choice if we impose the following two conditions:

(i)  $X \cdot (\mathfrak{g}(Y, Z)) = \mathfrak{g}(\nabla_X Y, Z) + \mathfrak{g}(Y, \nabla_X Z),$

(ii)  $[X, Y] = \nabla_X Y - \nabla_Y X,$

where $[\cdot, \cdot]$ is the Lie bracket. These conditions ensure that the connection is (i) metric-preserving and (ii) torsion-free, with the latter guaranteeing a unique connection and integrability on the manifold.

Using the metric and Levi-Cevita connection, we can define a number of key concepts:

**Geodesic.**  *Geodesics* extend the Euclidean notion of 'straight lines' to manifolds. They are defined as the unique path $\gamma : (0, 1) \to \mathcal{M}$ such that $\nabla_{\gamma'}\gamma' = 0$ and are the shortest path between two points on a manifold, in the sense that $L(\gamma) = \int_0^1 \sqrt{\mathfrak{g}(\gamma(t))(\gamma'(t), \gamma'(t))}\mathrm{d}t$ is minimal.

**Exponential map.**  The *exponential map* on a manifold is given by the mapping between an element $\boldsymbol{v} \in T_p\mathcal{M}$ of the tangent space at point $p$ and the endpoint of the unique geodesic $\gamma$ with $\gamma(0) = p$ and $\gamma'(0) = \boldsymbol{v}$.

**Intersection.**  The *intersection* along a geodesic is the first point at which the geodesic intersects the boundary. We recall that the boundary is defined by $f = 0$. We can define this via an optimisation problem: compute the minimum $t > 0$ such that we have that $\exp_{\mathfrak{g}}(x, tz)$ is a root of $f$: $f(\exp_{\mathfrak{g}}(x, tz)) = 0$. We will say that $\exp_{\mathfrak{g}}(x, tz) = \mathrm{intersect}_{\mathfrak{g}}(x, z; f)$ and that $t = \arg\mathrm{intersect}_{\mathfrak{g}}(x, z; f)$.

**Parallel transport.** We say that a vector field $X$ is *parallel* to a curve $\gamma : (0, 1) \to \mathcal{M}$ if $\nabla_{\gamma'} X = 0$, where $\gamma' : (0, 1) \to \mathrm{T}_{\gamma(t)}\mathcal{M}$. For two points on the manifold $p, q \in \mathcal{M}$ that are connected by a curve $\gamma$, and an initial vector $X_0 \in \mathrm{T}_p\mathcal{M}$, there is a unique vector field $X$ that is parallel to $\gamma$ such that $X(p) = X_0$. This induces a map between the tangent spaces at $p$ and $q$ $\tau_\gamma : \mathrm{T}_p\mathcal{M} \to \mathrm{T}_q\mathcal{M}$, which is referred to as the *parallel transport* of tangent vectors between $p$ and $q$ and satisfies the condition that for $\boldsymbol{v}, \boldsymbol{u} \in \mathrm{T}_p\mathcal{M}$ $\mathfrak{g}(p)(\boldsymbol{v}, \boldsymbol{u}) = \mathfrak{g}(q)(\tau_\gamma(\boldsymbol{v}), \tau_\gamma(\boldsymbol{u}))$.

**Reflection.** For an element $\boldsymbol{v} \in T_p\mathcal{M}$ in the tangent space of the manifold at point $p$ and a constraint characterised by its unit normal vector $\boldsymbol{n} \in T_p\mathcal{M}$, the *reflection* of $\boldsymbol{v}$ in the tangent space is given by $\boldsymbol{v}' = \boldsymbol{v} - 2\mathfrak{g}(\boldsymbol{v}, \boldsymbol{n})\boldsymbol{n}$.

## C Full Reflected Discretisation

Here, we reproduce the central algorithm for the full discretisation of the reflected Brownian motion (Algorithm 4) derived for Euclidean models in (Lou et al., 2023) and for Reimannian models in (Fishman et al., 2023). Its central component is the *Reflected Step Algorithm* (Algorithm 3), which gives a generic computation for the reflection in any manifold. Due to the need to balance speed and numerical instability issues around the boundary, an efficient practical implementation of the reflected step is highly non-trivial, even for simple polytopes in Euclidean space. More complex geometries and boundaries make this problem significantly worse: a constraint on the trace of SPD matrices under the log-Cholesky metric of (Lin, 2019) requires solving complex non-convex optimisation problems for each sample at each discretised sampling step in both the forward and reverse process. This motivates our work in this paper.

These problems motivated the development of our Metropolis approximation, which significantly simplifies the random walk. Instead of requiring the intersection, parallel transport and reflection, we simply need to be able to evaluate the constraint functions $f_i$. We highlight this simplicity in Algorithm 5.

---

**Algorithm 3** *Reflected Step Algorithm*. The algorithm operates by repeatedly taking geodesic steps until one of the constraints is violated, or until the step is fully taken. Upon hitting the boundary, we parallel-transport the tangent vector to the boundary and then reflect it against it. We then start a new geodesic from this point in the new direction. The $\arg \mathrm{intersect}_t$ function computes the distance one must travel along a geodesic in direction $\boldsymbol{s}$ until constraint $f_i$ is violated. For a discussion of paralleltransport, $\exp_{\mathfrak{g}}$ and reflect see appendix B.

---

**Require:** $x \in \mathcal{M}, \boldsymbol{v} \in \mathrm{T}_x\mathcal{M}, \{f_i\}_{i \in \mathcal{I}}$
    $\ell \leftarrow \|\boldsymbol{v}\|_{\mathfrak{g}}$
    $\boldsymbol{s} \leftarrow \boldsymbol{v}/\|\boldsymbol{v}\|_{\mathfrak{g}}$
    **while** $\ell \geq 0$ **do**
        $d_i = \arg \mathrm{intersect}_{\mathfrak{g}}(x, z; f_i)$
        $i \leftarrow \arg \min_i d_i$
        $\alpha \leftarrow \min(d_i, \ell)$
        $x' \leftarrow \exp_{\mathfrak{g}}(x, \alpha\boldsymbol{s})$
        $\boldsymbol{s} \leftarrow \mathrm{paralleltransport}_{\mathfrak{g}}(x, \boldsymbol{s}, x')$
        $\boldsymbol{s} \leftarrow \mathrm{reflect}(\boldsymbol{s}, f_i)$
        $\ell \leftarrow \ell - \alpha$
        $x \leftarrow x'$
    **end while**
    **return** $x$

---

**Algorithm 4** *Reflected Random Walk.* Discretisation of the SDE $\mathrm{d}\mathbf{X}_t = b(t, \mathbf{X}_t)\mathrm{d}t + \mathrm{d}\mathbf{B}_t - \mathrm{d}\mathbf{k}_t$.

**Require:** $T, N, X_0^\gamma, \{f_i\}_{i \in \mathcal{I}}$
$\gamma = T/N$
**for** $k \in \{0, \dots, N-1\}$ **do**
    $Z_{k+1} \sim \mathrm{N}(0, \mathrm{Id})$
    $X_{k+1}^\gamma = \mathrm{ReflectedStep}[X_k^\gamma, \sqrt{\gamma}Z_{k+1}, \{f_i\}_{i \in \mathcal{I}}]$
**end for**
**return** $\{X_k^\gamma\}_{k=0}^N$

---

**Algorithm 5** *Metropolis Random Walk.* Discretisation of the SDE $\mathrm{d}\mathbf{X}_t = b(t, \mathbf{X}_t)\mathrm{d}t + \mathrm{d}\mathbf{B}_t - \mathrm{d}\mathbf{k}_t$.

**Require:** $T, N, X_0^\gamma, \{f_i\}_{i \in \mathcal{I}}$
$\gamma = T/N$
**for** $k \in \{0, \dots, N-1\}$ **do**
    $Z_{k+1} \sim \mathrm{N}(0, \mathrm{Id})$
    $X' \leftarrow \exp_{\mathfrak{g}}\left(X_k^\gamma, \sqrt{\gamma}Z_{k+1}\right)$
    **if** $\max_{i \in \mathcal{I}} f_i(X') \leq 0$ **then**
        $X_{k+1}^\gamma = X'$
    **else**
        $X_{k+1}^\gamma = X_k^\gamma$
    **end if**
**end for**
**return** $\{X_k^\gamma\}_{k=0}^N$

# D  Convergence to the reflected process

In this note, we assume that $\mathcal{M} = \{x \in \mathbb{R}^d : \Phi(x) > 0\}$ is compact, with $\Phi \in C^2(\mathbb{R}^d, \mathbb{R})$. We have that $\partial\mathcal{M} = \{x \in \mathbb{R}^d : \Phi(x) = 0\}$. In addition, we assume that for any $x \in \partial\mathcal{M}$, $\|\nabla\Phi(x)\| = 1$ and that $\Phi$ is concave. The closure of $\mathcal{M}$ is denoted $\overline{\mathcal{M}}$. The assumption that $\Phi$ is concave is only used in Theorem 7-(d) and can be dropped. We consider it for simplicity.

Let $(\hat{X}_k^\gamma)_{k \in \mathbb{N}}$ given for any $\gamma > 0$ and $k \in \mathbb{N}$ by $\hat{X}_0^\gamma = x \in \overline{\mathcal{M}}$ and for $\hat{X}_{k+1}^\gamma = \hat{X}_k^\gamma + \sqrt{\gamma}Z_k^\gamma$ with $Z_k^\gamma$ a Gaussian random variable conditioned on $\hat{X}_k^\gamma + \sqrt{\gamma}Z_k^\gamma \in \overline{\mathcal{M}}$. In practice, $Z_k^\gamma$ is sampled using rejection sampling. We define $\hat{\mathbf{X}}^\gamma : \mathbb{R}_+ \to \overline{\mathcal{M}}$ given for any $k \in \mathbb{N}$ by $\hat{\mathbf{X}}_{k\gamma}^\gamma = \hat{X}_k^\gamma$ and for any $t \in [k\gamma, (k+1)\gamma)$, $\hat{\mathbf{X}}_t^\gamma = \hat{X}_k^\gamma$. Note that $(\mathbf{X}_t)_{t \in [0,T]}$ is a $D([0,T], \overline{\mathcal{M}})$ valued random variable, where $D([0,T], \overline{\mathcal{M}})$ is the space of right-continuous with left-limit processes which take values in $\overline{\mathcal{M}}$. We denote $\hat{\mathbb{P}}^\gamma$ the distribution of $(\hat{\mathbf{X}}_t^\gamma)_{t \in [0,T]}$ on $D([0,T], \overline{\mathcal{M}})$.

Our goal is to show the following theorem.

**Theorem 6.** *For any $T \geq 0$, $(\hat{\mathbf{X}}_t^\gamma)_{t \in [0,T]}$ weakly converges to $(\mathbf{X}_t)_{t \in [0,T]}$ such that for any $t \in [0,T]$*

$$\mathbf{X}_t = x + \mathbf{B}_t - \mathbf{k}_t, \qquad |\mathbf{k}|_t = \int_0^t \mathbb{1}_{\mathbf{X}_s \in \partial\mathcal{M}} d|\mathbf{k}|_s, \qquad \mathbf{k}_t = \int_0^t \mathbf{n}(\mathbf{X}_s) d|\mathbf{k}|_s. \qquad (6)$$

*Proof.* In order to prove the result, we prove that the distribution of the Markov chain seen as an element of $D([0,T], \overline{\mathcal{M}})$ converges to a solution of the Skorokhod problem (6). In particular, we first show that the limiting distribution satisfies a submartingale problem following (Stroock et al., 1971, Theorem 6.3). The transition from a solution of a submartingale problem to a weak solution of the Skorokhod problem is given by (Kang et al., 2017, Theorem 1, Proposition 2.12) and (Ramanan, 2006, Corollary 2.10). In order to apply (Stroock et al., 1971, Theorem 6.3), we define an intermediate drift and diffusion matrix, see (55) and (51). To prove the theorem one needs to control the drift and diffusion matrix inside $\mathcal{M}$ and show that it converges to $0$ and Id respectively. The technical part of the proof comes from the control of the drift coefficient near the boundary. In particular, we show that if the intermediate drift is large then we are close to the boundary and the intermediate drift is pointing inward. To investigate the local properties of the drift near the boundary we rely on the notion of tubular neighborhood, see (Lee et al., 2012, Theorem 6.24). $\qquad\square$

Some key properties of the tubular neighborhood are stated in Appendix D.1. We then establish a few technical lemmas about the tail probability of some distributions in Appendix D.2. Controls on the diffusion matrix and lower bounds on the probability of belonging in $\mathcal{M}$ are given in Appendix D.3. Properties of large drift terms are given in Appendix D.4. The convergence of the drift and diffusion matrix on compact sets is given in Appendix D.5. The convergence of the boundary terms is investigated in Appendix D.6. Finally, we conclude the proof in Appendix D.7.

## D.1  Properties of the tubular neighborhood

Using the results of (Lee et al., 2012), we establish the existence of an open set of $\overline{\mathcal{M}}$ (for the induced topology of $\mathbb{R}^d$) satisfying several important properties.

**Theorem 7.** *There exist $\mathsf{U} \subset \overline{\mathcal{M}}$ open and $C \geq 1, \bar{r} > 0$ such that for any $\gamma \in (0, \bar{\gamma})$ with $\bar{\gamma} = 1$ the following properties hold:*

(a) *For any $x \in \mathsf{U}$, there exist a unique $\bar{x} \in \partial\mathcal{M}$ and $\bar{\alpha} > 0$ such that $x = \bar{x} + \bar{\alpha}\nabla\Phi(\bar{x})$.*

(b) *For any $\bar{\alpha} \in [0, \bar{r}]$ and $\bar{x} \in \partial\mathcal{M}$ such that $\bar{x} + \bar{\alpha}\nabla\Phi(\bar{x}) \in \overline{\mathcal{M}}$, let $x = \bar{x} + \bar{\alpha}\nabla\Phi(\bar{x})$ and $\mathsf{C}(x, \gamma)$ such that $x + \sqrt{\gamma}z \in \mathsf{C}(x, \gamma)$ if*

$$-\bar{\alpha}\gamma^{-1/2} \leq \alpha < \bar{r}\gamma^{-1/2}, \qquad \|v\|^2 \leq (\alpha\gamma^{1/2} + \bar{\alpha})/(C\gamma), \qquad (7)$$

*with $z = \alpha\nabla\Phi(\bar{x}) + v$, with $v \perp \nabla\Phi(\bar{x})$. Then $\mathsf{C}(x, \gamma) \subset \overline{\mathcal{M}}$.*

(c) *$\mathsf{V} = \{\bar{x} + \alpha\nabla\Phi(\bar{x}) : \bar{x} \in \partial\mathcal{M}, \alpha \in [0, \bar{r})\}$ is open in $\overline{\mathcal{M}}$.*

(d) *For any $x \in \mathsf{U}$, $x + \sqrt{\gamma}z \in \overline{\mathcal{M}} \cap \mathsf{C}(x, \gamma)^c$ then $\alpha \geq \bar{r}\gamma^{-1/2}$ or $\|v\|^2 \geq (\alpha\gamma^{1/2} + \bar{\alpha})/(C\gamma)$ and $\alpha\gamma^{1/2} + \bar{\alpha} \geq 0$, with $z = \alpha\nabla\Phi(\bar{x}) + v$, with $\bar{x}$ given in (a) and $v \perp \nabla\Phi(\bar{x})$. .*

*(e) There exists $R > 0$ such that $\{x \in \overline{\mathcal{M}} \ : \ d(x, \partial\mathcal{M}) \le 2R\} \subset \mathsf{V}$.*

*Proof.* Let $\gamma \in (0, \bar\gamma)$ with $\bar\gamma = 1$. First, note that for any $\bar x \in \partial\mathcal{M}$, the normal space is given by $\{\alpha\nabla\Phi(\bar x) \ : \ \alpha \in \mathbb{R}\}$. Using this result and (Lee et al., 2012, Theorem 6.24) there exists $\tilde r_0 > 0$ such that $\mathsf{U}_0 = \{\bar x + \alpha\nabla\Phi(\bar x) \ : \ \bar x \in \partial\mathcal{M}, \ \alpha \in (-\tilde r_0, \tilde r_0)\} \subset \mathbb{R}^d$ is open[2]. We have that for any $\alpha \in [-r_0, 0)$ and $\bar x \in \partial\mathcal{M}$

$$\Phi(\bar x + \alpha\nabla\Phi(\bar x)) = \Phi(\bar x) + \alpha\|\nabla\Phi(\bar x)\|^2 + \int_0^1 \nabla^2\Phi(\bar x + t\alpha\nabla\Phi(\bar x))(\alpha\nabla\Phi(\bar x))^{\otimes 2}\mathrm{d}t \quad (8)$$
$$\le \alpha + \tilde C_0\alpha^2 < 0, \tag{9}$$

with $r_0 = \min(\tilde r_0, 1/(2\tilde C_0))$, where we have used that $\Phi(\bar x) = 0$, $\|\nabla\Phi(\bar x)\| = 1$ and defined $\tilde C_0 = \sup\{\|\nabla^2\Phi(\bar x + \alpha\nabla\Phi(\bar x))\| \ : \ \bar x \in \partial\mathcal{M}, \ \alpha \in [-\tilde r_0, \tilde r_0]\}$. Reciprocally, we have for any $\alpha \in [0, r_0)$ and $\bar x \in \partial\mathcal{M}$

$$\Phi(\bar x + \alpha\nabla\Phi(\bar x)) = \Phi(\bar x) + \alpha\|\nabla\Phi(\bar x)\|^2 + \int_0^1 \nabla^2\Phi(\bar x + t\alpha\nabla\Phi(\bar x))(\alpha\nabla\Phi(\bar x))^{\otimes 2}\mathrm{d}t \ge \alpha - C_0\alpha^2, \tag{10}$$

where we have used that $\Phi(\bar x) = 0$, $\|\nabla\Phi(\bar x)\| = 1$ and defined $C_0 = \sup\{\|\nabla^2\Phi(\bar x + \alpha\nabla\Phi(\bar x))\| \ : \ \bar x \in \partial\mathcal{M}, \ \alpha \in [-r_0, r_0]\}$. Let $r_1 = \min(r_0, 1/(2C_0))$. Then, $\mathsf{U}_1 = \{\bar x + \alpha\nabla\Phi(\bar x) \ : \ \bar x \in \partial\mathcal{M}, \ \alpha \in (-r_1, r_1)\} \subset \mathbb{R}^d$ is open and

$$\mathsf{U}_1 \cap \overline{\mathcal{M}} = \{\bar x + \alpha\nabla\Phi(\bar x) \ : \ \bar x \in \partial\mathcal{M}, \ \alpha \in [0, r_1)\}. \tag{11}$$

In what follows, we define $\mathsf{U} = \mathsf{U}_1 \cap \overline{\mathcal{M}}$. Note that $\mathsf{U}$ is open for the induced topology and that $\partial\mathcal{M} \subset \mathsf{U}$. In particular, $\partial\mathcal{M}$ is compact, $\mathsf{U}^c$ is closed and $\partial\mathcal{M} \cap \mathsf{U}^c = \varnothing$. Hence, there exists $r > 0$ such that $d(\partial\mathcal{M}, \mathsf{U}^c) \ge 4r$. Without loss of generality we can assume that $r \le 1/2$. We also have $\{x \in \overline{\mathcal{M}} \ : \ d(x, \partial\mathcal{M}) \le 2r\} \subset \mathsf{U}$. The proof of (a) follows from the definition of $\mathsf{U}_0$. In the rest of the proof, we define

$$C^{1/2} = 2\max(1, \sup\{\|\nabla^2\Phi(\bar x + u)\| \ : \ \bar x \in \partial\mathcal{M}, \ \|u\|^2 \le r(r+1)\}), \quad \bar r = \min(1/(2C^{1/2}), r/2). \tag{12}$$

Let us prove (b). Consider $x + \sqrt{\gamma}z \in \mathsf{C}(x, \gamma)$ with $\mathsf{C}(x, \gamma)$ given by (7) and $x = \bar x + \bar\alpha\nabla\Phi(\bar x)$ and $z = \alpha\nabla\Phi(\bar x) + v$ with $v \perp \nabla\Phi(\bar x)$. In particular, we recall that we have

$$-\bar\alpha\gamma^{-1/2} \le \alpha < \bar r\gamma^{-1/2}, \qquad \|v\|^2 \le (\alpha\gamma^{1/2} + \bar\alpha)/(C\gamma). \tag{13}$$

This implies that

$$\bar\alpha + \sqrt{\gamma}\alpha \le 2\bar r, \qquad \gamma\|v\|^2 \le 2\bar r/C. \tag{14}$$

First, using that $C \ge 1$, $\|\nabla\Phi(\bar x)\| = 1$, (14) and (12), we have

$$\|x + \sqrt{\gamma}z - \bar x\|^2 = (\bar\alpha + \sqrt{\gamma}\alpha)^2 + \gamma\|v\|^2 \le r^2 + r/C \le r(r+1). \tag{15}$$

Then, we have that

$$\Phi(x + \sqrt{\gamma}z) = \Phi(\bar x) + \bar\alpha + \sqrt{\gamma}\alpha + \int_0^1 \nabla^2\Phi(\bar x + t(x + \sqrt{\gamma}z - \bar x))(x + \sqrt{\gamma}z - \bar x)^{\otimes 2}\mathrm{d}t \quad (16)$$
$$\ge \bar\alpha + \sqrt{\gamma}\alpha - (C^{1/2}/2)((\bar\alpha + \sqrt{\gamma}\alpha)^2 + \gamma\|v\|^2), \tag{17}$$

where we recall that

$$C^{1/2} = 2\max(1, \sup\{\|\nabla^2\Phi(\bar x + u)\| \ : \ \bar x \in \partial\mathcal{M}, \ \|u\|^2 \le r(r+1)\}), \qquad \bar r = \min(1/(2C^{1/2}), r/2). \tag{18}$$

First, using that $r \le 1/2$ and (14), we have $\bar\alpha + \sqrt{\gamma}\alpha \le 2r \le 1$. Since, $\|v\|^2 \le (\bar\alpha + \sqrt{\gamma}\alpha)/(C\gamma)$ and we have that $\|v\|^2 < 1/(C\gamma)$. Let $P(X) = X - (C^{1/2}/2)X^2 - (C^{1/2}/2)\gamma\|v\|^2$. We have that $P(x) \ge 0$ if and only if $x \in [x_{\min}, x_{\max}]$ with

$$x_{\min} = (1 - (1 - C\gamma\|v\|^2)^{1/2})/C^{1/2}, \qquad x_{\max} = (1 + (1 - C\gamma\|v\|^2)^{1/2})/C^{1/2}. \tag{19}$$

Using that for any $t \in (0, 1)$, $(1 - t)^{1/2} \ge 1 - t$ we have that

$$x_{\min} \le \gamma C\|v\|^2/2, \qquad x_{\max} \ge 1/C^{1/2}. \tag{20}$$

---

[2]This is the tubular neighborhood theorem which is key to the rest of the proof.

Since $\|v\|^2 \leq (\sqrt{\gamma}\alpha + \bar{\alpha})/(\gamma C)$, we have that $\bar{\alpha} + \sqrt{\gamma}\alpha \geq x_{\min}$. In addition, using that $\bar{\alpha} + \sqrt{\gamma}\alpha \leq 2\bar{r} \leq 1/C^{1/2} \leq x_{\max}$, we get that $P(\bar{\alpha} + \sqrt{\gamma}\alpha) \geq 0$ and therefore $x + \sqrt{\gamma}z \in \overline{\mathcal{M}}$ since $\Phi(x + \sqrt{\gamma}z) \geq 0$. This concludes the proof of (b). Note that the condition $\alpha \geq -\gamma^{-1/2}\bar{\alpha}$ is implied by the condition $\|v\|^2 \leq (\sqrt{\gamma}\alpha + \bar{\alpha})/(\gamma C)$. Using that $\mathsf{V} \subset \{x \in \overline{\mathcal{M}} : d(x, \partial\mathcal{M}) \leq 2r\} \subset \mathsf{U}$, (c) is a direct consequence of (Lee et al., 2012, Theorem 6.24)]. Next, we prove (d). Let $x + \sqrt{\gamma}z \in \overline{\mathcal{M}} \cap \mathsf{C}(x, \gamma)^c$. If $\alpha < -\bar{\alpha}\gamma^{-1/2}$ then since $\Phi$ is concave, we have

$$\Phi(x + \sqrt{\gamma}z) = \Phi(\bar{x}) + \bar{\alpha} + \sqrt{\gamma}\alpha + \int_0^1 \nabla^2\Phi(\bar{x} + t(x + \sqrt{\gamma}z - \bar{x}))(x + \sqrt{\gamma}z - \bar{x})^{\otimes 2}\mathrm{d}t < 0,$$
(21)

where we have used that $\Phi(\bar{x}) = 0$. This is absurd, hence either $\alpha \geq \bar{r}\gamma^{-1/2}$ or $\|v\|^2 \geq (\alpha\gamma^{1/2} + \bar{\alpha})/(C\gamma)$ and $\alpha\gamma^{1/2} + \bar{\alpha} \geq 0$, which concludes the proof. The proof of (e) is similar to the proof that $\{x \in \overline{\mathcal{M}} : d(x, \partial\mathcal{M}) \leq 2r\} \subset \mathsf{U}$. $\square$

The main message of Theorem 7 is that using Theorem 7-(d), if we move in the direction of $\nabla\Phi(\bar{x})$ (the inward normal) with magnitude $\alpha$ then we are allowed to move in the orthonormal direction with magnitude $\alpha^{1/2}$. In the next paragraph, we discuss this fact in details and shows it is necessary for the rest of our study.

**The necessity of Theorem 7-(b).** At first sight one can wonder if the statement of Theorem 7-(b) could be simplify. In particular, it would be simpler to replace this statement with: for any $\bar{\alpha} \in [0, \bar{r}]$ and $\bar{x} \in \partial\mathcal{M}$ such that $\bar{x} + \bar{\alpha}\nabla\Phi(\bar{x}) \in \overline{\mathcal{M}}$, let $x = \bar{x} + \bar{\alpha}\nabla\Phi(\bar{x})$ and $\mathsf{C}(x, \gamma)$ such that $x + \sqrt{\gamma}z \in \mathsf{C}(x, \gamma)$ if

$$-\bar{\alpha}\gamma^{-1/2} \leq \alpha < \bar{r}\gamma^{-1/2}, \qquad \|v\|^2 \leq (\alpha\gamma^{1/2} + \bar{\alpha})^2/(C\gamma),$$
(22)

with $z = \alpha\nabla\Phi(\bar{x}) + v$, with $v \perp \nabla\Phi(\bar{x})$. Then $\mathsf{C}(x, \gamma) \subset \overline{\mathcal{M}}$. Note that $\|v\|^2 \leq (\alpha\gamma^{1/2} + \bar{\alpha})/(C\gamma)$ is replaced by $\|v\|^2 \leq (\alpha\gamma^{1/2} + \bar{\alpha})^2/(C\gamma)$, see Figure 8 for an illustration. However, in that case Theorem 7-(d) becomes: in addition, if $x + \sqrt{\gamma}z \in \overline{\mathcal{M}} \cap \mathsf{C}(x, \gamma)^c$ then $\alpha \geq r\gamma^{-1/2}$ or $\|v\|^2 \geq (\alpha\gamma^{1/2} + \bar{\alpha})^2/(C\gamma)$ and $\alpha\gamma^{1/2} + \bar{\alpha} \geq 0$.

In what follows, when controlling the properties of large drift, see the proof of Proposition 18 and the proof of Proposition 21, we need to control quantities of the form $\mathbb{P}(x + \sqrt{\gamma}Z \in \mathsf{C}(x, \gamma)^c \cap \overline{\mathcal{M}})/\sqrt{\gamma}^3$ Using the original Theorem 7-(d) it is possible to show that this quantity is bounded. However, if one uses the updated version of Theorem 7-(d) then one needs to show that there exists $M \geq 0$ and $\bar{\gamma} > 0$ such that for any $\gamma \in (0, \bar{\gamma})$ (here we have assumed that $\bar{\alpha} = 0$, i.e. $x \in \partial\mathcal{M}$ for simplicity)

$$\int_0^{r/\gamma^{-1/2}} \int_{\nabla\Phi(\bar{x})^\perp} \mathbb{1}_{\|v\|^2 \geq \alpha^2}\varphi(v)\varphi(\alpha)\mathrm{d}v\mathrm{d}\alpha \leq M\sqrt{\gamma},$$
(23)

which is absurd.

### D.2 Technical lemmas

We start with a few technical lemmas which will allow us to control some Gaussian probabilities outside of a compact set. We denote $\Psi : \mathbb{R}_+ \times \mathbb{N} \to [0, 1]$ such that for any $k \in \mathbb{N}$, $\Psi(\cdot, k)$ is the tail probability of a $\chi$-squared random variable with parameter $k$, i.e. for any $k \in \mathbb{N}$ and $t \geq 0$ we have

$$\Psi(t, k) = \mathbb{P}(\|Z\|^2 \geq t),$$
(24)

with $Z$ a Gaussian random variable in $\mathbb{R}^k$ with zero mean and identity covariance matrix. We will make extensive use of the following lemma which is a direct consequence of (Laurent et al., 2000, Section 4, Lemma 1).

**Lemma 8.** *For any $k \in \mathbb{N}$ and $t \in \mathbb{R}_+$ with $t \geq 5k$, $\Psi(t, k) \leq \exp[-t/5]$.*

*Proof.* Let $k \in \mathbb{N}$. First, note that for any $x \geq k$, we have that $k + 2(kx)^{1/2} + 2x \leq 5x$. Combining this result and (Laurent et al., 2000, Section 4, Lemma 1, Equation (4.3)), we have that for any $x \geq k$

$$\mathbb{P}(\|X\|^2 \geq 5x) \leq \exp[-x],$$
(25)

with $X$ a $\mathbb{R}^k$-valued Gaussian random variable with zero mean and identity covariance matrix. This concludes the proof upon letting $t = 5x$. $\square$

---

[3] The division by $\sqrt{\gamma}$ comes from the definition of the intermediate drift (55).

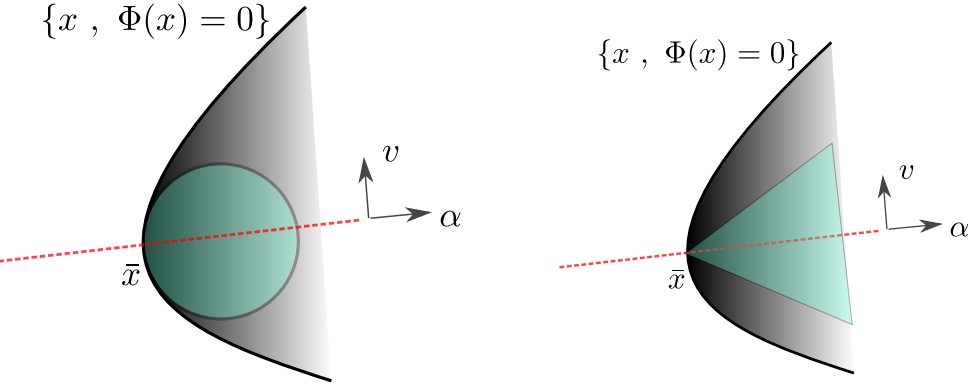

Figure 8: The grey shaded area represents $\overline{\mathcal{M}}$ while the blue shaded area represents $\mathsf{C}(x, \gamma)$ for an arbitrary value of $\gamma$ and $x = \bar{x} \in \partial \mathcal{M}$.

Let $\varphi : \mathbb{R}^p \to \mathbb{R}_+$ given for any $u \in \mathbb{R}$ by $\varphi(u) = (2\pi)^{-p/2} \exp[-\|u\|^2/2]^4$, i.e. the density of a real Gaussian random variable with zero mean and unit variance. While Lemma 9 appears technical, it will be central to provide quantitative upper bounds on the *rejection* probability, see Lemma 12 for instance.

**Lemma 9.** *For any $k \in \mathbb{N}$, $\alpha_0 > 0$, $\beta_0 \in (0, 1]$ and $\delta > 0$ we have*

$$\psi(\delta) = \sup\{\textstyle\int_0^{+\infty} \Psi(\alpha_0 t/\delta, k)^{\beta_0} \varphi(t - t_0/\delta)\mathrm{d}t \; : \; t_0 \geq 0\} \leq C_0\delta, \tag{26}$$

*with $C_0 = 5(2\pi)^{-1/2}(k+1)/(\alpha_0\beta_0)$.*

*Proof.* Let $k \in \mathbb{N}$, $\alpha_0 > 0$, $\beta_0 \in (0, 1]$ and $\delta > 0$. Let $t_\delta = 5k\delta/\alpha_0$. Note that if $t \geq t_\delta$ then, $\alpha_0 t/\delta \geq 5k$. In addition, we have

$$\textstyle\int_0^{+\infty} \Psi(\alpha_0 t/\delta, k)^{\beta_0} \varphi(t - t_0/\delta)\mathrm{d}t \leq (2\pi)^{-1/2} \int_0^{+\infty} \Psi(\alpha_0 t/\delta, k)^{\beta_0}\mathrm{d}t \tag{27}$$

$$\leq (2\pi)^{-1/2} \textstyle\int_0^{t_\delta} \Psi(\alpha_0 t/\delta, k)^{\beta_0}\mathrm{d}t + (2\pi)^{-1/2} \int_{t_\delta}^{+\infty} \Psi(\alpha_0 t/\delta, k)^{\beta_0}\mathrm{d}t. \tag{28}$$

Using that for any $w > 0$, $\int_0^{+\infty} \exp[-wt]\mathrm{d}t \leq (1/w)$, that for any $u \geq 0$, $\Psi(u, k) \leq 1$ and that if $u \geq 5k$, $\Psi(u, k) \leq \exp[-u/5]$, we get for any $t_0 \geq 0$

$$\textstyle\int_0^{+\infty} \Psi(\alpha_0 t/\delta, k)\varphi(t - t_0/\delta) \leq (2\pi)^{-1/2}[5k\delta/\alpha_0 + 5\delta/(\alpha_0\beta_0)] \leq (5(2\pi)^{-1/2}(k+1)/(\alpha_0\beta_0))\delta, \tag{29}$$

which concludes the proof. $\qquad\square$

Finally, we have the following lemma, which is similar to Lemma 8 but will be used to control quantities related to the norm.

**Lemma 10.** *For any $k \in \mathbb{N}$, $\alpha_0 > 0$, $\beta_0 \in (0, 1]$ and $\delta > 0$ we have*

$$\psi(\delta) = \textstyle\int_0^{+\infty} \Psi(\alpha_0 t/\delta, k)^{\beta_0} t\varphi(t)\mathrm{d}t \leq C_0\delta^2, \tag{30}$$

*with $C_0 = 25(2\pi)^{-1}(k^2 + 1)/(\alpha_0\beta_0)^2$.*

*Proof.* Let $k \in \mathbb{N}$, $\alpha_0 > 0$, $\beta_0 \in (0, 1]$ and $\delta > 0$. Let $t_\delta = 5k\delta/\alpha_0$. Note that if $t \geq t_\delta$ then, $\alpha_0 t/\delta \geq 5k$. In addition, we have

$$\textstyle\int_0^{+\infty} \Psi(\alpha_0 t/\delta, k)^{\beta_0} t\varphi(t)\mathrm{d}t \leq (2\pi)^{-1} \int_0^{t_\delta} \Psi(\alpha_0 t/\delta, k)^{\beta_0} t\mathrm{d}t + (2\pi)^{-1} \int_{t_\delta}^{+\infty} \Psi(\alpha_0 t/\delta, k)^{\beta_0} t\mathrm{d}t. \tag{31}$$

---

[4]In the rest of the supplementary, we never precise the dimension $p \in \mathbb{N}$ which can be deduced from the variable.

In addition, using that if $u \geq 5k$ then $\Psi(u,k) \leq \exp[-u/5]$, we get

$$(2\pi)^{-1} \int_{t_\delta}^{+\infty} \Psi(\alpha_0 t/\delta, k)^{\beta_0} t \mathrm{d}t \leq (2\pi)^{-1} \int_0^{+\infty} \exp[-\alpha_0\beta_0 t/(5\delta)] t \mathrm{d}t \leq (2\pi)^{-1} 25\delta^2/(\alpha_0\beta_0)^2. \tag{32}$$

Finally, using that for any $u \geq 0$, $\Psi(u,k) \leq 1$, we have

$$(2\pi)^{-1} \int_0^{t_\delta} \Psi(\alpha_0 t/\delta, k)^{\beta_0} t \mathrm{d}t \leq (2\pi)^{-1} 25 k^2 \delta^2/\alpha_0^2, \tag{33}$$

which concludes the proof. □

### D.3 Lower bound on the inside probability and control of moments of order two and higher

**Lower bound on the inside probability.** We begin with the following lemma which controls the expectation of $1 + \|Z\|$ *outside* of $\mathsf{C}(x,\gamma)$. We recall that $\mathsf{V}$ is defined in Theorem 7-(c).

**Lemma 11.** *Let $\bar{\gamma} = 1$. Let $x \in \mathsf{V}$, $Z \in \sim \mathrm{N}(0,\mathrm{Id})$ and $\gamma \in (0,\bar{\gamma})$ then we have*

$$\max(\mathbb{E}[\mathbb{1}_{x+\sqrt{\gamma}Z\in\overline{\mathcal{M}}\cap\mathsf{C}(x,\gamma)^c}], \mathbb{E}[\langle Z, \nabla\Phi(\bar{x})\rangle \mathbb{1}_{x+\sqrt{\gamma}Z\in\overline{\mathcal{M}}\cap\mathsf{C}(x,\gamma)^c}]) \leq \psi(\gamma), \tag{34}$$

*with $\psi : \mathbb{R}_+ \to \mathbb{R}_+$ such that $\limsup_{t\to 0} \psi(t)/t^{1/2} < +\infty$.*

*Proof.* Let $\bar{r} > 0$ given by Theorem 7. First, we have that

$$\int_{\mathbb{R}} \int_{\mathbb{R}^{d-1}} (1 + |\alpha| + \|v\|) \mathbb{1}_{\alpha \geq \bar{r}/\gamma^{1/2}} \varphi(\alpha)\varphi(v)\mathrm{d}\alpha\mathrm{d}v \tag{35}$$

$$\leq d \int_{\mathbb{R}} (1 + |\alpha|) \mathbb{1}_{\alpha \geq \bar{r}/\gamma^{1/2}} \varphi(\alpha)\mathrm{d}\alpha \leq d(\Psi(\bar{r}^2/\gamma, 1) + \exp[-\bar{r}^2/(2\gamma)]). \tag{36}$$

Second, using Lemma 9, we have that

$$\int_{\mathbb{R}} \int_{\mathbb{R}^{d-1}} \mathbb{1}_{\|v\|^2 \geq (\bar{\alpha}+\sqrt{\gamma}\alpha)/(C\gamma)} \mathbb{1}_{\bar{\alpha}+\sqrt{\gamma}\alpha \geq 0} \varphi(\alpha)\varphi(v)\mathrm{d}\alpha\mathrm{d}v \tag{37}$$

$$\leq \int_{\mathbb{R}} \mathbb{1}_{\bar{\alpha}+\sqrt{\gamma}\alpha \geq 0} \Psi((\bar{\alpha}+\sqrt{\gamma}\alpha)/(C\gamma), d-1)\varphi(\alpha)\mathrm{d}\alpha \tag{38}$$

$$\leq \int_0^{+\infty} \Psi(\alpha/C\gamma^{1/2}, d-1)\varphi(\alpha - \bar{\alpha}/\gamma^{1/2})\mathrm{d}\alpha \leq \Psi_1(\gamma^{1/2}). \tag{39}$$

Second, using Lemma 10, we have that

$$\int_{\mathbb{R}} \int_{\mathbb{R}^{d-1}} \alpha \mathbb{1}_{\|v\|^2 \geq (\bar{\alpha}+\sqrt{\gamma}\alpha)/(C\gamma)} \mathbb{1}_{\bar{\alpha}+\sqrt{\gamma}\alpha \geq 0} \varphi(\alpha)\varphi(v)\mathrm{d}\alpha\mathrm{d}v \tag{40}$$

$$= \int_{\mathbb{R}} \alpha \Psi((\bar{\alpha}+\sqrt{\gamma}\alpha)/(C\gamma), d-1) \mathbb{1}_{\bar{\alpha}+\sqrt{\gamma}\alpha \geq 0} \varphi(\alpha)\mathrm{d}\alpha \tag{41}$$

$$\leq \int_0^{+\infty} \Psi(\alpha/C\gamma^{1/2}, d-1)\alpha\varphi(\alpha)\mathrm{d}\alpha \leq \Psi_2(\gamma^{1/2}). \tag{42}$$

Note that we have $\limsup_{\gamma\to 0} \Psi_2(\gamma^{1/2}) + \Psi_1(\gamma^{1/2}) < +\infty$. We conclude upon combining (36), (39) and (42) with Theorem 7-(d) and the fact that $\|\Phi(\bar{x})\| = 1$. □

The following lemma allow us to give a lower bound to the quantity $\mathbb{E}[\mathbb{1}_{x+\sqrt{\gamma}Z\in\overline{\mathcal{M}}}]$ uniformly w.r.t $x \in \overline{\mathcal{M}}$.

**Lemma 12.** *There exists $\bar{\gamma} > 0$ such that for any $\gamma \in (0,\bar{\gamma})$ and for any $x \in \overline{\mathcal{M}}$, $\gamma \in (0,\bar{\gamma})$ and $Z \sim \mathrm{N}(0,\mathrm{Id})$ we have*

$$\mathbb{E}[\mathbb{1}_{x+\sqrt{\gamma}Z\in\overline{\mathcal{M}}}] \geq 1/4 . \tag{43}$$

*Proof.* Let $\gamma \in (0,\bar{\gamma})$. If $x \notin \mathsf{V}$ then $\mathrm{B}(x, 2R) \subset \mathcal{M}$ using Theorem 7-(e) and therefore $\mathbb{E}[\mathbb{1}_{x+\sqrt{\gamma}Z\in\overline{\mathcal{M}}}] \geq 1/4$ for $\bar{\gamma} > 0$ small enough. Now, assume that $x \in \mathsf{V}$. Using Lemma 11, we have that $\mathbb{E}[\mathbb{1}_{x+\sqrt{\gamma}Z\in\overline{\mathcal{M}}\cap\mathsf{C}(x,\gamma)^c}] \leq \psi(\gamma)$. In addition, using Theorem 7-(b), we have that for any $\gamma > 0$

$$\mathbb{E}[\mathbb{1}_{x+\sqrt{\gamma}Z\in\overline{\mathcal{M}}}] \geq \mathbb{E}[\mathbb{1}_{x+\sqrt{\gamma}Z\in\mathsf{C}(x,\gamma)}] \tag{44}$$

$$\geq \int_{-\bar{\alpha}\gamma^{-1/2}}^{r\gamma^{-1/2}} \int_{\nabla\Phi(\bar{x})^\perp} \mathbb{1}_{\|v\|^2 \leq (\bar{\alpha}+\gamma^{1/2}\alpha)/(C\gamma)} \varphi(\alpha)\varphi(v)\mathrm{d}\alpha\mathrm{d}v \tag{45}$$

$$\geq \int_{-\bar{\alpha}\gamma^{-1/2}}^{r\gamma^{-1/2}} (1 - \Psi((\bar{\alpha}+\gamma^{1/2}\alpha)/(C\gamma), d-1))\varphi(\alpha)\mathrm{d}\alpha \tag{46}$$

$$\geq (1/2) - \Psi(r^2/\gamma, 1) - \int_{-\bar{\alpha}\gamma^{-1/2}}^{+\infty} \Psi((\bar{\alpha}+\gamma^{1/2}\alpha)/(C\gamma), d-1)\varphi(\alpha)\mathrm{d}\alpha. \tag{47}$$

Hence, using Lemma 8 and Lemma 9, there exists $\bar{\gamma} > 0$ such that for any $\gamma \in (0,\bar{\gamma})$, $\Psi(r^2/\gamma, 1) + \int_0^{+\infty} \Psi(\alpha/(C\gamma^{1/2}), d)\varphi(\alpha - \gamma^{1/2}\bar{\alpha})\mathrm{d}\alpha \leq 1/4$, which concludes the proof. □

Note that the result of Lemma 12 can be improved to $1/2 - \varepsilon$ for any $\varepsilon > 0$. In particular this result tells us that for $\gamma > 0$ small enough, $\overline{\mathcal{M}}$ looks like the *hyperplane* from the point of view of the Gaussian with variance $\gamma$ centered on $\partial \mathcal{M}$.

**Bound on moments of order two and higher.** In what follows, we define for any $\gamma > 0$, $\Delta^\gamma : \overline{\mathcal{M}} \to \mathbb{R}_+$ given for any $x \in \overline{\mathcal{M}}$ by

$$\Delta^\gamma(x) = (1/\gamma) \int_{\mathbb{R}^d} \mathbb{1}_{x+\sqrt{\gamma}z \in \mathcal{M}} \|\sqrt{\gamma}z\|^4 \varphi(z)\mathrm{d}z / \int_{\mathbb{R}^d} \mathbb{1}_{x+\sqrt{\gamma}z \in \mathcal{M}} \varphi(z)\mathrm{d}z. \tag{48}$$

**Proposition 13.** *We have* $\lim_{\gamma \to 0} \sup\{\Delta^\gamma(x) : x \in \overline{\mathcal{M}}\} = 0$.

*Proof.* Let $\bar{\gamma} > 0$ given by Lemma 12. Let $x \in \overline{\mathcal{M}}$ and $\gamma \in (0, \bar{\gamma})$. We have using Lemma 12

$$\int_{\mathbb{R}^d} \mathbb{1}_{x+\sqrt{\gamma}z \in \mathcal{M}} \varphi(z)\mathrm{d}z \geq 1/4. \tag{49}$$

We also have that

$$(1/\gamma) \int_{\mathbb{R}^d} \mathbb{1}_{x+\sqrt{\gamma}z \in \mathcal{M}} \|\sqrt{\gamma}z\|^4 \varphi(z)\mathrm{d}z \leq 3\gamma d^2. \tag{50}$$

Therefore, we get that for any $\gamma \in (0, \bar{\gamma})$, $\Delta^\gamma(x) \leq 12\gamma d^2$, which concludes the proof. $\square$

In what follows, we define for any $\gamma > 0$, $\hat{\Sigma}^\gamma : \overline{\mathcal{M}} \to \mathrm{S}_d^+(\mathbb{R})$ given for any $x \in \overline{\mathcal{M}}$ by

$$\hat{\Sigma}^\gamma(x) = \int_{\mathbb{R}^d} \mathbb{1}_{x+\sqrt{\gamma}z \in \mathcal{M}} z \otimes z \varphi(z)\mathrm{d}z / \int_{\mathbb{R}^d} \mathbb{1}_{x+\sqrt{\gamma}z \in \mathcal{M}} \varphi(z)\mathrm{d}z. \tag{51}$$

**Proposition 14.** *There exists* $\bar{\gamma} > 0$ *such that for any* $x \in \overline{\mathcal{M}}$ *and* $\gamma \in (0, \bar{\gamma})$ *we have*

$$\|\hat{\Sigma}^\gamma(x)\| \leq 4d. \tag{52}$$

*Proof.* Let $x \in \overline{\mathcal{M}}$ and $\bar{\gamma} > 0$ given by Lemma 12. For any $\gamma \in (0, \bar{\gamma})$, we have using Lemma 12

$$\int_{\mathbb{R}^d} \mathbb{1}_{x+\sqrt{\gamma}z \in \mathcal{M}} \varphi(z)\mathrm{d}z \geq 1/4. \tag{53}$$

We also have that

$$\int_{\mathbb{R}^d} \mathbb{1}_{x+\sqrt{\gamma}z \in \mathcal{M}} \|z\|^2 \varphi(z)\mathrm{d}z \leq d, \tag{54}$$

which concludes the proof. $\square$

### D.4 Properties of large drift terms

Finally, we define for any $\gamma > 0$, $\hat{b}^\gamma : \overline{\mathcal{M}} \to \mathbb{R}^d$ given for any $x \in \overline{\mathcal{M}}$ by

$$\hat{b}^\gamma(x) = \gamma^{-1/2} \int_{\mathbb{R}^d} \mathbb{1}_{x+\sqrt{\gamma}z \in \mathcal{M}} z \varphi(z)\mathrm{d}z / \int_{\mathbb{R}^d} \mathbb{1}_{x+\sqrt{\gamma}z \in \mathcal{M}} \varphi(z)\mathrm{d}z. \tag{55}$$

First, we show away from the boundary the drift $\hat{b}^\gamma$ converges to zero.

**Proposition 15.** *There exists* $\bar{\gamma} > 0$ *such that for any* $\gamma \in (0, \bar{\gamma})$, $r > 0$ *and* $x \in \overline{\mathcal{M}}$ *such that* $d(x, \partial \mathcal{M}) \geq r$ *we have* $\|\hat{b}^\gamma(x)\| \leq 2d\Psi(r/\gamma, d)^{1/2}/\gamma^{1/2}$.

*Proof.* Let $x \in \overline{\mathcal{M}}$ and $\bar{\gamma} > 0$ given by Lemma 12. For any $\gamma \in (0, \bar{\gamma})$ we have using Lemma 12

$$\int_{\mathbb{R}^d} \mathbb{1}_{x+\sqrt{\gamma}z \in \mathcal{M}} \varphi(z)\mathrm{d}z \geq 1/4. \tag{56}$$

We also have that

$$\| \int_{\mathbb{R}^d} \mathbb{1}_{x+\sqrt{\gamma}z \in \mathcal{M}} z \varphi(z)\mathrm{d}z \| \leq \| \int_{\mathbb{R}^d} \mathbb{1}_{\|z\| \leq r/\gamma^{1/2}} z \varphi(z)\mathrm{d}z \| + \int_{\mathbb{R}^d} \mathbb{1}_{\|z\| \geq r/\gamma^{1/2}} \|z\| \varphi(z)\mathrm{d}z \tag{57}$$

$$\leq 2 \int_{\mathbb{R}^d} \mathbb{1}_{\|z\| \geq r/\gamma^{1/2}} \|z\| \varphi(z)\mathrm{d}z \leq 2d\Psi(r/\gamma, d)^{1/2}/\gamma^{1/2}, \tag{58}$$

which concludes the proof. $\square$

We have the following corollary.

**Corollary 16.** *There exists* $\bar{\gamma} > 0$ *such that for any* $\delta > 0$ *there exists* $M_\delta > 0$ *such that for any* $\gamma \in (0, \bar{\gamma})$ *and* $x \in \overline{\mathcal{M}}$, $\|\hat{b}^\gamma(x)\| \geq M_\delta$, *then* $\Phi(x) \leq \delta$.

*Proof.* Let $\bar{\gamma} > 0$ given by Lemma 12. Let $f : \mathbb{R}_+ \to \mathbb{R}_+$ given for any $r > 0$ by $f(r) = \sup\{\gamma > 0 : \Psi(r/\gamma, 1)^{1/2}/\gamma^{1/2}\}$. We have that $f$ is non-increasing and $\lim_{r \to 0} f(r) = +\infty$. Let $\delta > 0$ and $M_\delta = 2df(\delta/C)$ with $C = \sup\{\|\nabla\Phi(x)\| : x \in \overline{\mathcal{M}}\}$. Let $\gamma \in (0, \bar{\gamma})$ and $x \in \overline{\mathcal{M}}$ such that $\|\hat{b}^\gamma(x)\| \geq M_\delta$ then using Proposition 15 we have that $d(x, \partial\mathcal{M}) \leq \delta/C$. Let $\bar{x} \in \partial\mathcal{M}$ such that $\|x - \bar{x}\| = d(x, \partial\mathcal{M})$. We have

$$\Phi(x) \leq \Phi(\bar{x}) + \int_0^1 \langle \nabla\Phi(\bar{x} + t(x - \bar{x})), x - \bar{x}\rangle \mathrm{d}t \leq \delta, \tag{59}$$

which concludes the proof. $\qquad\square$

For ease of notation, for any $\gamma > 0$, we define $\bar{b}^\gamma = \gamma^{1/2}\hat{b}^\gamma$, the *renormalized* version of the drift. First, we have the following result which will ensure that the drift projected on the normal component does not vanish.

**Lemma 17.** *There exists $\bar{\gamma} > 0$ such that for any $\gamma \in (0, \bar{\gamma})$ and $x \in \mathsf{V}$ we have*

$$\langle \bar{b}^\gamma(x), \nabla\Phi(\bar{x})\rangle \geq \|\bar{b}^\gamma(x)\| - \psi(\gamma), \tag{60}$$

*with $\psi : \mathbb{R}_+ \to \mathbb{R}_+$ such that $\limsup_{\gamma \to 0} \psi(\gamma)/\sqrt{\gamma} < +\infty$.*

*Proof.* Let $x \in \overline{\mathcal{M}}$ and $\bar{\gamma} > 0$ given by Lemma 12. For any $\gamma \in (0, \bar{\gamma})$ we have using Lemma 12

$$\int_{\mathbb{R}^d} \mathbb{1}_{x+\sqrt{\gamma}z \in \mathcal{M}}\varphi(z)\mathrm{d}z \geq 1/4. \tag{61}$$

In addition, we have

$$\int_{\mathbb{R}^d} \mathbb{1}_{x+\sqrt{\gamma}z \in \mathcal{M}}\langle z, \nabla\Phi(\bar{x})\rangle\varphi(z)\mathrm{d}z \geq \int_{\mathbb{R}^d} \mathbb{1}_{x+\sqrt{\gamma}z \in \mathsf{C}(x,\gamma)}\langle z, \nabla\Phi(\bar{x})\rangle\varphi(z)\mathrm{d}z \tag{62}$$

$$- \int_{\mathbb{R}^d} \mathbb{1}_{x+\sqrt{\gamma}z \in \mathcal{M} \cap \mathsf{C}(x,\gamma)^c}\langle z, \nabla\Phi(\bar{x})\rangle\varphi(z). \tag{63}$$

Using Lemma 11, we get that

$$\int_{\mathbb{R}^d} \mathbb{1}_{x+\sqrt{\gamma}z \in \mathcal{M}}\langle z, \nabla\Phi(\bar{x})\rangle\varphi(z)\mathrm{d}z \geq \int_{\mathbb{R}^d} \mathbb{1}_{x+\sqrt{\gamma}z \in \mathsf{C}(x,\gamma)}\langle z, \nabla\Phi(\bar{x})\rangle\varphi(z)\mathrm{d}z - \psi(\gamma). \tag{64}$$

Let $\{e_i\}_{i=1}^{d-1}$ a basis of $\nabla\Phi(\bar{x})^\perp$. Using Theorem 7-(b), we have that for any $i \in \{1, \ldots, d-1\}$

$$\int_{\mathbb{R}^d} \mathbb{1}_{x+\sqrt{\gamma}z \in \mathsf{C}(x,\gamma)}\langle z, e_i\rangle\varphi(z)\mathrm{d}z = \int_{-\bar{\alpha}/\gamma^{1/2}}^{r/\gamma^{1/2}} \int_{\nabla\Phi(\bar{x})^\perp} \mathbb{1}_{\|v\|^2 \leq (\gamma^{1/2}\alpha+\bar{\alpha})/\gamma}\langle v, e_i\rangle\varphi(v)\varphi(\alpha)\mathrm{d}v\mathrm{d}\alpha. \tag{65}$$

Hence, combining this result and the Cauchy-Schwarz inequality we have for any $i \in \{1, \ldots, d-1\}$

$$\left(\int_{\mathbb{R}^d} \mathbb{1}_{x+\sqrt{\gamma}z \in \mathsf{C}(x,\gamma)}\langle z, e_i\rangle\varphi(z)\mathrm{d}z\right)^2 = \left(\int_{-\bar{\alpha}/\gamma^{1/2}}^{r/\gamma^{1/2}} \int_{\nabla\Phi(\bar{x})^\perp} \mathbb{1}_{\|v\|^2 \geq (\gamma^{1/2}\alpha+\bar{\alpha})/\gamma}\langle v, e_i\rangle\varphi(v)\varphi(\alpha)\mathrm{d}v\mathrm{d}\alpha\right)^2 \tag{66}$$

$$\leq \int_{\nabla\Phi(\bar{x})^\perp} \langle v, e_i\rangle^2\varphi(v)\mathrm{d}v \left(\int_{-\bar{\alpha}/\gamma^{1/2}}^{r/\gamma^{1/2}} \Psi((\bar{\alpha}+\alpha\gamma^{1/2})/\gamma, d-1)^{1/2}\varphi(\alpha)\mathrm{d}\alpha\right)^2 \tag{67}$$

$$\leq \left(\int_{-\bar{\alpha}/\gamma^{1/2}}^{r/\gamma^{1/2}} \Psi((\bar{\alpha}+\alpha\gamma^{1/2})/\gamma, d-1)^{1/2}\varphi(\alpha)\mathrm{d}\alpha\right)^2. \tag{68}$$

Hence, using Lemma 9, we get that

$$\sum_{i=1}^{d-1}\left(\int_{\mathbb{R}^d} \mathbb{1}_{x+\sqrt{\gamma}z \in \mathsf{C}(x,\gamma)}\langle z, e_i\rangle\varphi(z)\mathrm{d}z\right)^2 \leq (d-1)\psi^2(\gamma), \tag{69}$$

with $\psi$ given by Lemma 9 with $\beta_0 = 1/2$. Therefore, we get that

$$\left(\int_{\mathbb{R}^d} \mathbb{1}_{x+\sqrt{\gamma}z \in \mathsf{C}(x,\gamma)}\langle z, \nabla\Phi(\bar{z})\rangle\varphi(z)\mathrm{d}z\right)^2 \tag{70}$$

$$= \left(\int_{\mathbb{R}^d} \mathbb{1}_{x+\sqrt{\gamma}z \in \mathcal{M}}\varphi(z)\mathrm{d}z\right)^2\|\bar{b}^\gamma(x)\|^2 - \sum_{i=1}^{d-1}\left(\int_{\mathbb{R}^d} \mathbb{1}_{x+\sqrt{\gamma}z \in \mathsf{C}(x,\gamma)}\langle z, e_i\rangle\varphi(z)\mathrm{d}z\right)^2 \tag{71}$$

$$\geq \left(\int_{\mathbb{R}^d} \mathbb{1}_{x+\sqrt{\gamma}z \in \mathcal{M}}\varphi(z)\mathrm{d}z\right)^2\|\bar{b}^\gamma(x)\|^2 - \psi(\gamma)^2. \tag{72}$$

We conclude the proof upon using that for any $a, b \geq 0$, $(a+b)^{1/2} \leq a^{1/2} + b^{1/2}$ and (61). $\qquad\square$

We are now ready to state the following lower bound on the drift.

**Proposition 18.** *There exist $\bar{\gamma} > 0$, $M \geq 0$ and $c > 0$ such that for any $x \in \overline{\mathcal{M}}$ and $\gamma \in (0, \bar{\gamma})$ if $\|\hat{b}^\gamma(x)\| \geq M$ then $x \in \mathsf{V}$ and*

$$\min(\langle \hat{b}^\gamma(x), \nabla\Phi(x)\rangle, \langle \hat{b}^\gamma(x), \nabla\Phi(\bar{x})\rangle) \geq c\|\hat{b}^\gamma(x)\|. \tag{73}$$

*Proof.* Let $\bar{\gamma} > 0$ given by Lemma 12 and $M_0 = 4\sup\{\psi(\gamma)/\gamma^{1/2} : \gamma \in (0, \bar{\gamma}]\}$. In addition, let $c = 1/4$. Using Proposition 15 and Theorem 7-(e), there exists $M_1 \geq 0$ such that for any any $x \in \overline{\mathcal{M}}$, if $\|\hat{b}^\gamma(x)\| \geq M_1$ then $x \in \mathsf{V}$ and $x = \bar{x} + \alpha\nabla\Phi(\bar{x})$ with $\alpha \leq 1/(4C)$ and $C = \sup\{\|\nabla^2\Phi(x)\| : x \in \overline{\mathcal{M}}\}$. We denote $M = \max(M_0, M_1)$. Let $\gamma \in (0, \bar{\gamma})$ and $x \in \overline{\mathcal{M}}$ such that $\|\hat{b}^\gamma(x)\| \geq M$. Using Lemma 17, we have that

$$\langle \hat{b}^\gamma(x), \nabla\Phi(\bar{x})\rangle \geq \|\hat{b}^\gamma(x)\| - \psi(\gamma)/\gamma^{1/2}. \tag{74}$$

Using that $\psi(\gamma)/\gamma^{1/2} \leq M/2 \leq \|\hat{b}^\gamma(x)\|/2$, we have

$$\langle \hat{b}^\gamma(x), \nabla\Phi(\bar{x})\rangle \geq (1/2)\|\hat{b}^\gamma(x)\|. \tag{75}$$

Since $\|x - \bar{x}\| \leq \alpha \leq 1/(4C)$ we have $\langle \hat{b}^\gamma(x), \nabla\Phi(x)\rangle \geq (1/2 - C\alpha)\|\hat{b}^\gamma(x)\| \geq \|\hat{b}^\gamma(x)\|/4$, which concludes the proof. □

### D.5 Convergence on compact sets

In this section, we show the convergence of the drift and diffusion matrix on compact sets. We recall that $\mathcal{M}$ does *not* include its boundary $\partial\mathcal{M}$.

**Proposition 19.** *For any compact set $\mathsf{K} \subset \mathcal{M}$ and $\varepsilon > 0$, there exists $\bar{\gamma} > 0$ such that for any $\gamma \in (0, \bar{\gamma})$ we have for any $x \in \mathsf{K}$*

$$\|\hat{b}^\gamma(x)\| \leq \varepsilon, \qquad \|\hat{\Sigma}^\gamma(x) - \mathrm{Id}\| \leq \varepsilon. \tag{76}$$

*Proof.* Let $\mathsf{K} \subset \mathcal{M}$ be a compact set and $\gamma > 0$. Since $\mathsf{K} \cap \partial\mathcal{M} = \varnothing$, there exists $r > 0$ such that for any $x \in \mathsf{K}$, $d(x, \partial\mathcal{M}) > r$. Therefore, we have that for any $x \in \mathsf{K}$

$$\|\hat{b}^\gamma(x)\| = \gamma^{-1/2}\|\int_{x+\sqrt{\gamma}z\in\mathcal{M}} z\varphi(z)\mathrm{d}z\|/\int_{x+\sqrt{\gamma}z\in\mathcal{M}}\varphi(z)\mathrm{d}z. \tag{77}$$

In addition, using the Cauchy-Schwarz inequality we have

$$\|\int_{x+\sqrt{\gamma}z\in\mathcal{M}} z\varphi(z)\mathrm{d}z\| \leq \|\int_{\mathbb{R}^d} z\varphi(z)\mathrm{d}z\| + \int_{\mathcal{M}^c}\|z\|\varphi(z)\mathrm{d}z \tag{78}$$

$$\leq \int_{\mathbb{R}^d}\mathbb{1}_{\|z\|\geq r/\gamma^{1/2}}\|z\|\varphi(z)\mathrm{d}z \leq \sqrt{d}\Psi(r^2/\gamma, d)^{1/2}. \tag{79}$$

Using Lemma 8 and Lemma 12, there exists $\bar{\gamma}_0 > 0$ such that for any $\gamma \in (0, \bar{\gamma}_0)$ we have that for any $x \in \mathsf{K}$

$$\|\hat{b}^\gamma(x)\| \leq 4d\Psi(r^2/\gamma, 1)^{1/2}/\gamma^{1/2} \leq \varepsilon, \tag{80}$$

which concludes the first part of the proof. Similarly, we have that for any $x \in \mathsf{K}$

$$\|\int_{x+\sqrt{\gamma}z\in\mathcal{M}}(z\otimes z - \mathrm{Id})\varphi(z)\mathrm{d}z\| \leq \|\int_{\mathbb{R}^d}(z\otimes z - \mathrm{Id})\varphi(z)\mathrm{d}z\| + \int_{\mathcal{M}^c}\|z\|\varphi(z)\mathrm{d}z \tag{81}$$

$$\leq \int_{\mathbb{R}^d}\mathbb{1}_{\|z\|\geq r/\gamma^{1/2}}\|z\otimes z - \mathrm{Id}\|\varphi(z)\mathrm{d}z \tag{82}$$

$$\leq \sqrt{2}(1 + 3d^2)^{1/2}\Psi(r^2/\gamma, d)^{1/2}. \tag{83}$$

Using Lemma 8 and Lemma 12, there exists $\bar{\gamma}_1 > 0$ such that for any $\gamma \in (0, \bar{\gamma}_1)$, we have that for any $x \in \mathsf{K}$

$$\|\hat{\Sigma}^\gamma(x) - \mathrm{Id}\| \leq 4\sqrt{2}(1 + 3d^2)^{1/2}\Psi(r^2/\gamma, 1)^{1/2} \leq \varepsilon, \tag{84}$$

which concludes the proof upon letting $\bar{\gamma} = \min(\bar{\gamma}_0, \bar{\gamma}_1)$. □

### D.6 Convergence on the boundary

Finally, we investigate the behavior at the boundary of the diffusion matrix and the drift. First, we show that there is a lower bound to the diffusion matrix near the boundary. Second, we show that the renormalized drift converges to the outward normal.

**Proposition 20.** *There exist $c > 0$ and $\bar{\gamma} > 0$ such that for any $\gamma \in (0, \bar{\gamma})$, $u \in \mathbb{R}^d$ and $x \in \mathsf{V}$ we have*

$$\langle u, \hat{\Sigma}^\gamma(x)u \rangle \geq c\|u\|^2. \tag{85}$$

*In particular, there exist $r, \varepsilon > 0$ such that for any $\gamma \in (0, \bar{\gamma})$ and $x \in \overline{\mathcal{M}}$ with $d(x, \partial\mathcal{M}) \leq r$*

$$\langle \nabla\Phi(x), \hat{\Sigma}^\gamma(x)\nabla\Phi(x) \rangle \geq \varepsilon. \tag{86}$$

*Proof.* First, we show (85). Let $x \in \mathsf{V}$. We have for any $u \in \mathbb{R}^d$

$$\langle u, \hat{\Sigma}^\gamma(x)u \rangle = \int_{\mathbb{R}^d} \mathbb{1}_{x + \sqrt{\gamma}z \in \mathcal{M}} \langle z, u \rangle^2 \varphi(z)\mathrm{d}z / \int_{\mathbb{R}^d} \mathbb{1}_{x + \sqrt{\gamma}z \in \mathcal{M}}\mathrm{d}z \tag{87}$$

$$\geq \int_{\mathbb{R}^d} \mathbb{1}_{x + \sqrt{\gamma}z \in \mathsf{C}(x,\gamma)} \langle z, u \rangle^2 \varphi(z)\mathrm{d}z. \tag{88}$$

For any $u \in \mathbb{R}^d$, let $\alpha_u = \langle u, \nabla\Phi(\bar{x}) \rangle$. Using Theorem 7-(b) we have for any $u \in \mathbb{R}^d$

$$\int_{\mathbb{R}^d} \mathbb{1}_{x + \sqrt{\gamma}z \in \mathsf{C}(x,\gamma)} \langle z, u \rangle^2 \varphi(z)\mathrm{d}z \tag{89}$$

$$= \int_{-\bar{\alpha}/\gamma^{1/2}}^{r/\gamma^{1/2}} \int_{\nabla\Phi(\bar{x})^\perp} (\langle u, v \rangle + \alpha\alpha_u)^2 \mathbb{1}_{\|v\|^2 \leq (\alpha\gamma^{1/2} + \bar{\alpha})/\gamma}\varphi(v)\varphi(\alpha)\mathrm{d}v\mathrm{d}\alpha \tag{90}$$

$$\geq \int_0^{r/\gamma^{1/2}} \int_{\nabla\Phi(\bar{x})^\perp} (\langle u, v \rangle^2 + \alpha^2\alpha_u^2) \mathbb{1}_{\|v\|^2 \leq (\alpha\gamma^{1/2} + \bar{\alpha})/\gamma}\varphi(v)\varphi(\alpha)\mathrm{d}v\mathrm{d}\alpha \tag{91}$$

$$\geq \alpha_u^2 \int_0^{r/\gamma^{1/2}} \alpha^2\varphi(\alpha)\mathrm{d}\alpha + \int_{-\bar{\alpha}/\gamma^{1/2}}^{r/\gamma^{1/2}} \int_{\nabla\Phi(\bar{x})^\perp} \langle u, v \rangle^2 \mathbb{1}_{\|v\|^2 \leq (\alpha\gamma^{1/2} + \bar{\alpha})/\gamma}\varphi(v)\varphi(\alpha)\mathrm{d}v\mathrm{d}\alpha. \tag{92}$$

Using Cauchy-Schwarz inequality, we have

$$\int_0^{r/\gamma^{1/2}} \alpha^2\varphi(\alpha)\mathrm{d}\alpha = (1/2) - \int_{r/\gamma^{1/2}}^{+\infty} \alpha^2\varphi(\alpha)\mathrm{d}\alpha \geq (1/2) - 3\Phi(r^2/\gamma, 1)^{1/2}. \tag{93}$$

In addition, using the Cauchy-Schwarz inequality, we have that

$$\int_{-\bar{\alpha}/\gamma^{1/2}}^{r/\gamma^{1/2}} \int_{\nabla\Phi(\bar{x})^\perp} \langle u, v \rangle^2 \mathbb{1}_{\|v\|^2 \leq (\alpha\gamma^{1/2} + \bar{\alpha})/\gamma}\varphi(v)\varphi(\alpha)\mathrm{d}v\mathrm{d}\alpha \tag{94}$$

$$= \int_{\nabla\Phi(\bar{x})^\perp} \langle u, v \rangle^2 \varphi(v)\mathrm{d}v \int_{-\bar{\alpha}/\gamma^{1/2}}^{r/\gamma^{1/2}} \varphi(\alpha)\mathrm{d}\alpha \tag{95}$$

$$- \int_{-\bar{\alpha}/\gamma^{1/2}}^{r/\gamma^{1/2}} \int_{\nabla\Phi(\bar{x})^\perp} \langle u, v \rangle^2 \mathbb{1}_{\|v\|^2 \geq (\alpha\gamma^{1/2} + \bar{\alpha})/\gamma}\varphi(v)\varphi(\alpha)\mathrm{d}v\mathrm{d}\alpha \tag{96}$$

$$\geq (\|u\|^2 - \alpha_u^2)((1/2) - \Phi(r^2/\gamma, 1)) \tag{97}$$

$$- \sqrt{3}(d-1)\|u\|^2 \int_0^{+\infty} \Phi(\alpha/\gamma^{1/2}, d-1)^{1/2}\varphi(\alpha - \bar{\alpha}/\gamma^{1/2})\mathrm{d}\alpha. \tag{98}$$

Combining this result, (93), (92) and Lemma 9 there exists $\bar{\gamma} > 0$ such that for any $\gamma \in (0, \bar{\gamma}]$ and $u \in \mathbb{R}^d$

$$\int_{\mathbb{R}^d} \mathbb{1}_{x + \sqrt{\gamma}z \in \mathsf{C}(x,\gamma)} \langle z, u \rangle^2 \varphi(z)\mathrm{d}z \geq (1/4)\|u\|^2, \tag{99}$$

which concludes the proof of (85). Finally, using Theorem 7-(e), we have that for any $x \in \overline{\mathcal{M}}$ if $d(x, \partial\mathcal{M}) \leq R$ then $x \in \mathsf{V}$. Let $r = \min(R, 1/(2C))$ with $C = \sup\{\|\nabla^2\Phi(x)\| : x \in \overline{\mathcal{M}}\}$. We have that for any $x \in \overline{\mathcal{M}}$ such that $d(x, \partial\mathcal{M}) \leq r$

$$\|\nabla\Phi(x)\| \geq \|\nabla\Phi(\bar{x}_0)\| - Cr \geq 1/2, \tag{100}$$

where $\bar{x}_0$ is such that $\|x - \bar{x}_0\| \leq r$ and $\bar{x}_0 \in \partial\mathcal{M}$. Combining this result and (99) concludes the proof upon letting $\varepsilon = 1/16$. $\square$

Finally, we investigate the behavior of the normalized drift near the boundary.

**Proposition 21.** *For any $\bar{x}_0 \in \partial\mathcal{M}$ and $\varepsilon > 0$, there exist $\bar{\gamma}, r, M > 0$ such that for any $x \in \overline{\mathcal{M}}$ and $\gamma \in (0, \bar{\gamma})$ with $\|x - \bar{x}_0\| \leq r$ and $\|\hat{b}^\gamma(x)\| \geq M$*

$$\|\hat{b}^\gamma(x)/\langle \hat{b}^\gamma(x), \nabla\Phi(x) \rangle - \nabla\Phi(\bar{x}_0)\| \leq \varepsilon. \tag{101}$$

*Proof.* Let $\bar{\gamma}$ be given by Proposition 18. Let $\psi$ given by Lemma 9 and $M_0 = \sup\{\psi(\gamma)/\gamma^{1/2} : \gamma \in (0,\bar{\gamma})\} < +\infty$. Let $M = 16M_0/(c\varepsilon^{1/2})$ with $c$ given in Proposition 18. Let $R > 0$ given by Theorem 7-(e) such that for any $x \in \overline{\mathcal{M}}$ with $d(x, \partial\mathcal{M})$ there exist $\bar{x} \in \partial\mathcal{M}$ and $\alpha \in [0, c\varepsilon/(4C)]$ such that $x = \bar{x} + \alpha\nabla\Phi(\bar{x})$ with $C = \sup\{\|\nabla^2\Phi(x)\| : x \in \overline{\mathcal{M}}\}$ and $c$ given in Proposition 18. Let $r = \min(\bar{r}, c\varepsilon/4, R)$ and $x \in \overline{M}$ with $\|x - \bar{x}_0\| \leq r$. First, since $d(x, \partial\mathcal{M}) \leq R$, there exist $\bar{x} \in \partial\mathcal{M}$ and $\alpha \in [0, \varepsilon/(4C)]$ such that $x = \bar{x} + \alpha\nabla\Phi(\bar{x})$. Therefore, we get that $\|\bar{x} - \bar{x}_0\| \leq \varepsilon/(2C)$ and therefore $\|\nabla\Phi(\bar{x}_0) - \nabla\Phi(\bar{x})\| \leq \varepsilon/2$. In addition, we have that

$$\|\hat{b}^\gamma(x)/\langle\hat{b}^\gamma(x), \nabla\Phi(x)\rangle - \hat{b}^\gamma(x)/\langle\hat{b}^\gamma(x), \nabla\Phi(\bar{x})\rangle\| \tag{102}$$

$$\leq \|\hat{b}^\gamma(x)\|^2\|\nabla\Phi(x) - \nabla\Phi(\bar{x})\|/(\langle\hat{b}^\gamma(x), \nabla\Phi(x)\rangle\langle\hat{b}^\gamma(x), \nabla\Phi(\bar{x})\rangle). \tag{103}$$

Using Proposition 18, we get that

$$\|\hat{b}^\gamma(x)/\langle\hat{b}^\gamma(x), \nabla\Phi(x)\rangle - \hat{b}^\gamma(x)/\langle\hat{b}^\gamma(x), \nabla\Phi(\bar{x})\rangle\| \leq \varepsilon/4. \tag{104}$$

In what follows, we show that

$$\|\hat{b}^\gamma(x)/\langle\hat{b}^\gamma(x), \nabla\Phi(\bar{x})\rangle - \nabla\Phi(\bar{x})\|^2 \leq \varepsilon/2. \tag{105}$$

In particular, we show that for any $u \in \nabla\Phi(\bar{x})^\perp$ with $\|u\| = 1$,

$$\langle\hat{b}^\gamma(x), u\rangle^2 \leq (\varepsilon/16)\langle\hat{b}^\gamma(x), \nabla\Phi(\bar{x})\rangle^2. \tag{106}$$

Assuming (106), letting $u = (\hat{b}^\gamma(x) - \langle\hat{b}^\gamma(x), \nabla\Phi(\bar{x})\rangle)/(\|\hat{b}^\gamma(x)\|^2 - \langle\hat{b}^\gamma(x), \nabla\Phi(\bar{x})^2\rangle)^{1/2}$ and using that $\hat{b}^\gamma(x) = \langle\hat{b}^\gamma(x), u\rangle u + \langle\hat{b}^\gamma(x), \nabla\Phi(\bar{x})\rangle\nabla\Phi(\bar{x})$ we have

$$\|\hat{b}^\gamma(x)/\langle\hat{b}^\gamma(x), \nabla\Phi(x)\rangle - \nabla\Phi(\bar{x})\| \leq \|\hat{b}^\gamma(x)/\langle\hat{b}^\gamma(x), \nabla\Phi(\bar{x})\rangle - \nabla\Phi(\bar{x})\| \tag{107}$$

$$+ \|\hat{b}^\gamma(x)/\langle\hat{b}^\gamma(x), \nabla\Phi(x)\rangle - \hat{b}^\gamma(x)/\langle\hat{b}^\gamma(x), \nabla\Phi(\bar{x})\rangle\| \tag{108}$$

$$\leq |\langle\hat{b}^\gamma(x), u\rangle/\langle\hat{b}^\gamma(x), \nabla\Phi(\bar{x})\rangle| + \varepsilon/4 \leq \varepsilon/2, \tag{109}$$

which concludes the proof. Let $u \in \nabla\Phi(\bar{x})^\perp$ with $\|u\| = 1$ and $\{e_i\}_{i=1}^{d-1}$ an orthonormal basis of $\nabla\Phi(\bar{x})^\perp$. There exist $\{a_i\}_{i=1}^{d-1}$ such that $\sum_{i=1}^{d-1} a_i^2 = 1$ and $u = \sum_{i=1}^{d-1} a_i e_i$. Using Theorem 7-(b), we have that for any $i \in \{1, \ldots, d-1\}$

$$\int_{\mathbb{R}^d} \mathbb{1}_{x+\sqrt{\gamma}z \in \mathsf{C}(x,\gamma)}\langle z, e_i\rangle\varphi(z)\mathrm{d}z = \int_{-\bar{\alpha}/\gamma^{1/2}}^{r/\gamma^{1/2}} \int_{\nabla\Phi(\bar{x})^\perp} \mathbb{1}_{\|v\|^2 \leq (\gamma^{1/2}\alpha+\bar{\alpha})/\gamma}\langle v, e_i\rangle\varphi(v)\varphi(\alpha)\mathrm{d}v\mathrm{d}\alpha \tag{110}$$

$$= \int_{-\bar{\alpha}/\gamma^{1/2}}^{r/\gamma^{1/2}} \int_{\nabla\Phi(\bar{x})^\perp} \mathbb{1}_{\|v\|^2 \geq (\gamma^{1/2}\alpha+\bar{\alpha})/\gamma}\langle v, e_i\rangle\varphi(v)\varphi(\alpha)\mathrm{d}v\mathrm{d}\alpha \tag{111}$$

Hence, combining this result and the Cauchy-Schwarz inequality we have for any $i \in \{1, \ldots, d-1\}$

$$(\int_{\mathbb{R}^d} \mathbb{1}_{x+\sqrt{\gamma}z \in \mathsf{C}(x,\gamma)}\langle z, e_i\rangle\varphi(z)\mathrm{d}z)^2 = (\int_{-\bar{\alpha}/\gamma^{1/2}}^{r/\gamma^{1/2}} \int_{\nabla\Phi(\bar{x})^\perp} \mathbb{1}_{\|v\|^2 \geq (\gamma^{1/2}\alpha+\bar{\alpha})/\gamma}\langle v, e_i\rangle\varphi(v)\varphi(\alpha)\mathrm{d}v\mathrm{d}\alpha)^2 \tag{112}$$

$$\leq \int_{\nabla\Phi(\bar{x})^\perp}\langle v, e_i\rangle^2\varphi(v)\mathrm{d}v(\int_{-\bar{\alpha}/\gamma^{1/2}}^{r/\gamma^{1/2}} \Psi((\bar{\alpha}+\alpha\gamma^{1/2})/\gamma, d-1)^{1/2}\varphi(\alpha)\mathrm{d}\alpha)^2. \tag{113}$$

Hence, we get that

$$\sum_{i=1}^{d-1} a_i^2(\int_{\mathbb{R}^d} \mathbb{1}_{x+\sqrt{\gamma}z \in \mathsf{C}(x,\gamma)}\langle z, e_i\rangle\varphi(z)\mathrm{d}z)^2 \leq \|u\|^2\psi^2(\gamma), \tag{114}$$

with $\psi$ given by Lemma 9. Recalling that $\|\hat{b}^\gamma(x)\| \geq M$ we have

$$\langle\hat{b}^\gamma(x), u\rangle^2 \leq 16\psi(\gamma)^2/\gamma \leq c^2(\varepsilon/16)M^2 \leq (\varepsilon/16)\langle\hat{b}^\gamma(x), \nabla\Phi(\bar{x})\rangle^2, \tag{115}$$

which concludes the proof. $\square$

## D.7 Submartingale problem and weak solution

We are now ready to conclude the proof.

**Theorem 22.** *There exists $\mathbb{P}^\star$ a distribution on $\mathrm{D}([0,T],\overline{\mathcal{M}})$ such that $\lim_{\gamma \to 0} \hat{\mathbb{P}}^\gamma = \mathbb{P}^\star$. In addition, for any $f \in \mathrm{C}^{1,2}([0,T] \times \overline{\mathcal{M}}, \mathbb{R})$ with $\langle \nabla \Phi(\bar{x}), \nabla f(x) \rangle \geq 0$ for any $t \in [0,T]$ and $x \in \partial \mathcal{M}$, we have that the process $(f(t, \omega(t)))_{t \in [0,T]}$ given for any $t \in [0,T]$*

$$f(t, \omega(t)) - \int_0^t (\partial_s f(s, \omega(s)) + \tfrac{1}{2} \Delta f(s, \omega(s))) \mathbb{1}_{\mathcal{M}}(\omega(s)) \mathrm{d}s, \tag{116}$$

*is a $\mathbb{P}$ submartingale.*

*Proof.* Condition (A) (Stroock et al., 1971, p.197) is a consequence of Proposition 13. Condition (B) (Stroock et al., 1971, p.197) is a consequence of Proposition 18. Condition (C) (Stroock et al., 1971, p.198) is a consequence of Corollary 16. Condition (D) (Stroock et al., 1971, p.198) is a consequence of Proposition 14. We fix $\rho = 0$ and condition (1) (Stroock et al., 1971, p.203) is a consequence of Proposition 19. Condition (2)-(iii) (Stroock et al., 1971, p.203) is a consequence of Proposition 20. Condition (2)-(iv) (Stroock et al., 1971, p.203) is a consequence of Proposition 21. We conclude upon using (Stroock et al., 1971, Theorem 6.3) and (Stroock et al., 1971, Theorem 5.8). □

We finally conclude the proof of Theorem 6 upon using the results of (Kang et al., 2017) which establish the link between a weak solution to the reflected SDE and the solution to a submartingale problem.

**Theorem 23.** *For any $T \geq 0$, $(\hat{\mathbf{X}}_t^\gamma)_{t \in [0,T]}$ weakly converges to $(\mathbf{X}_t)_{t \in [0,T]}$ such that for any $t \in [0,T]$*

$$\mathbf{X}_t = x + \mathbf{B}_t - \mathbf{k}_t, \qquad |\mathbf{k}|_t = \int_0^t \mathbb{1}_{\mathbf{X}_s \in \partial \mathcal{M}} \mathrm{d}|\mathbf{k}|_s, \qquad \mathbf{k}_t = \int_0^t \mathbf{n}(\mathbf{X}_s) \mathrm{d}|\mathbf{k}|_s. \tag{117}$$

*Proof.* Using Theorem 22 and (Kang et al., 2017, Theorem 1, Proposition 2.12), we have that $\mathbb{P}$ in Theorem 22 is associated with a solution to the extended Skorokhod problem. We conclude that a solution to the extended Skorokhod problem is a solution to the Skorokhod problem using (Ramanan, 2006, Corollary 2.10). □

## D.8 Extension to the Metropolis process

We recall that the Metropolis process is defined as follows. Let $(X_k^\gamma)_{k \in \mathbb{N}}$ given for any $\gamma > 0$ and $k \in \mathbb{N}$ by $X_0^\gamma = x \in \overline{\mathcal{M}}$ and for $X_{k+1}^\gamma = X_k^\gamma + \sqrt{\gamma} Z_k$ if $X_k^\gamma + \sqrt{\gamma} Z_k^\gamma \in \overline{\mathcal{M}}$ and $X_k^\gamma$ otherwise, $Z_k \sim \mathrm{N}(0, \mathrm{Id})$. We recall that $\hat{b}^\gamma$, $\hat{\Sigma}^\gamma$ and $\hat{\Delta}^\gamma$ are given by (48), (51) and (55). In particular, denoting $\hat{\mathrm{K}}^\gamma$ the Markov kernel associated with $(\hat{X}_k^\gamma)_{k \in \mathbb{N}}$, i.e. $\hat{\mathrm{K}}^\gamma : \mathcal{M} \times \mathcal{B}(\mathcal{M}) \to [0,1]$ such that for any $x \in \mathcal{M}$, $\hat{\mathrm{K}}^\gamma(x, \cdot)$ is a probability measure, for any $\mathsf{A} \in \mathcal{B}(\mathcal{M})$, $\hat{\mathrm{K}}^\gamma(\cdot, \mathsf{A})$ is a measurable function and $\mathbb{E}[\mathbb{1}_\mathsf{A}(\hat{X}_1^\gamma) \mid \hat{X}_0^\gamma = x] = \hat{\mathrm{K}}^\gamma(x, \mathsf{A})$. We have that for any $\gamma > 0$ and $x \in \mathcal{M}$

$$\hat{b}^\gamma(x) = (1/\gamma) \int_{\mathcal{M}} (y - x) \hat{\mathrm{K}}^\gamma(x, \mathrm{d}y), \tag{118}$$

$$\hat{\Sigma}^\gamma(x) = (1/\gamma) \int_{\mathcal{M}} (y - x)^{\otimes 2} \hat{\mathrm{K}}^\gamma(x, \mathrm{d}y), \tag{119}$$

$$\hat{\Delta}^\gamma(x) = (1/\gamma) \int_{\mathcal{M}} \|y - x\|^4 \hat{\mathrm{K}}^\gamma(x, \mathrm{d}y). \tag{120}$$

In what follows, we denote $a^\gamma(x) = \mathbb{E}[\mathbb{1}_{x + \sqrt{\gamma} Z_0 \in \mathcal{M}}]$. Denote $\mathrm{K}^\gamma$ the kernel associated with $(X_k^\gamma)_{k \in \mathbb{N}}$. We have that for any $\mathsf{A} \in \mathcal{B}(\mathcal{M})$, $\gamma > 0$ and $x \in \mathcal{M}$

$$\mathrm{K}^\gamma(x, \mathsf{A}) = \mathbb{E}[\mathbb{1}_{X_{k+1}^\gamma \in \mathsf{A}} \mathbb{1}_{x + \sqrt{\gamma} Z_{k+1} \in \mathcal{M}}] + (1 - a^\gamma(x)) \mathbb{1}_\mathsf{A}(x) \tag{121}$$

$$= a^\gamma(x) \hat{\mathrm{K}}^\gamma(x, \mathsf{A}) + (1 - a^\gamma(x)) \mathbb{1}_\mathsf{A}(x). \tag{122}$$

We define for any $\gamma > 0$ and $x \in \mathcal{M}$

$$b^\gamma(x) = (1/\gamma) \int_{\mathcal{M}} (y - x) \mathrm{K}^\gamma(x, \mathrm{d}y), \tag{123}$$

$$\Sigma^\gamma(x) = (1/\gamma) \int_{\mathcal{M}} (y - x)^{\otimes 2} \mathrm{K}^\gamma(x, \mathrm{d}y), \tag{124}$$

$$\Delta^\gamma(x) = (1/\gamma) \int_{\mathcal{M}} \|y - x\|^4 \mathrm{K}^\gamma(x, \mathrm{d}y). \tag{125}$$

Using (122), we get that for any $\gamma > 0$ and $x \in \mathcal{M}$

$$b^\gamma(x) = a^\gamma(x)\hat{b}^\gamma(x), \qquad \Sigma^\gamma(x) = a^\gamma(x)\hat{\Sigma}^\gamma(x), \qquad \Delta^\gamma(x) = a^\gamma(x)\hat{\Delta}^\gamma(x). \tag{126}$$

Using Lemma 12, we have that for any $\gamma \in (0, \bar{\gamma})$ and $x \in \mathcal{M}$, $a^\gamma(x) \geq 1/4$.

In order to conclude for the convergence of the Metropolis process we adapt Theorem 22 and Theorem 23. We define $\mathbf{X}^\gamma : \mathbb{R}_+ \to \overline{\mathcal{M}}$ given for any $k \in \mathbb{N}$ by $\mathbf{X}^\gamma_{k\gamma} = X^\gamma_k$ and for any $t \in [k\gamma, (k+1)\gamma)$, $\mathbf{X}^\gamma_t = X^\gamma_k$. Note that $(\mathbf{X}_t)_{t \in [0,T]}$ is a $\mathrm{D}([0,T], \overline{\mathcal{M}})$ valued random variable, where $\mathrm{D}([0,T], \overline{\mathcal{M}})$ is the space of right-continuous with left-limit processes which take values in $\overline{\mathcal{M}}$. We denote $\mathbb{P}^\gamma$ the distribution of $(\mathbf{X}_t)_{t \in [0,T]}$ on $\mathrm{D}([0,T], \overline{\mathcal{M}})$.

**Theorem 24.** *There exists $\mathbb{P}^\star$ a distribution on $\mathrm{D}([0,T], \overline{\mathcal{M}})$ such that $\lim_{\gamma \to 0} \mathbb{P}^\gamma = \mathbb{P}^\star$. In addition, for any $f \in \mathrm{C}^{1,2}([0,T] \times \overline{\mathcal{M}}, \mathbb{R})$ with $\langle \nabla\Phi(\bar{x}), \nabla f(x) \rangle \geq 0$ for any $t \in [0,T]$ and $x \in \partial\mathcal{M}$, we have that the process $(f(t, \omega(t)))_{t \in [0,T]}$ given for any $t \in [0,T]$*

$$f(t, \omega(t)) - \int_0^t (\partial_s f(s, \omega(s) + \tfrac{1}{2}\Delta f(s, \omega(s)))) \mathbb{1}_{\mathcal{M}}(\omega(s))\mathrm{d}s, \tag{127}$$

*is a $\mathbb{P}$ submartingale.*

*Proof.* Condition (A) (Stroock et al., 1971, p.197) is a consequence of Proposition 13 and (126). Condition (B) (Stroock et al., 1971, p.197) is a consequence of Proposition 18 and (126). Condition (C) (Stroock et al., 1971, p.198) is a consequence of Corollary 16 and (126). Condition (D) (Stroock et al., 1971, p.198) is a consequence of Proposition 14 and (126). We fix $\rho = 0$ and condition (1) (Stroock et al., 1971, p.203) is a consequence of Proposition 19 and that $\lim_{\gamma \to 0} a^\gamma = 1$ uniformly on compact subsets $\mathsf{K} \subset \mathcal{M}$. Condition (2)-(iii) (Stroock et al., 1971, p.203) is a consequence of Proposition 20 and (126). Condition (2)-(iv) (Stroock et al., 1971, p.203) is a consequence of Proposition 21 and (126). We conclude upon using (Stroock et al., 1971, Theorem 6.3) and (Stroock et al., 1971, Theorem 5.8). $\square$

**Theorem 25.** *For any $T \geq 0$, $(\mathbf{X}^\gamma_t)_{t \in [0,T]}$ weakly converges to $(\mathbf{X}_t)_{t \in [0,T]}$ such that for any $t \in [0,T]$*

$$\mathbf{X}_t = x + \mathbf{B}_t - \mathbf{k}_t, \qquad |\mathbf{k}|_t = \int_0^t \mathbb{1}_{\mathbf{X}_s \in \partial\mathcal{M}}\mathrm{d}|\mathbf{k}|_s, \qquad \mathbf{k}_t = \int_0^t \mathbf{n}(\mathbf{X}_s)\mathrm{d}|\mathbf{k}|_s. \tag{128}$$

*Proof.* The proof is identical to Theorem 23. $\square$

## E  Modelling geospatial data within non-convex boundaries

To demonstrate the ability of the proposed method to model distributions whose support is restricted to manifolds with highly non-convex boundaries, we derived a geospatial dataset based on the historical wildfire incidence rate within the continental United States (described in in Appendix E.1) and, using the corresponding country borders, trained a constrained diffusion model by adapting the point-in-spherical-polytope conditions outlined in (Ketzner et al., 2022) (described in Appendix E.2).

### E.1  Derivation of bounded geospatial dataset

Specifically, we retrieved the rasterised version of the wildfire data provided by Welty et al. (2020), converted it to a spherical geodetic coordinate system using the CARTOPY library (Met Office, 2010 - 2015), and drew a weighted subsample of size $1 \times 10^6$. We then retrieved the country borders of the continental United States from (Natural Earth, 2023) and mapped them to the same geodetic reference frame as the wildfire data. A visualization of the resulting dataset is presented in Figure 9.

### E.2  Point-in-spherical-polytope algorithms

The support of the data-generating distribution we aim to approximate is thus restricted to a highly non-convex spherical polytope $\mathbb{P} \in \mathcal{S}^2$ given by the country borders of the continental United States. To determine whether a query point $q \in \mathcal{S}^2$ is within $\mathbb{P}$, we adapt an efficient reformulation of the point-in-spherical-polygon algorithm (Bevis et al., 1989) presented in (Ketzner et al., 2022). The algorithm requires the provision of a reference point $r \in \mathcal{S}^2$ known to be located in $\mathbb{P}$ and determines

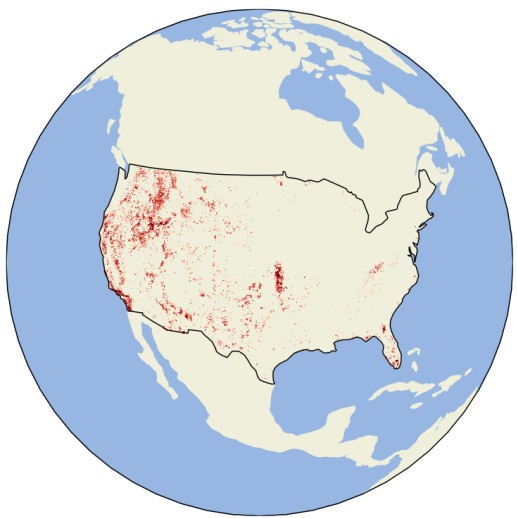

Figure 9: Orthographic projection of the wildfire dataset described in Appendix E. The projection is aligned with the centroid of the continental United States and zoomed in ten-fold for visual clarity. All visualisations of geospatial data were generated using the GEOVIEWS (Rudiger et al., 2023) and DATASHADER (Bednar et al., 2023) libraries.

whether $q$ is inside or outside the polygon by checking whether the geodesic between $r$ and $q$ crosses the polygon an even or odd number of times. Letting $\hat{x} \in \mathbb{R}^3$ denote the Cartesian coordinates of a point $x \in \mathcal{S}^2$, (Ketzner et al., 2022) rely on a Cartesian reference coordinate system $\hat{Q}$ (with its $z$-axis given by $\hat{r}$) and the corresponding spherical coordinate system $Q$ to decompose the edge-crossing condition of Bevis et al. (1989) into two efficiently computable parts. That is, the geodesic between $q$ and $r$ crosses an edge $e_i = (v_i, v_j)$ of the polygon if:

(i) the longitude of $q$ in $Q$ is bounded by the longitudes of $v_i$ and $v_j$ in $Q$, i.e.

$$\phi_Q(q) \in [\min(\phi_Q(v_i), \phi_Q(v_j)), \max(\phi_Q(v_i), \phi_Q(v_j))],$$

(ii) the plane specified by the normal vector $\hat{p}_i = \hat{v}_i \times \hat{v}_j$ represents an equator that separates $q$ and $r$ into two different hemispheres, i.e.

$$\mathrm{sign}(\langle \hat{p}_i, \hat{r} \rangle \cdot \langle \hat{p}_i, \hat{q} \rangle) = -1.$$

Especially when $\mathbb{P}$ is fixed and the corresponding coordinate transformations and normal vectors can be precomputed for each edge, this algorithm affords an efficient and parallelisable approach to determining whether any given point on $\mathcal{S}^2$ is contained by a spherical polytope.

## F  Supplementary Experimental Results

### F.1  Evaluating log-barrier and Euclidean models

Following (Fishman et al., 2023), we approached the empirical evaluation of our Metropolis model by computing the maximum mean discrepancy (MMD) (Gretton et al., 2012) between samples from the true distribution and the trained diffusion models. The MMD is a statistic that quantifies the similarity of two samples by computing the distance of their respective mean embeddings in a reproducing kernel Hilbert space. For this, we use an RBF kernel with the same length scales as the standard deviations of the normal distributions used to generate the synthetic distribution. We sum these RBF kernels by the weights of the corresponding components of the synthetic Gaussian mixture model.

This is essential to be able to include the log-barrier in the comparison since the log-barrier methods suffer severe instabilities around the boundary, as the space is stretched to more and more. These instabilities cause the problems in fitting the log-barrier model and in computing the likelihood using the log-barrier model.

Table 5: Maximum mean discrepancy (MMD) ($\downarrow$) of a held-out test set from a synthetic bimodal distribution over convex subsets of $\mathbb{R}^d$ bounded by the hypercube $[-1, 1]^d$ and unit simplex $\Delta^d$. Means and standard deviations are computed over 3 different runs.

| Manifold | Dimension | Process | MMD | | % in Manifold |
| | | | mean | std | mean |
| --- | --- | --- | --- | --- | --- |
| $\Delta^d$ | 2 | Euclidean | 0.027 | 0.011 | 0.969 |
| | | Log-Barrier | 0.050 | 0.012. | 1.000 |
| | | Reflected | 0.041 | 0.008 | 1.000 |
| | | Rejection | 0.030 | 0.002 | 1.000 |
| | 3 | Euclidean | 0.032 | 0.015 | 0.969 |
| | | Log-Barrier | 0.238 | 0.009 | 1.000 |
| | | Reflected | 0.179 | 0.013 | 1.000 |
| | | Rejection | 0.111 | 0.002 | 1.000 |
| | 10 | Euclidean | 0.028 | 0.001 | 0.946 |
| | | Log-Barrier | 0.275 | 0.0015 | 1.000 |
| | | Reflected | 0.233 | 0.004 | 1.000 |
| | | Rejection | 0.226 | 0.005 | 1.000 |
| $[0, 1]^d$ | 2 | Euclidean | 0.069 | 0.004 | 0.992 |
| | | Log-Barrier | 0.66 | 0.006 | 1.000 |
| | | Reflected | 0.048 | 0.012 | 1.000 |
| | | Rejection | 0.025 | 0.005 | 1.000 |
| | 3 | Euclidean | 0.074 | 0.004 | 0.991 |
| | | Log-Barrier | 0.209 | 0.0077 | 1.000 |
| | | Reflected | 0.085 | 0.006 | 1.000 |
| | | Rejection | 0.049 | 0.006 | 1.000 |
| | 10 | Euclidean | 0.086 | 0.007 | 0.968 |
| | | Log-Barrier | 0.330 | 0.004 | 1.000 |
| | | Reflected | 0.314 | 0.049 | 1.000 |
| | | Rejection | 0.138 | 0.007 | 1.000 |

From the results in Table 5, it is clear that the log-barrier approach performs significantly worse than the Reflected model and the Metropolis models across all settings. This, in conjunction with numerical instabilities we encountered when attempting to evaluate sample likelihoods with the log-barrier models as presented in (Fishman et al., 2023), motivated us to focus on the Reflected and Metropolis models in the main text.

Additionally, we note that the unconstrained Euclidean models outperform the constrained methods on both the simplex and the hypercube as the dimensionality of the problem space increases. Especially on the simplex, we attribute this performance primarily to the fact that the synthetic distribution is simply a standard Normal with only a small portion close to the boundary. The amount of reflection needed to model the distribution decreases in higher dimensions, as the mass of the Normal distribution gets increasingly concentrated—which Euclidean diffusion models will fit well. This same dynamic is partially responsible for the hypercube performance.

### F.2 Implementational details

All source code that is needed to reproduce the results presented below is made available under https://github.com/oxcsml/score-sde/tree/metropolis, which requires a supporting package to handle the different geometries that is available under https://github.com/oxcsml/geomstats/tree/polytope.

We use the same architecture in all of our experiments: a 6-layer MLP with 512 hidden units and sine activation functions, except in the output layer, which uses a linear activation function. Following (Fishman et al., 2023), we implement a simple linear function that scales the score by the distance to the boundary, approaching zero within $\epsilon = 0.01$ of the boundary. This ensures the score obeys the Neumann boundary conditions required by the reflected Brownian Motion. For the geospatial dataset within non-convex country borders, we do not use distance rescaling. Instead, we substitute

it with a series of step functions to rescale the score. This is a proof-of-concept to show that even when computing the distance is hard, simple and efficient approximations suffice. When constructing Riemannian diffusion models on the torus and sphere for the protein and geospatial datasets, we follow (De Bortoli et al., 2022) and include an additional preconditioner for the score on the manifold. We *do not* use the residual trick or the standard deviation trick, which are both common score-rescaling functions in image model architectures; in our setting, we find that they adversely affect model training.

For the forward/reverse process we always set $T = 1$, $\beta_0 = 1 \times 10^{-3}$ and then tune $\beta_1$ to ensure that the forward process just reaches the invariant distribution with a linear $\beta$-schedule. At sampling time we use $N = 100$ steps of the discretised process. We discretise the training process by selecting a random $N$ between 0 and 100 for each example, rolling out to that time point. This lets us cheaply implement a simple variance reduction technique: we take multiple samples from this trajectory by selecting multiple random $N$ to save for each example. This technique was originally described in (Fishman et al., 2023) and we find it is also helpful for our Metropolis models. For all experiments, we use the ism loss with a modified weighting function of $(1 + t)$, which we found to be essential to model training. All experiments use a batch size of 256 with 8 repeats per batch. For training, we use a learning rate of $2 \times 10^{-4}$ with a cosine learning rate schedule. We trained for 100,000 batches on the synthetic examples and 300,000 batches on the real-world examples (robotics, proteins, wildfires).

We selected these hyperparameters from a systematic search over learning rates ($6 \times 10^{-4}$, $2 \times 10^{-4}$, $6 \times 10^{-5}$, $2 \times 10^{-5}$), learning rate schedules (cosine, log-linear), and batch sizes (128, 256, 512, 1024) on synthetic examples for the reflected and log-barrier models. Similar parameters worked well for both, and we used those for our Metropolis models to allow a straightforward comparison. We tried $N = 100, 1000$ for several synthetic examples but found that very large rollout times actually hurt performance for the Metropolis model, though the log-barrier performed a bit better with longer rollouts and the reflected was the same.

All models were trained on a single NVIDIA GeForce GTX 1080 GPU. All of the Metropolis models presented here can easily be trained on this hardware in under 4 hours. The runtime for the log-barrier and reflected models is considerably longer.

## F.3 Synthetic Distributions on Constrained Manifolds of Increasing Dimensionality

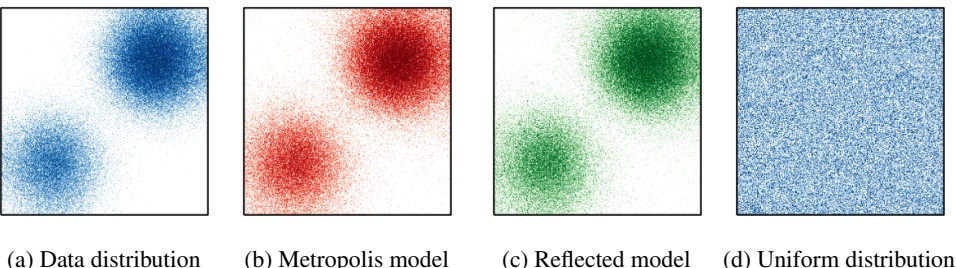

    (a) Data distribution    (b) Metropolis model    (c) Reflected model    (d) Uniform distribution

Figure 10: Qualitiative comparison of samples from the data distribution, our Metropolis model, a Reflected model and the uniform distribution for a synthetic bimodal distribution on $[-1, 1]^2$.

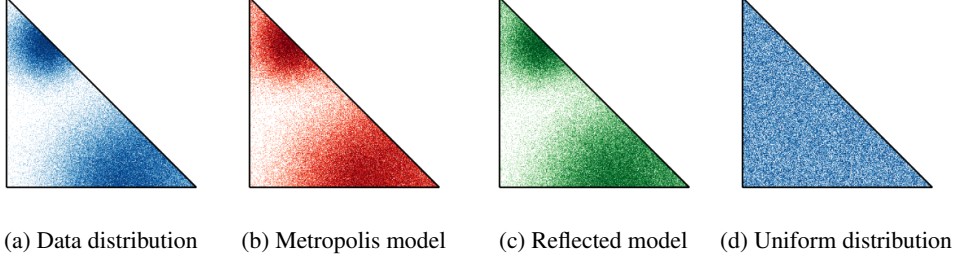

    (a) Data distribution    (b) Metropolis model    (c) Reflected model    (d) Uniform distribution

Figure 11: Qualitiative comparison of samples from the data distribution, our Metropolis model, a Reflected model and the uniform distribution for a synthetic bimodal distribution on $\Delta^2$.

## F.4 Constrained Configurational Modelling of Robotic Arms

The following univariate marginal and pairwise bivariate plots visualise the distribution of different samples in

(i) the three dimensions needed to describe an ellipsoid $M = \begin{bmatrix} l_1 & l_2 \\ l_2 & l_3 \end{bmatrix} \in \mathcal{S}_{++}^2$ and

(ii) the two dimensions needed to describe a location in $\mathbb{R}^2$.

### F.4.1 Visualisation of samples from the data distribution

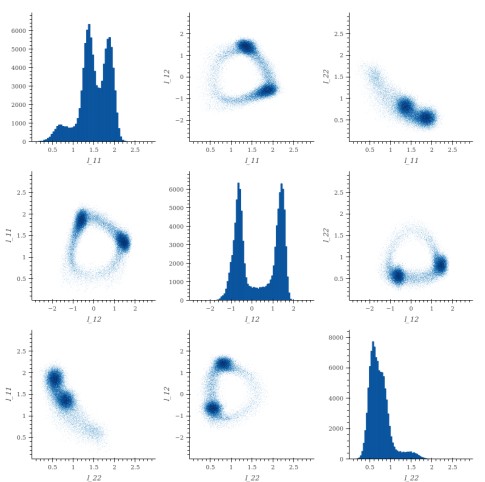

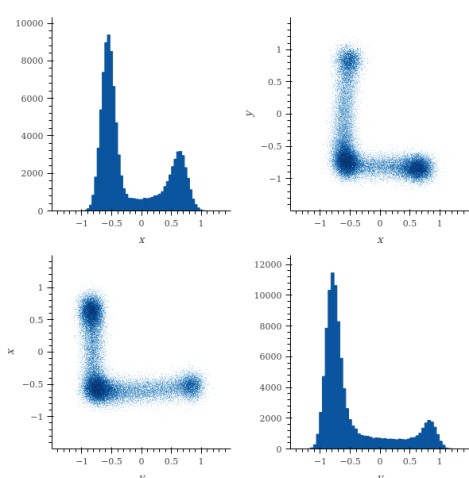

(a) Plots of the univariate marginal and pairwise bivariate distributions of $1 \times 10^5$ samples from the data distribution in $\mathcal{S}_{++}^2$.

(b) Plots of the univariate marginal and pairwise bivariate distributions of $1 \times 10^5$ samples from the data distribution in $\mathbb{R}^2$.

Figure 12: Visualisation of the data distribution in $\mathcal{S}_{++}^2 \times \mathbb{R}^2$ using univariate marginal and pairwise bivariate plots.

### F.4.2 Visualisation of samples from our Metropolis sampling-based diffusion model

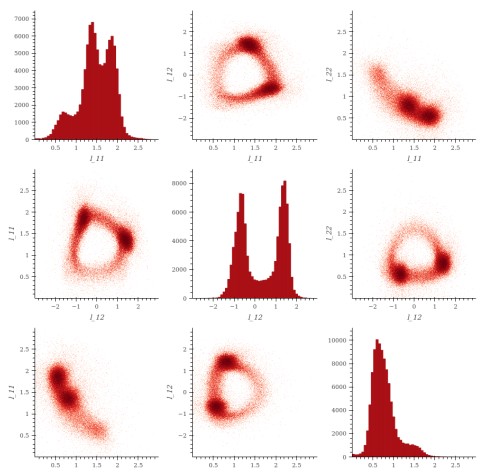

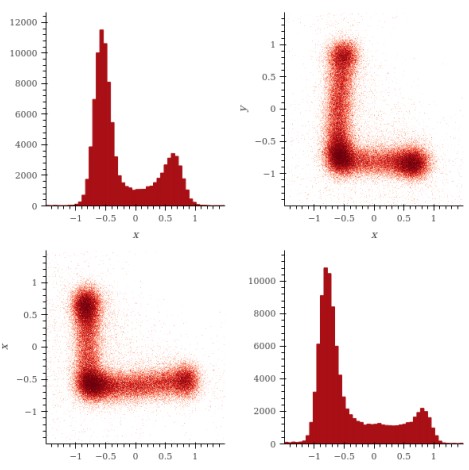

(a) Plots of the univariate marginal and pairwise bivariate distributions of $1 \times 10^5$ samples from our Metropolis sampling-based diffusion model in $\mathcal{S}_{++}^2$.

(b) Plots of the univariate marginal and pairwise bivariate distributions of $1 \times 10^5$ samples from our Metropolis sampling-based diffusion model in $\mathbb{R}^2$.

Figure 13: Visualisation of the distribution learned by our Metropolis sampling-based diffusion model in $\mathcal{S}_{++}^2 \times \mathbb{R}^2$ using univariate marginal and pairwise bivariate plots.

### F.4.3 Visualisation of samples from a reflected Brownian motion-based diffusion model

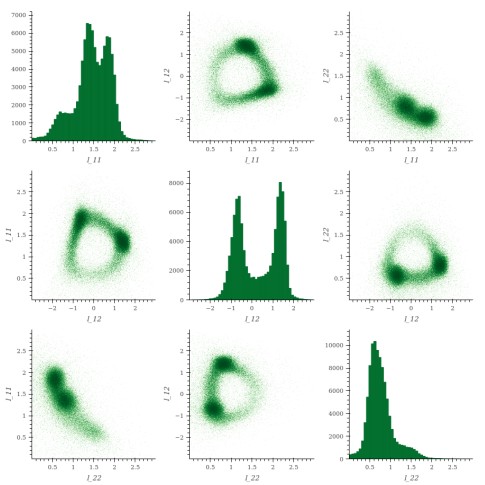

(a) Plots of the univariate marginal and pairwise bivariate distributions of $1 \times 10^5$ samples from a reflected Brownian motion-based diffusion model in $\mathcal{S}_{++}^2$.

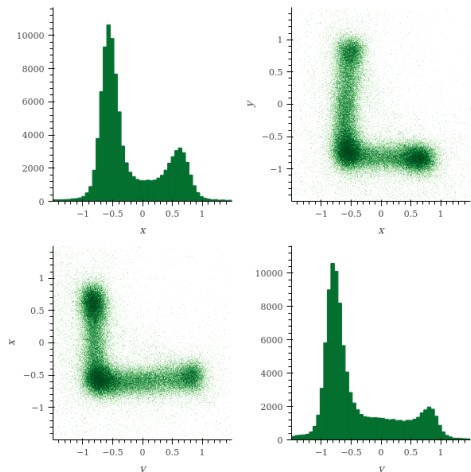

(b) Plots of the univariate marginal and pairwise bivariate distributions of $1 \times 10^5$ samples from a reflected Brownian motion-based diffusion model in $\mathbb{R}^2$.

Figure 14: Visualisation of the distribution learned by a reflected Brownian motion-based diffusion model in $\mathcal{S}_{++}^2 \times \mathbb{R}^2$ using univariate marginal and pairwise bivariate plots.

### F.4.4 Visualisation of samples from the uniform distribution

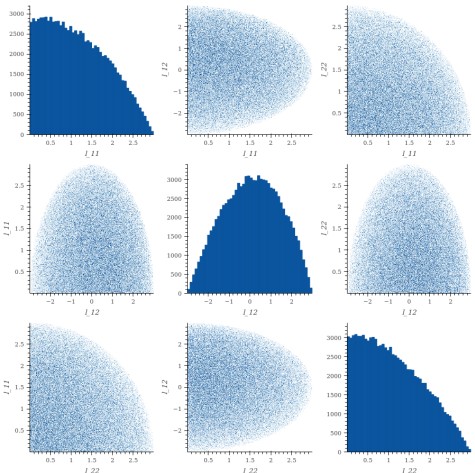

(a) Plots of the univariate marginal and pairwise bivariate distributions of $1 \times 10^5$ samples from the uniform distribution in $\mathcal{S}_{++}^2$.

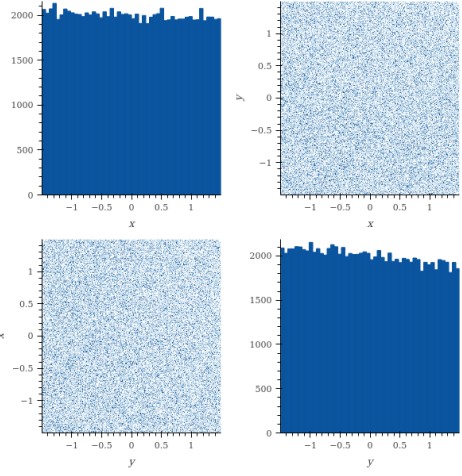

(b) Plots of the univariate marginal and pairwise bivariate distributions of $1 \times 10^5$ samples from the uniform distribution in $\mathbb{R}^2$.

Figure 15: Visualisation of the uniform distribution in $\mathcal{S}_{++}^2 \times \mathbb{R}^2$ using univariate marginal and pairwise bivariate plots.

### F.5 Conformational Modelling of Protein Backbones

The following univariate marginal and pairwise bivariate plots visualise the distribution of different samples in (i) the polytope $\mathbb{P} \subset \mathbb{R}^3$ and (ii) the torus $\mathbb{T}^4$ used to parametrise the conformations of a polypeptide chain of length $N = 6$ with coinciding endpoints. We refer to (Han et al., 2006) for full detail on the reparametrisation and to (Fishman et al., 2023) for a full description of the dataset.

### F.5.1 Visualisation of samples from the data distribution

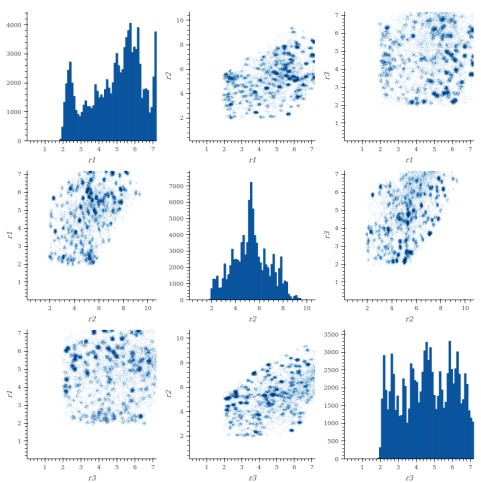

(a) Plots of the univariate marginal and pairwise bivariate distributions of $1 \times 10^5$ samples from the data distribution in $\mathbb{P} \subset \mathbb{R}^3$.

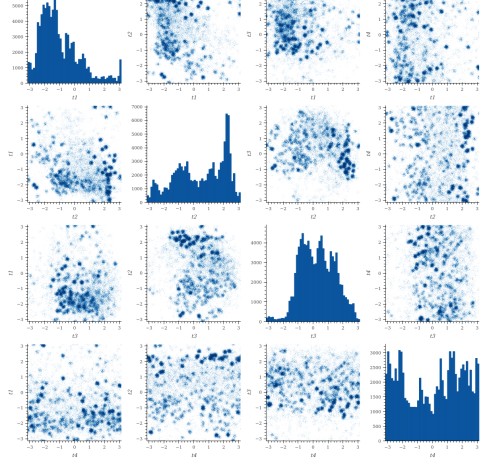

(b) Plots of the univariate marginal and pairwise bivariate distributions of $1 \times 10^5$ samples from the data distribution in $\mathbb{T}^4$.

Figure 16: Visualisation of the data distribution in $\mathbb{P} \subset \mathbb{R}^3 \times \mathbb{T}^4$ using univariate marginal and pairwise bivariate plots.

### F.5.2 Visualisation of samples from our Metropolis sampling-based diffusion model

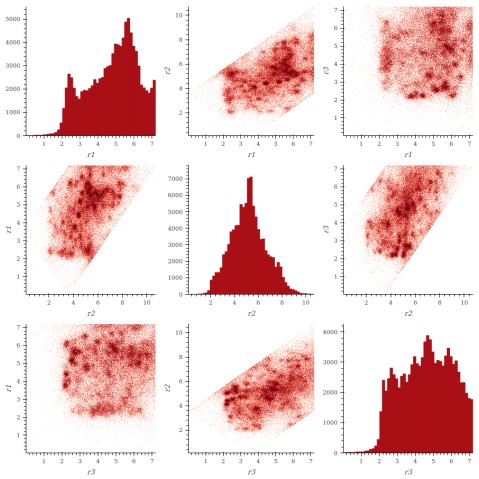

(a) Plots of the univariate marginal and pairwise bivariate distributions of $1 \times 10^5$ samples from our Metropolis model in $\mathbb{P} \subset \mathbb{R}^3$.

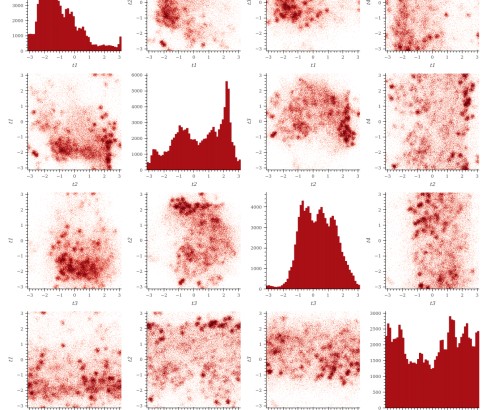

(b) Plots of the univariate marginal and pairwise bivariate distributions of $1 \times 10^5$ samples from our Metropolis model in $\mathbb{T}^4$.

Figure 17: Visualisation of the distribution learned by our Metropolis model in $\mathbb{P} \subset \mathbb{R}^3 \times \mathbb{T}^4$ using univariate marginal and pairwise bivariate plots.

### F.5.3 Visualisation of samples from a reflected Brownian motion-based diffusion model

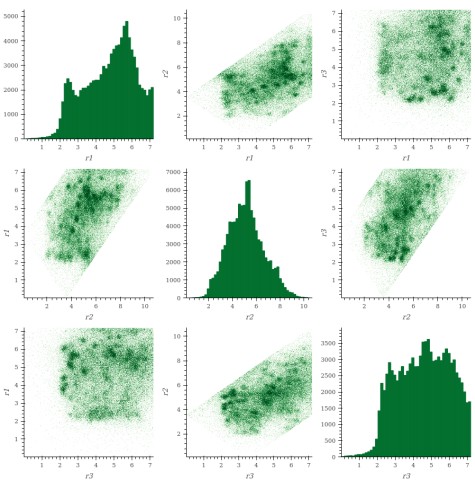 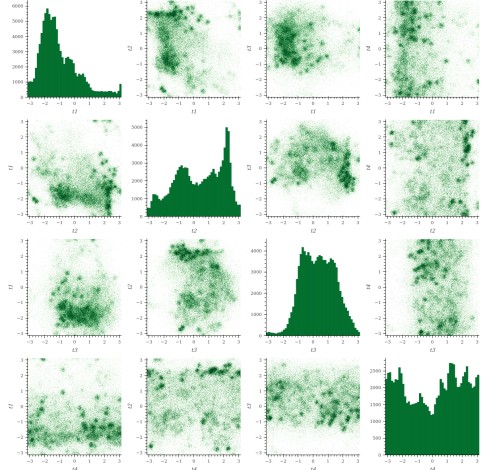

(a) Plots of the univariate marginal and pairwise bivariate distributions of $1 \times 10^5$ samples from a reflected Brownian motion-based diffusion model in $\mathbb{P} \subset \mathbb{R}^3$.

(b) Plots of the univariate marginal and pairwise bivariate distributions of $1 \times 10^5$ samples from a reflected Brownian motion-based diffusion model in $\mathbb{T}^4$.

Figure 18: Visualisation of the distribution learned by a reflected Brownian motion-based diffusion model in $\mathbb{P} \subset \mathbb{R}^3 \times \mathbb{T}^4$ using univariate marginal and pairwise bivariate plots.

### F.5.4 Visualisation of samples from the uniform distribution

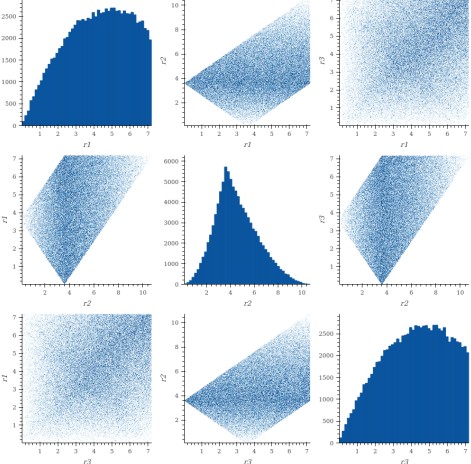 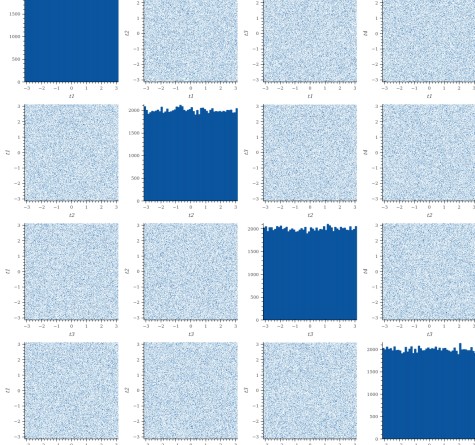

(a) Plots of the univariate marginal and pairwise bivariate distributions of $1 \times 10^5$ samples from the uniform distribution in $\mathbb{P} \subset \mathbb{R}^3$.

(b) Plots of the univariate marginal and pairwise bivariate distributions of $1 \times 10^5$ samples from the uniform distribution in $\mathbb{T}^4$.

Figure 19: Visualisation of the uniform distribution in $\mathbb{P} \subset \mathbb{R}^3 \times \mathbb{T}^4$ using univariate marginal and pairwise bivariate plots.

