# OpenReview forum: "Metropolis Sampling for Constrained Diffusion Models"
_NeurIPS.cc/2023/Conference — NeurIPS 2023 poster_

### Official Review · Reviewer_vCsC · 2023-06-30

**Soundness:** 2 fair
**Presentation:** 2 fair
**Contribution:** 2 fair
**Rating:** 3
**Confidence:** 4

**Summary:**

The paper presents a rejection-sampling technique applied to diffusion models when sampling defined on manifolds.

The paper contains some theoretical results on the convergence of the proposed algorithm, in particular the assymptotic convergence to the reflected Brownian motion. The also paper contains a few comparisons to the other methods, in particular, methods using regularization (such as barrier functionals) and reflected stochastic processes.

**Strengths:**

The paper applies the notion of rejection sampling to denoising diffusion models.
The proposed algorithm is very simple to modify to code, and it require much less intrucive modifications to the sampling algorithm compared to the alternatives the authors mention.

**Weaknesses:**

- It needs to be clarified what the algorithmic innovation is in the paper. Rejection sampling is a standard technique, and the application to diffusion models directly applies an old idea to a trendy topic. Also, it is not clear how valuable is the theoretical treatment. The authors show that the methodology is asymptotically equivalent to RBM, but then in the experiments, the authors claim that their algorithm outperforms RBM.

- The authors mention that scalability is one of the main advantages of its method. However, it is not clear what the scalability is with respect to. Also, for algorithms (which is the case here), scalability is often required to show how a bigger problem can be split into different pieces that can be run independently so that the overall runtime remains low. In this case, only one experiment showcasing scalability can be found in Section 5.1. The authors seem to claim their method is scalable with respect to the dimension, but still most of the wall-time are not that different. Along the same lines, I am not sure how interesting sampling from a cube (or a polytope) is in this case. I would assume that most of the "mass" of the distribution is along the boundary, which would make the problem easier, as seen in Table 2, where both methods start to do better as d increases.

- Most of the theory seems to be around Manifolds of co-dimension 0, which is only mildly interesting. There is not much theory when the manifold has a positive co-dimension. Also, there is a mismatch between the definitions of $\mathcal{M}$. In Line 72, it is defined as an open subset of another Riemannian manifold $\mathcal{N}$, which is not clearly defined, using several inequalities, and there are the assumptions in Line 207, which greatly simplifies the setup. What are the assumptions that this paper is considering?.

- The manifolds are smooth and of co-dimension 0; however, some of the experiments consider non-smooth manifolds, and others with manifolds of co-dimension greater than 0, thus, there seems to be a disconnect between the theory and experiments. Also, there is no quantitative metric for the sampling quality in section 5.3. They seem to "look" better than the very weak baseline of a uniform sampling, but it is not clear that the sampling does capture the correct statistics.

- In general, the experiments are fairly weak, particularly the baselines. Using a uniform distribution is by no means a baseline, and it should be used as a sniff test. One baseline that the authors could have used is the sample on a sphere using their algorithm (seen the sphere embedded in a 3-dimensional space) and using a diffusion model defined on the sphere. Such a comparison would make the point that it can outperform a method specifically tailored for the geometry. Also, the reference to the figures is not really informative, particularly when compared to weak baselines. I would prefer to have some quantitative information.

**Questions:**

See above.

**Limitations:**

The authors mentions some of the limitations of their work, and there is not direct adverse impact of this work.

---

> ### Author Rebuttal · Authors · 2023-08-09
>
> Thank you for taking the time to review our manuscript and for providing helpful and constructive feedback. We were happy to see your review emphasising the implementational simplicity of our approach, as well as the resuting ease with which it can be integrated into existing sampling algorithms.
>
> The concerns you raised in your review relate to the originality and scalability of our method, our choice of baselines, and the type of manifolds covered by our theoretical justification. We will address each of these points in turn below.
>
> ## Algorithmic Innovation
>
> > It needs to be clarified what the algorithmic innovation is in the paper. ... Also, it is not clear how valuable is the theoretical treatment.
>
> Rejection sampling is indeed a standard technique; in this manuscipt we present several contributions. First is exactly as the reviewers notes: an application of this old idea to a trendy topic, and evidence that using this idea in this setting is more effective than existing approaches in the literature. Second is the theoretical treatment we give which is necessary to justify the use of rejection sampling in the diffusion models we present. The asymptotic equivalence establishes the RBM is the continuous time process corresponding to the metropolis discretization; this means we can swap the metropolis discretization in for a reflected walk in reflected diffusion models using the same losses and time-reversal. This is highly non-obvious as the dynamics of the two discretizations are superficially quite different.
>
> > The authors show that the methodology is asymptotically equivalent to RBM, but ... claim that their algorithm outperforms RBM.
>
> Yes, this is because our approach affords much more tractable learning dynamics (see e.g. the discussion in Section 3.1), which enables it to unambiguously outperform the less tractable RBM-based diffusion models across a range of constraint geometries and applications, both in terms of empirical performance and computational efficiency.
>
> ## Scalability
>
> > The authors mention that scalability is one of the main advantages of its method. ...
>
> The proposed method is scalable in two senses: first it is much more efficient in general than the existing approaches in the literature, and hence is inherently _more_ scalable than existing approaches; second it is in fact scalable in exactly the way the reviewer notes when the underlying manifold is a product of constrained manifolds, e.g. a product of simplices or a product of SPD matrices with bounded trace. In this case we indeed have that the problem can be broken up into pieces and run independently, allowing straightforward scaling of this more efficient method in this specific regime.
>
> ## Empirical Evaluation
>
> > In general, the experiments are fairly weak, ... I would prefer to have some quantitative information.
>
> We apologize for the confusion and would like to emphasize that we benchmark our method against recently developed state-of-the-art constrained diffusion models, providing detailed empirical comparisons with models based on the reflected Brownian motion [1,2] and the log-barrier metric [2].
> As neither of these methods extends to manifolds with non-convex bundaries, we currently do not include a baseline for the wildfire example on the sphere. We only provide the uniform distribution as it is a relatively standard practice in diffusion papers to highlight what the invariant distribution looks like, especially for non-standard invariant distributions like the uniform over the constrained set we investigate here. Motivated by your comment, we have run an additional experiment on the sphere to provide a quantitative comparison to an unconstrained model. We just present MMD results here:
>
> | Model    | MMD | % in boundary |
> | - | - | - |
> | Unconstrained | $0.1567 \pm 0.013$ | $63.3%$ |
> | Constrained  | $0.1388 \pm 0.015 $  | $100%$ |
>
> We compute the RBF-MMD as described in the appendix, removing samples outside the boundary.
>
> ## Manifolds covered by our proof
>
> >Most of the theory seems to be around Manifolds of co-dimension 0, which is only mildly interesting. ... What are the assumptions that this paper is considering?.
> >
> > The manifolds are smooth and of co-dimension 0 ... there seems to be a disconnect between the theory and experiments.
>
> We agree with the reviewer that the theoretical results of Theorem 3 and Theorem 4 are stated for manifolds of co-dimension 0. We believe that the results can be extended to a more general setting. One approach to do so would be to replace the local parameterization of the manifold as $\\{x \in \mathbb{R}^d,\Phi(x) < 0\\}$ by similar inequalities in local charts. Using the paracompact property of the manifolds under consideration it seems possible to extend our results. However, this approach seems to be highly technical and we postpone it to future work.
>
> The definition in line 72 is to provide the broad class of manifolds which our method cover. Our experiments fall in the class of the manifolds defined in line 72. However, we greatly simplify the structure of the manifold to give the theoretical result in line 207. So to summarize:
> * The methodology is defined on the class defined in line 72. Experiments are defined w.r.t. manifolds defined in line 72.
> * In order to simplify the analysis, the validity of the discretization scheme is only assessed under stronger assumptions (as pointed out by the reviewer).
>
> We thank the reviewer for pushing us to clarify this point and will update the paper accordingly. We leave the extension of our theoretical results to future work (as highlighted above this extension is likely to be highly technical).
>
>
> ---
>
> We hope that our additional clarifications and discussion address all of your questions and concerns.
>
> Please let us know if you have any further questions!
>
> ---
>
>
> ## References
>
> [1] Lou and Ermon. "Reflected diffusion models." ICML 2023
>
> [2] Fishman et al. "Diffusion Models for Constrained Domains." TMLR, 2023

---

> > ### Comment · Reviewer_vCsC · 2023-08-17
> > **Thank you for the response.**
> >
> > - The proposed method is scalable in two senses: ...
> >
> > It seems that in the response of the authors claim that the are able to reduce the constants in the computational complexity, which does not render the method inherently scalable. As I mentioned before there is a clear definition of scalability for algorithms.
> > If the claim is scalability, which appears in the introduction when the properties of the method are being presented, then it should be backed up by experiments or at least a small section. In addition, following the argument of scalability of the proposed method when the "manifold is a product of constrained manifolds", this renders the following claim of the paper "Accurately modelling distributions on constrained Riemannian manifolds is a challenging problem with a range of impactful practical applications. In this work, we have proposed a mathematically principled and computationally scalable extension of the existing diffusion model methodology to this setting." misleading. This sentence seems to imply the scalability for any constrained Riemannian manifold, and from the authors' response, it seems to be the case on a subset.
> >
> > In addition, the authors claims that their method is more "numerically stable" but there is not evidence of this claim in the paper.
> >
> > - We believe that the results can be extended to a more general setting...
> >
> > In this case, I would strongly encourage the authors to properly nuance the introduction of the method along with the theoretical contributions. Otherwise, the sentence " Our core theoretical contribution is to show that this new discretisation converges to the reflected SDE by using the invariance principle for SDEs with boundary" is also misleading, given that the theoretical contribution is not with all Riemannian manifolds but only a subset.

---

> > > ### Author Response · Authors · 2023-08-18
> > > **Additional Experiments and Clarification**
> > >
> > > Thank you for taking the time to engage with our rebuttal. We were happy to see that our clarifications and additional experimental results addressed your concerns regarding the algorithmic innovation and empirical evaluation of our manuscript. We hope to address your remaining concerns regarding our claims of scalability, numerical stability and the generality of our theoretical justification below.
> > >
> > > &nbsp;
> > > # 1. Scalability
> > > We agree with the reviewer that we do not present a method that provably achieves the definition of scalability that they are outlining, nor do we currently present empirical evidence justifying the term scalable given that definition. To that end we carried out an additional experiment, running the forward process for a fixed number of discretization steps with a fixed $\beta$. The forward process is the only difference between the reflected and Metropolis diffusion models, so it is what explains the difference in runtime. We used the hypercube because it is easy to check for convergence of the random walk (by binning the hypercube with hyperrectangles and computing the relative vs. expected frequency). We tuned $\beta$ so that the discretized walk approximately converged for both the reflected and Metropolis random walks in the last steps of the walk. We then used the `timeit` utility to compute the following means and standard deviations for running the reflected and Metropolis random walks.
> > >
> > > | Dimension | Reflected Mean ± Std. Dev | Metropolis Mean ± Std. Dev | Ratio of Reflected / Metropolis |
> > > |:---:|:---:|:---:|:---:|
> > > |2|0.080&pm;0.000|0.020&pm;0.000|3.95713|
> > > |4|0.141&pm;0.001|0.036&pm;0.001|3.89396|
> > > |8|0.267&pm;0.001|0.069&pm;0.000|3.85959|
> > > |16|0.576&pm;0.023|0.130&pm;0.001|4.43903|
> > > |32|1.405&pm;0.027|0.205&pm;0.001|6.84785|
> > > |64|3.238&pm;0.040|0.300&pm;0.003|10.7962|
> > > |128|9.148&pm;0.075|0.491&pm;0.009|18.6326|
> > > |256|27.600&pm;0.434|1.004&pm;0.013|27.4842|
> > >
> > > As a preliminary analysis, we have fit this data with the symbolic regression package `Pysr`, yielding the following formulas for the Reflected `((0.0002893396 * d) + 0.03379444) * d` and Metropolis `0.003400629 * ((log(d) * 6.0751796) + d)` curves. These would correspond to complexities of $\mathcal{O}(d^2)$ and $\mathcal{O}(d + \log d)=\mathcal{O}(d)$ respectively, though we point out that they are only approximate empirical indicators of scaling beaviour.
> > >
> > > We hope that these results are convincing in justifying our use of the term "scalable". It is possible that these numbers would change by a small amount with a more careful tuning procedure for $\beta$, but expect the clear trend we observe to hold up. We will produce a more comprehensive table using an automated tuning procedure for even higher dimensions and other polytopes and additionally ensure that our claims of scalability are appropriately contextualized in light of these experimental results and your comments. We will also add a discussion of the product structure we describe above and the benefits we can expect in that regime, even though, as the reviewer is correct to note, it is not central to our claims of scalability.
> > >
> > > &nbsp;
> > > # 2. Numerical Stability
> > >
> > > The Metropolis method is exactly as numerically stable as a standard diffusion model on a Riemannian manifold since the constraints are handled by the binary constraint function. This is in stark contrast to the baselines we compare against, which all suffer from serious numerical instability issues:
> > > * The numerical instabilities of the log-barrier approach are so pathological they make it impossible to run the ODE evaluation, producing extreme values and NaNs.
> > > * While the numerical issues of the reflected method are less extreme, we found the operation of reflecting off the boundary to induce hard-to-resolve numerical instabilities even in the case of a polytope (never mind more complex geometries), which necessitated extensive hyperparameter tuning to resolve.
> > >
> > > We provide a brief discussion of these issues in Appendix C, but agree with the reviewer that we do not clearly articulate the numerical issues of the existing approaches. We will mention these issues explicitly in our discussion of the limitations of existing constrained diffusion models in Section 3.1 and add a more thorough discussion to Appendix C.
> > > &nbsp;
> > > # 3. Gaps Between Theory and Applications
> > > Thank you, we agree that a more nuanced presentation of our theoretical contributions is necessary to ensure the theory matches our claims. Below we outline the steps necessary to generalize the results; we will update the manuscript along the lines of the outline and make sure to adjust the respective sections of the Abstract and Introduction accordingly.
> > > &nbsp;
> > >
> > > We appreciate the thoroughness of your review and hope that these additional clarifications and experimental results address your remaining concerns. Please let us know if you have any further questions.
> > >
> > > Sincerely,
> > >
> > > The Authors

---

> > > > ### Author Response · Authors · 2023-08-21
> > > > **Extending our theory to more general manifolds**
> > > >
> > > > In what follows, we give a high level intuition for the extension of the proof to more general manifolds.
> > > >
> > > > In their approach [Theorem 6.3, 1] the authors consider the case where the manifold is given by $\Phi(x) < 0$. This indeed defines a manifold of codimension $0$ in $\mathbb{R}^d$. In our case a meaningful extension would be to consider the setting where the manifold of interest is defined via the inequality $f_i(x) < 0$ for $i \in \{1, \dots, N\}$ and $f_i: \ \mathcal{M} \to \mathbb{R}$. Note that contrary to the previous case: 1) Each $f_i$ is defined on a manifold 2) we have a finite number of $f_i$. This requires us to extend the proof in these two directions. First, we need to take into account the underlying geometry of $\mathcal{M}$. Where does this geometry show up in our proof? We rely on the Taylor expansion of the function $\Phi$ (now $f_i$) and therefore similar extensions should be considered in the manifold setting, see for instance [Lemma S.8, 2]. The other key place where geometry is key is when we use the notion of tubular neighbourhood to decompose the space in a tangential and a normal part. This decomposition is still valid in the manifold setting see [3]. In all of the proofs (and the assumptions of [1, Theorem 6.3]) the norm between elements should be replaced by the Riemannian metric of $\mathcal{M}$. Finally, one needs to extend the proof of [1] to the manifold setting. This is actually doable (since only smoothness arguments are used in [1] and we have already seen that they can be extended to the manifold setting using [Lemma S.8, 2] for instance). Finally the last modification is to consider a finite number of constraints instead of just one. This extension is straightforward both in our proof and in the framework of [1]
> > > >
> > > > [1] Stroock, Varadhan -- Diffusion processes with Boundary conditions
> > > >
> > > > [2] Durmus et al. -- On Riemannian Stochastic Approximation Schemes with Fixed Step-Size
> > > >
> > > > [3] Lee -- Introduction to Riemannian Manifolds

---

### Official Review · Reviewer_a8R9 · 2023-07-03

**Soundness:** 3 good
**Presentation:** 2 fair
**Contribution:** 3 good
**Rating:** 6
**Confidence:** 3

**Summary:**

The paper proposes a new discretization of the forward and backward SDEs used for learning and sampling in the diffusion generative model framework in manifolds with inequality constraints. The method relies on "metropolising" a given discretisation of the "unconstrained" manifold (either Euler-Maruyama if the manifold is $\mathbb{R}^d$ or Geodesic random walks if it's not). The discretisation is shown to converge weakly to the reflected SDE and is shown to be numerically simpler to implement and faster for diffusion models on manifolds.

**Strengths:**

The paper proposes a simple approach for sampling and training diffusion models on constrained manifolds. The approach is theoretically justified and is shown to be numerically more efficient than the state of the art. Thus, the proposed algorithm is an interesting tool for practitioners. The paper is mostly well written and accounts well for the state of the art. Several examples with increasing degree of complexity are shown.

**Weaknesses:**

The paper possesses some shortcomings in the presentation, namely some typos that I will refer to in the bottom part of this section.
There are also some fundamental things missing. For example, I could not find the code, since it's not given in the appendix and the link to the anonymous github was inactive when I tried to access it. This is a problem, specially when for example the paper do not explicitly tells how it calculates the log likelihood of the test samples, thus I'm admitting it uses the ODE formulation as presented for example in [1].

I also feel that some other metric would also be of great value. The paper uses only the log likelihood of the held out test set as a metric, but some other distribution related metric such as the sliced wasserstein can be applied in the first example, specially in the lower dimensional examples.

Some typos:
Section 3.1 Practical imitations → Pratical limitations
Proposition 5 repeat the definition of Z


[1] De Bortoli, V., Mathieu, E., Hutchinson, M., Thornton, J., Teh, Y. W., & Doucet, A. (2022). Riemannian score-based generative modelling. Advances in Neural Information Processing Systems, 35, 2406-2422.

**Questions:**

I feel that a lot of the implementation could be better explained, even though the algorithm is simple. The code should be made available and I would also like the authors to be more precise into how do they calculate the log likelihood of the test set, even though this can be considered somehow standard in the literature.

How the epochs were chosen for the log likelihood calculations? It'd be interesting to have a graph of computing time vs log-likelihood of the held out test set to understand how those behave in time for several dimensions for each algorithm (metropolised vs reflected).

The geospatial data example seems to show that the learned distribution is « coarser » than the target distribution and has some concentration around the boundaries. Is this something to expect? Maybe rejecting a lot of steps in the more non convex parts of boundaries lead to an augmentation of the likelihood?

---

> ### Author Rebuttal · Authors · 2023-08-09
>
> Thank you for taking the time to review our manuscript and for providing helpful and constructive feedback.
> We were happy to see your review emphasising the improved computational efficiency and empirical performance of our approach and acknowledging its usefulness to a range of practitioners.
>
> The concerns you raised in your review relate to the availability of our code, the log-likelihood evaluation and the use of an additional performance metric. We will address each of these points in turn below.
>
> ---
>
> ## Code Availability
>
> > For example, I could not find the code, since it's not given in the appendix and the link to the anonymous github was inactive when I tried to access it. This is a problem, specially when for example the paper do not explicitly tells how it calculates the log likelihood of the test samples, thus I'm admitting it uses the ODE formulation as presented for example in [1].
> >
> > The code should be made available and I would also like the authors to be more precise into how do they calculate the log likelihood of the test set, even though this can be considered somehow standard in the literature.
>
>
> We have checked the anonymised links on multiple different devices and they seem to work correctly, as they did when we originally added them to the manuscript. If the code anonymisation site was down at any point during the review period, we apologise for the inconvenience. Hopefully you can access them now.
>
> ## Additional Performance Metric
>
> > The paper uses only the log likelihood of the held out test set as a metric, but some other distribution related metric such as the sliced wasserstein can be applied in the first example, specially in the lower dimensional examples.
>
> We would like to point out that we present further evaluations in Appendix F.1 using the MMD distance, as well as a comparison to an additional baseline method presented in [1]. If you would like us to additionally compare the models using the sliced Wasserstein distance, we are happy to do so during the author discussion period. The agreement between the MMD and the log-likelihood seems to signal the log-likelihood is a reasonable metric.
>
> ## Log-likelihood Evaluation
>
> >  I would also like the authors to be more precise into how do they calculate the log likelihood of the test set, even though this can be considered somehow standard in the literature.
>
> We will add details of how we compute the likelihood in the main paper, but it is indeed as an ODE in the reviewers reference. More precisely, we compute the log-likelihood using the equivalence between diffusion models and continuous normalizing flows (see [2,3] for instance). In the case of reflected diffusion models this equivalence was first used in [4]. As in continuous normalizing flow approaches, the log-likelihood computation is performed leveraging tools from Neural ODE [5].
>
> > How the epochs were chosen for the log likelihood calculations? It'd be interesting to have a graph of computing time vs log-likelihood of the held out test set to understand how those behave in time for several dimensions for each algorithm (metropolised vs reflected).
>
> We fixed the number of optimization steps for the two methods at a point when the optimization had converged and used the same end point for both. This choice is somewhat arbitrary, and we can easily add a plot showing how the log-likelihood evolves over the course of training during the discussion period if the reviewer would like.
>
> > The geospatial data example seems to show that the learned distribution is « coarser » than the target distribution and has some concentration around the boundaries. Is this something to expect? Maybe rejecting a lot of steps in the more non convex parts of boundaries lead to an augmentation of the likelihood?
>
> We agree that there may be some concentration around the non-convex parts of the boundaries. This is particularly interesting given the clear convergence of the forward process to the invariant distribution. We think it is likely that it is caused not by the Metropolis discretization but by the added complexity the score network must learn to "jump" out of the non-convex parts of the constrained set. It seems plausible that more careful architecture engineering could fix this problem, but we think the example in the manuscript serves as a proof of concept for the method.
>
> ---
>
> We hope that our additional clarifications and discussion address all of your questions and concerns.
>
> Please let us know if you have any further questions!
>
> ## References
>
> [1] Fishman et al. "Diffusion Models for Constrained Domains." TMLR, 2023
>
> [2] Song et al. "Score-Based Generative Modeling through Stochastic Differential Equations", 2021
>
> [3] Huang et al. "A Variational Perspective on Diffusion-Based Generative Models and Score Matching", 2021
>
> [4] Lou and Ermon. "Reflected Diffusion Models", 2023
>
> [5] Chen et al. "Neural Ordinary Differential Equations", 2018

---

> > ### Comment · Reviewer_a8R9 · 2023-08-16
> >
> > I thank the authors for the response to my questions. I do think the MMD is a better suited metric. I have also checked the link and it seems to be working now. Therefore I'm raising my score.

---

### Official Review · Reviewer_fvg9 · 2023-07-03

**Soundness:** 2 fair
**Presentation:** 3 good
**Contribution:** 3 good
**Rating:** 3
**Confidence:** 5

**Summary:**

This paper proposes a Metropolis sampling for constraint diffusion in the context of generative modeling. The authors show that the proposed algorithm is a discretization of reflected Brownian motion on manifold (with rejection). The authors also apply the proposed algorithm to several synthetic and real data sets.

**Strengths:**

This paper considers a Metropolis sampling algorithm on manifolds in light of the recent development of denoising diffusion models in generative modeling. The paper is well written, and I enjoyed reading it.

**Weaknesses:**

There are several weakness.

(1) As the authors pointed out themselves, the idea of (Metropolis) sampling on manifolds is much related to ball walk. There have been extensive work in this direction, see e.g. Dwivedi, Chen, Wainwright, and Yu's paper Log-concave sampling: Metropolis-Hastings algorithms are fast, JMLR, 2018. The authors failed to make connections to recent developments (and literature).

(2) In the MCMC literature, there are two general "discretization' techniques: ULA (unadjusted Langevein Algo) and MALA (Metropolis adjusted Langevin Algo). The proposed algorithm is close to the idea of MALA, which the authors failed to make connections.

(3) The main theorems (Theorem 2 & 4) are quite standard to experts in reflected Brownian motion. This generally from follows the techniques developed by Burdzy, Ramanan, Williams..., in the light of Stroock-Varadhan. Only Proposition 5 seems to be new (but not hard).

(4) One of the most important aspects of MCMC sampling is the mixing time (or convergence rate). This paper does not seem to provide any related analysis. Though on the continuous level this can be read from existing results, the mixing time in the discrete algorithm is important (especially concerning the choice of the step size $\gamma$).

**Questions:**

See weaknesses for my comments on the paper.

---

> ### Author Rebuttal · Authors · 2023-08-09
>
> Thank you for taking the time to review our manuscript and for the helpful and constructive feedback. We were happy to see you emphasize that we extend methods for generative modeling on constrained Riemannian manifolds [1,2] and are glad you enjoyed the paper.
>
> The concerns you raised relate to connections to existing prior art and an analysis of the mixing time of our algorithm. We will address each of these points in turn below.
>
> ## Prior Literature
>
> ### 1
>
> We agree with the reviewer that the proposed forward process is linked with the ball walk (and related algorithms). We do not claim that our algorithm is superior for sampling from an unnormalized density defined on the constrained manifold. We are aware that there is a strong line of works dealing this problem (see e.g. [10] exhibiting strong theoretical guarantees). One of the contributions of our paper is to showcase that we can use these methods to derive efficient numerical schemes for generative modeling under constraints. To the best of our knowledge, the ball walk and derived algorithms have not been applied to define forward noising process for diffusion model algorithms. Nevertheless, we will extend our discussion of related work with [3,7,8,9] and the references therein.
>
> ### 2
>
> We will add a paragraph making the connection with MALA and the distinction with ULA. However, our point of view is somewhat different than the usual motivation behind MALA, which, in the unconstrained setting, acts as a way to remove the bias from ULA, see Besag's comment: "if instead one uses s' merely as a Hastings proposal for the next state, then the usual acceptance probability ensures that p is an exact stationary distribution of the modified Markov chain" in [4]. Our proposal can be recast in this context but our motivation is different since we aim at a computationally efficient discretisation of the forward noising  of the reflected diffusion.
>
> ### 3
>
> This is an important point. We are not aware of any existing proof of Theorem 2 and Theorem 4. We would be grateful if the reviewer could point us towards such a reference. We agree that the proof is not difficult and that our work is an application of the techniques of Stroock and Varadhan to prove the sub-martingale property. However, we would like to make the following comments:
> * If the proof does not exist in the literature, we believe it is valuable for the community (and especially newcomers to the field, as not everyone in ML is familiar with reflected processes). We strongly believe that making such results available to the community will help to foster the interaction between the ML community and the MCMC community.
> * While the proof is technical (and again we agree that our aim is to verify the conditions of [3], and not introduce any novel techniques to show that a numerical scheme is a valid discretisation), the main difficulty is to obtain the lower bounds in Proposition 5 and to control the behavior of the drift near the boundary (see for example Proposition 20). To do so, we needed Theorem 7 which ensures that, provided enough regularity on the boundary, one can obtain such controls. To the best of our knowledge, this approach is new.
>
> We would appreciate if the reviewer could point us to a reference proving Theorems 2 and 4. Regarding Prop 5, we do not believe a result must be hard to serve as the theoretical foundation for a novel algorithm exhibiting state-of-the-art empirical performance.
>
> ## 4 Mixing Time
>
> We want to emphasize that our paper is not an MCMC paper. The only target we consider for the forward noising process is the uniform distribution on the constrained manifold under consideration. In practice, we set the parameter $T$ (running time of the forward noising process) so that, at time $T$, the distribution of the forward process is close to the uniform. While the parameter $T$ is important, it is easily tunable and in all of our experiments we observe fast mixing of the Markov chain. We agree that it would be valuable to obtain 1) quantitative non-asymptotic result regarding the convergence of the proposed scheme (note: we only target the uniform distribution) 2) bounds on the bias incur by the choice of the stepsize.
>
> However, in practice the mixing time and the discretization stepsize are not bottlenecks in performance of the algorithm. As shown in [5,6], the main source of error in denoising diffusion models (in the unconstrained setting) is the approximation of the score. This is because the convergence to the invariant distribution is exponential (with respect to the total variation distance for instance) and the bias incurred by the discretization stepsize is dominated by the neural approximation of the score.
>
> ---
>
> We hope that our additional clarifications and discussion address your questions and concerns.
>
> Please let us know if you have any further questions!
>
> ## References
>
> [1] Lou and Ermon. "Reflected diffusion models." ICML 2023
>
> [2] Fishman, et al. "Diffusion Models for Constrained Domains." TMLR, 2023
>
> [3] Dwivedi, Chen, Wainwright, and Yu's Log-concave sampling: Metropolis-Hastings algorithms are fast, JMLR, 2018
>
> [4] J. Besag (1994) --  "Comments on "Representations of knowledge in complex systems"
>
> [5] Sitan Chen, Sinho Chewi, Jerry Li, Yuanzhi Li, Adil Salim, Anru R. Zhang -- Sampling is as easy as learning the score: theory for diffusion models with minimal data assumptions (2022)
>
> [6] De Bortoli -- Convergence of denoising diffusion models under the manifold hypothesis (2022)
>
> [7] Vempala -- Geometric random walks: a survey
>
> [8] Cousins, Vempala -- Bypassing KLS: Gaussian Cooling and an $O(n^3)$ Volume Algorithm (2015)
>
> [9] L. Lovasz and M. Simonovits. Random walks in a convex body and an improved volume algorithm. (1993)
>
> [10] Kook, Lee, Shen, Vempala -- Condition-number-independent convergence rate of Riemannian Hamiltonian Monte Carlo with numerical integrators (2022)

---

> > ### Author Response · Authors · 2023-08-21
> > **Extending theoretical results to more general manifolds**
> >
> > We also want to note that in our response to Reviewer vCsC we additionally outline how to extend our results to a more general class of manifolds, which may be relevant to the reviewer's evaluation here.

---

### Official Review · Reviewer_u5aU · 2023-07-07

**Soundness:** 3 good
**Presentation:** 2 fair
**Contribution:** 3 good
**Rating:** 5
**Confidence:** 3

**Summary:**

This paper introduces a new approach for generative modelling on constrained Riemannian manifolds, building upon diffusion models. The proposed method implements a Metropolis sampling scheme and offers computational efficiency and improved empirical performance over previous models, particularly with increased constraint complexity. It provides a valid discretisation of the reflected Brownian motion and demonstrates its scalability and flexibility across several application domains.

**Strengths:**

The paper has good originality of incorporating MCMC type of sampling with diffusion model on constrained Riemannian manifolds.

The overall quality of the paper is good, the proposed method should be of interests to a group of researchers working on diffusion models.

**Weaknesses:**

The paper is a bit hard to follow, it would be helpful to further refine the presentation of the methodology sections. Specifically, it would be nice to have a diagram illustrating the relationship between diffusion model, Metropolis sampler, and constrained Riemannian manifolds.

In my view, some of the mathematics in section 2 can be moved to supplementary, too much symbols in this section makes readers to spend time on each definition and formula, and hard to understand the whole picture and the relationship between different components.

Some illustrations might need further refinement, for example, Figure 5 (b) and (c) it is very hard to tell the difference between them and it conveys little information.

Numerical comparison with existing methods and evaluation metrics are relatively limited, the proposed method is mainly compared with the reflected method. Limited metric, does log likelihood comprehensively reflect the main performance improvement for such different datasets (robotics and proteins) Especially the proteins dataset, the changes is relative small, i.e., from 15.2 to 15.33.

**Questions:**

MCMC types of approaches sometimes suffer from long 'burn-in', I am wondering whether the authors have consider this case and evaluate its impact to the performance of the proposed method

Is it possible that Metropolis will give some samples that are not good enough (not representative of the latent space) for diffusion model, and mislead the training, - as samples accepted by Metropolis doesn't guarantee they are good samples.

**Limitations:**

Limitations are stated, e.g., potential poor performance on certain constraint geometries. The future work sounds sensible.

---

> ### Author Rebuttal · Authors · 2023-08-09
>
> Thank you for taking the time to review our manuscript and for providing helpful and constructive feedback.
> We were happy to see your review emphasising the improved computational efficiency and empirical performance of our approach, as well as acknowledging the quality of the paper and its relevance to other researchers in this space.
>
> The concerns you raised in your review relate to the presentational clarity and empirical evaluation of the manuscript. We will address each of these points in turn below.
>
> ## Empirical Evaluation
>
> > Numerical comparison with existing methods and evaluation metrics are relatively limited, the proposed method is mainly compared with the reflected method. Limited metric, does log likelihood comprehensively reflect the main performance improvement for such different datasets (robotics and proteins) Especially the proteins dataset, the changes is relative small, i.e., from 15.2 to 15.33.
>
> We provide both additional comparisons to the Log-Barrier method proposed in [4] as well as comparisons using the MMD instead of the log-likelihood in Section F.1 of the Appendix, to which we refer at the beginning of Section 5. The agreement between the MMD and the log-likelihood signals the log-likelihood is a reasonable metric. It is true that for the proteins dataset the improvement is small; this is corroborated by both the visualizations and the MMD. The time to train (as well as the sampling time) is still a factor of 10 faster though, which is a crucial advantage for scaling this approach up to larger proteins.
>
>
> ## Presentational Clarity
>
>
> > In my view, some of the mathematics in section 2 can be moved to supplementary, too much symbols in this section makes readers to spend time on each definition and formula, and hard to understand the whole picture and the relationship between different components.
>
> While we agree that shortening the outline of the proof would make the paper more accessible to applications-driven researchers, we believe that the theory presented in Section 3.2 is one of the central contributions of the manuscript is, which is why we chose to include it. We believe the proof is interesting and useful for machine learning theorists working on non-standard diffusion models.
>
> > Some illustrations might need further refinement, for example, Figure 5 (b) and (c) it is very hard to tell the difference between them and it conveys little information.
>
> The difference between Figures 5b and 5c is hard to discern because the models fit the data distribution equally well - though, as mentioned in the previous section, our approach fits it slightly better and is 10 times faster. We believe that this makes any qualitative visual comparison difficult, and would like to mention that Figures 4 and 5 are mainly included to provide an intuitive visualisation of the data, learned and uniform distribution on constraint geometries that may not be well-known in the manuscript's intended target audience.
>
> > Specifically, it would be nice to have a diagram illustrating the relationship between diffusion model, Metropolis sampler, and constrained Riemannian manifolds.
>
> We agree that such a diagram would make the connection between Figure 1 and the methodology outlined in Sections 2 and 3 more apparent. We have added a figure to the appendix to clarify how these components are related and will refer to it in the main text.
>
> ## Questions
>
> > MCMC types of approaches sometimes suffer from long 'burn-in', I am wondering whether the authors have consider this case and evaluate its impact to the performance of the proposed method
>
>
> We want to emphasize that our paper is not a MCMC paper. We are only marginally concerned with the mixing time of the algorithm as our target is always the uniform distribution for the forward noising process. We agree that it would be valuable to obtain 1) quantitative non-asymptotic result regarding the convergence of the proposed scheme (however, we again emphasize that we only target the uniform distribution) 2) bounds on the bias incurred by the choice of the stepsize.
>
> However, in practice the mixing time and the discretization stepsize are not the bottlenecks of the performance of the algorithm. As shown in [1,2], the main source of error in denoising diffusion models (in the unconstrained setting) is the approximation of the network. This is because the convergence to the invariant distribution is exponential (with respect to the total variation distance for instance) and the bias incurred by the discretization stepsize is largely dominated by the neural network approximation of the score.
>
> > Is it possible that Metropolis will give some samples that are not good enough (not representative of the latent space) for diffusion model, and mislead the training, - as samples accepted by Metropolis doesn't guarantee they are good samples.
>
> This is a valid point that may warrant further exploration. However, we think that this is unlikely, as our central theoretical contribution shows that the Metropolis approach is a valid discretisation of the reflected Brownian motion, which was demonstrated to produce high-quality samples on the constrained geometries we consider [3,4].
>
> ---
>
> We hope that our additional clarifications and proposed changes to the layout of the paper address all of your questions and concerns.
>
> Please let us know if you have any further questions!
>
> ---
>
> ## References
>
> [1] Sitan Chen, Sinho Chewi, Jerry Li, Yuanzhi Li, Adil Salim, Anru R. Zhang – Sampling is as easy as learning the score: theory for diffusion models with minimal data assumptions (2022)
>
> [2] De Bortoli – Convergence of denoising diffusion models under the manifold hypothesis (2022)
>
> [3] Lou and Ermon. "Reflected diffusion models." ICML 2023
>
> [4] Fishman, et al. "Diffusion Models for Constrained Domains." TMLR, 2023

---

> > ### Comment · Reviewer_u5aU · 2023-08-18
> >
> > Thanks for the author's response. The answers indeed make it a bit more clear for me to understand the paper, especially the Uniform distribution one. Will keep the rating to borderline accept if the author could follow the response to improve the overall presentation of the methodology.

---

### Author Rebuttal · Authors · 2023-08-10

# General Response

We would like to thank all reviewers for the time and effort they have put into reviewing our manuscript and for the valuable and constructive feedback they have provided.

---

In our manuscript, we present a new method for generative modelling on constrained Riemannian manifolds that affords substantial gains in computational efficiency and empirical performance compared to the current state of the art [1, 2].

We are delighted to see that reviewers appreciated the practical significance of our method across a range of settings, highlighting that it is both "of interest to a group of researchers working on diffusion models" (**reviewer u5aU**) and an "interesting tool for practitioners" (**reviewer a8R9**).

We are also happy to see that reviewers appreciated the "good originality" (**reviewer u5aU**) and "theoretical justification" (**reviewer a8R9**) of our approach, noting that "the proposed algorithm is very simple to modify [and] code" (**reviewer vCsC**), "requires much less intrusive modifications to the sampling algorithm" (**reviewer vCsC**) and is "numerically more efficient than the state of the art" (**reviewer a8R9**).

Additionally, we are pleased that reviewers found the paper to be "well-written" (**reviewers fvg9 and a8R9**) and that they "enjoyed reading it" (**reviewer fvg9**).

---

The reviewers have also raised a range of points regarding different parts of the paper. To address these, we have responded with additional discussions and clarifications of our work under the respective reviews. We have also added a baseline model to the paper, which further shows the strength of the proposed method (discussed in response to **reviewer vCsC**). We hope that these remarks convincingly address the reviewers' concerns and are happy to answer any further questions.


Sincerely,

The Authors

---


## References

[1] Lou and Ermon. "Reflected diffusion models." ICML 2023

[2] Fishman et al. "Diffusion Models for Constrained Domains." TMLR, 2023

---

> ### Author Response · Authors · 2023-08-21
> **Summary of Additional Empirical Results and Changes to the Manuscript**
>
> &nbsp;
>
> As the author discussion period is coming to an end, we would like to thank all reviewers once again for the time and effort they have put into reviewing our manuscript and engaging with our rebuttal. We will briefly summarize the discussions we have had and the changes we will make to the manuscript below.
>
> ---
>
> Motivated by reviewer suggestions, we have carried out a range of explicitly requested additional experiments.
>
> **Additional Baselines:** We have trained an unconstrained diffusion model on the geospatial data within non-convex country borders, demonstrating that it is clearly outperformed by our constrained model in terms of a quantitative performance metric (MMD) of the samples within the constrained set. We strongly believe that this result corroborates the state-of-the-art empirical performance that our method displays in all of the other settings we evaluated it in and will add the following table to Section 5.3 of our paper.
>
> | Model         | MMD  | % in boundary |
> | ------------- | ---- | ------------- |
> |Unconstrained|0.1567&pm;0.013|63.3|
> | Constrained | **0.1388&pm;0.015** |100|
>
> &nbsp;
>
> **Evidence of Scaling Behaviour:** We have conducted a more extensive runtime analysis of the existing Reflected diffusion process and our proposed Metropolis approach to substantiate our claims of scalability and to demonstrate the superior scaling behaviour that our method exhibits as the dimensionality of the problem increases. As part of our preliminary analysis, we have fitted this data using the symbolic regression package `Pysr`, resulting in the following formulas for the Reflected model `((0.0002893396 * d) + 0.03379444) * d` and Metropolis `0.003400629 * ((log(d) * 6.0751796) + d)` models. These formulas correspond to complexities of $\mathcal{O}(d^2)$ and $\mathcal{O}(d + \log d)=\mathcal{O}(d)$ respectively, though we emphasize that they are only an approximate empirical indicator of scaling behaviour.
>
> | Dimension | Reflected Mean ± Std. Dev | Metropolis Mean ± Std. Dev | Ratio of Reflected / Metropolis |
> |:---:|:---:|:---:|:---:|
> |2|0.080&pm;0.000|0.020&pm;0.000|3.95713|
> |4|0.141&pm;0.001|0.036&pm;0.001|3.89396|
> |8|0.267&pm;0.001|0.069&pm;0.000|3.85959|
> |16|0.576&pm;0.023|0.130&pm;0.001|4.43903|
> |32|1.405&pm;0.027|0.205&pm;0.001|6.84785|
> |64|3.238&pm;0.040|0.300&pm;0.003|10.7962|
> |128|9.148&pm;0.075|0.491&pm;0.009|18.6326|
> |256|27.600&pm;0.434|1.004&pm;0.013|27.4842|
>
> &nbsp;
>
> In addition to these experimental results, we have clarified how our approach relates to prior work, and we will ensure to include this discussion in Section 4 to properly contextualize the significance of our theoretical contributions.
>
> Similarly, we have provided reviewers with an outline of how we aim to extend our proof to more general manifolds, which we will feature in Section 3.2 and Appendix D of our manuscript. Furthermore, we will adjust our claims in the Abstract and Introduction of our manuscript to ensure that they adequately represent the classes of manifolds covered by our proof.
>
> Finally, we will incorporate the reviewers' feedback regarding the clarity of our paper's presentation into the draft to present the strengths of our method as clearly as possible.
>
> ---
>
> We are pleased that these additional experimental results, clarifications and proposed changes to the paper have moved two reviewers to increase their scores and appreciate the actionable feedback that the reviewers have provided.
>
> Sincerely,
>
> The Authors

---

### Decision · Program_Chairs · 2023-09-21

**Decision:**

Accept (poster)

**Comment:**

The paper offers a novel approach to challenging issues in generative modeling by extending by now standard denoising diffusion probabilistic models (DDPM) to Riemannian manifolds. Efficient solutions for handling manifolds defined by numerous constraints— commonly faced in various applied sectors like natural sciences— remain outside the reach of most existing methodologies. Previous works attempted to address this problem through complex noising processes (eg. introducing complex project steps), leading to higher computation demands.

The proposed strategy in this paper introduces a simplified noising scheme based on rejection sampling (i.e. simply reject "increments" when stepping out of the boundaries), contributing significant improvements in computational efficiency and empirical performance. The independent analysis proving the validity of this new process as a discretization of the reflected Brownian motion is both technically sophisticated and interesting (while the proposed method is very simple). Despite its simplicity, the method's effectiveness is convincingly demonstrated by numerical experiments that show that it can substantially outperform the current state-of-the-art. The authors manage to convincingly apply the proposed method to a range of problem settings.

Some reviewers proposed a relatively low score. Nevertheless, after careful reading of these reviews, it was concluded that they were either not relevant to the paper (eg. discussion of other MCMC schemes or mixing times considerations), or convincingly answered by the authors during the rebuttal phase (eg. numerical stability, scalability, numerical experiments).

In conclusion, the paper's proposed solution, with both its theoretical analysis and its proven practical impact, makes a strong case for acceptance.